# Analysis of the Coupling Effect and Space-Time Difference between China’s Digital Economy Development and Carbon Emissions Reduction

**DOI:** 10.3390/ijerph20010872

**Published:** 2023-01-03

**Authors:** Nan Li, Beibei Shi, Rong Kang

**Affiliations:** 1School of Economics and Management, Northwest University, Xi’an 710127, China; 2Carbon Neutrality College (Yulin), Northwest University, Xi’an 710127, China; 3National & Local Joint Engineering Research Center of Carbon Capture and Storage Technology, Northwest University, Xi’an 710069, China; 4Shaanxi Key Laboratory for Carbon Neutral Technology, Northwest University, Xi’an 710069, China

**Keywords:** digital economy, carbon emissions reduction, space-time difference, source of difference, unbalanced development, coupling and coordination degree, Dagum Gini coefficient, spatial correlation

## Abstract

Previously conducted studies have established that the digital economy has a one-way inhibition effect on carbon emissions. Against this background, this paper aims to analyze the coordinated development effect of the interaction between the digital economy and carbon emissions reduction. The entropy weight method, coupling and coordination degree model, Dagum Gini coefficient and Moran’s I index have been carried out as research methods in this paper. The results showed that: (1) The coupling and coordination of China’s digital economy and carbon emissions reduction shows an overall growth trend, but the coupling and coordination among regions, provinces and cities show a large imbalance. (2) In the sample period, the overall difference in the coupling and coordination between digital economy development and carbon emissions reduction shows an expanding trend, and the overall difference results are attributed to regional differences. (3) There is a significant spatial correlation in the coupling and coordination degree of digital economy development and carbon emissions reduction among cities. The paper systematically grasps the status of coupling and coordination development, the source of difference and spatial correlation between the digital economy and carbon reduction in Chinese cities. A dependence relationship has been established, which is digital economy development and carbon emissions reduction, and an interactive promotion pattern has been revealed between the digital economic system and the carbon emissions reduction system.

## 1. Introduction

A digital economy is an economic system with technology as its core driving force, and the digitalization of economic activities is a typical feature of digital economic systems. The new development mode led by technological innovation-driven digital economy development has laid the foundation for the coordinated promotion of digital economy and green development. The digital economy relies on technological innovation to continuously promote industrial integration [1] and economic restructuring, while the transformation of industrial and economic development modes will directly cause changes in carbon dioxide emissions [2]. At the same time, against the background of “emission peak” and “carbon neutrality,” the digital regulation on which the carbon emissions reduction process is based will also have a certain impact on the development of the digital economy. Therefore, it is urgent to systematically answer the interactive development relationship between the digital economy and carbon emissions reduction so that the development of the digital economy can support the advancement of carbon emissions reduction.

Specifically, what is the status of the interaction between digital economy development and carbon emissions reduction? Are there differences in different regions, provinces and city types? Is there any spatial “spillover” impact on the interaction between the digital economy and carbon emission reduction among regions? In view of this, this paper calculates the degree of coupling and coordination between the digital economy development system and the carbon emissions reduction system. Based on this result, this research identifies the differences among regions, provinces and cities. Finally, the spatial relevance of the degree of coupling and coordination between the digital economy and carbon emissions reduction has been tested. It is helpful to provide policy enlightenment and theoretical guidance for China to systematically review the relationship between digital economy development and carbon emissions reduction, adjust the layout of digital economy development and explore diversified carbon emission reduction paths.

Relying on the new digital infrastructure, the digital economy takes knowledge and information as the key production factors, modern information networks as the carrier and the use of information technology as the key driving technology for development, which profoundly affects and fundamentally changes the economic development mode and social activity structure [3]. The analysis of the interaction between the digital economy and carbon emissions reduction is the analysis of the coordinated development between the digital economy and carbon emissions reduction. Coordinated development includes two explanations. On the one hand, from the perspective of coordination, it emphasizes the positive interaction between digital economy development and carbon emission reduction and the formation of a virtuous circle situation that promotes each other. Coordination enables the two systems to work together effectively and sustainably [4,5]. On the other hand, from the perspective of development, it emphasizes the promotion of the digital economy and carbon emissions reduction. The coordinated development between the digital economy and carbon emissions reduction requires that digital economy elements can help promote the process of carbon emissions reduction through technology, which in turn can continue to provide technical and environmental support for digital development. This reflection of the connection between the two systems is coupling [6].

Data show that the market size of China’s digital economy has increased from 31.3 trillion yuan in 2018 to 40.5 trillion yuan in 2020, with a growth rate of 29.39%. From the perspective of data center construction, the market size of China’s data centers increased by 195.8 billion yuan by 2020, an increase of 25.27% compared with the previous year. For the development of 5G, China built 1.425 million 5G base stations by 2021, accounting for more than 60% of the world’s total. Compared with the development of the digital economy, China’s carbon emissions reduction situation is not optimistic. According to the data released by “Global Carbon Atlas,” China’s carbon emissions increased from 2421 tons in 1990 to 9751 tons in 2017, with a growth rate of 302.77%, its share in global total carbon emissions increased from 10.67% in 1990 to 27.32% in 2017. There is a long way to go to promote the carbon emissions reduction process [7]. Therefore, promoting the research on the status and trend of coordinated development of the digital economy and carbon emissions reduction is of practical significance for improving the layout of digital economy elements and promoting green development through digitalization.

At present, the research on the development of the digital economy has been mainly shown in two aspects: On the one hand, the connotation of the digital economy has been intensively discussed. It is believed that the real digital economy refers to the part of economic output completely determined by digital technology, with a business framework based on digital services and goods [8]. The integration of communication and computing technologies in the network and the flow of technology and data strengthens the transformation of e-commerce and large-scale business [9]. Computer-aided data flow has enabled digitization to transform various parts of the current economy [10]. Therefore, the digital economy is also divided into four different parts: digital services and goods, mixed digital services and goods, IT-based intensive production services, and IT industry [11]. It is believed that the digital economy is an economic activity based on data application and data technology innovation, with data as the core element [12]. Data technology is the basic support for the development of the digital economy. It is a new general technology, mainly through the combination of digital information and the Internet [13], including hardware, software and network technology [14]. There is an opinion that the digital economy is a more advanced economic form after the industrial economy [15]. 

On the other hand, it focuses on the analysis of digital economy measurement. Most of the existing literature measures the digital economy based on the method of summarizing the digital industry output, digital product classification, and digital-related comprehensive indicators. Scholars first discussed the boundary problem of the digital economy, which provides a useful reference for measuring the digital economy [16]. The digital economy is also divided into five parts for measurement, namely infrastructure, e-commerce, industry structure, digital labor and digital price [17]. Some organizations have proposed an accounting framework with intelligent infrastructure investment, social promotion, innovation release, growth and employment as the main body to measure the digital economy [18]. In addition, there are also scholars who calculate the digital economy by building a comprehensive indicator system [19,20,21]. 

The research on carbon emissions reduction mainly involves two directions: on the one hand, the measurement of carbon emissions. The calculation of carbon dioxide emissions is mainly based on different energy consumption. China’s industrial energy carbon emissions have been calculated based on industrial energy consumption [22]. Based on the IPCC inventory method, research uses the multiplication method of different fuel consumption and fuel carbon emission factors to calculate the regional power carbon emissions [23]. A study took a telecommunications company in Slovenia as the research object, calculated the greenhouse gas emissions within enterprise scope 3, and took the purchase of electricity, the commuting of employees, the use of vehicles owned by the company and heating as the measured objects [24]. In addition, the research calculates the grid emission factors, deduces the emission factors of specific European countries, and calculates the carbon intensity based on the power generation of power plants [25]. The other is the exploration of the carbon emissions reduction path. It is generally believed that economic development is closely related to carbon emissions [26]. The exploration of the carbon emissions reduction path mainly involves the improvement of the carbon emissions trading market [27,28], the analysis of the policy effect of the carbon emissions reduction pilot [29,30] and the research of other paths [31]. In the study of other carbon emissions reduction paths, another scholar believed that carbon capture, utilization and storage technology (CCUS) is the only technological path to achieve net zero emissions [32]. Another study showed that the promotion of “Nature-Based Solutions” (NBS) could provide a positive reference for China to solve climate and environmental problems [33]. Meanwhile, another study carried out from the perspective of energy showed that the development of renewable energy is of great significance for carbon emissions reduction in resource-based countries [34]. Another study assessed the existing climate policy portfolio and believed that the energy technology portfolio needs to be adjusted [35]. Some scholars discussed the environmental impact of mineral mining based on the mining industry [36,37]. The adjustment of savings also contributes to the development of a green economy [38]. In addition, international climate cooperation and climate assistance can also effectively achieve carbon emissions reduction targets [39].

The existing literature on the relationship between the digital economy and carbon emission reduction mainly starts from two aspects. One is the impact of digital economy development on carbon emissions. Some scholars believe that the digital economy can inhibit carbon emissions [40,41,42]. A study has proved the role of Taiwan’s digital development in the circular economy. It is found that through digital waste management, the consumption of raw materials can be reduced by 25% by 2030, and half of greenhouse gas emissions can be avoided [43]. One study showed that ICT could contribute to the construction of low-carbon cities [44], mainly through the governance capacity of the government [21] and energy systems [45]. Studies have also targeted African countries and found an inverted “U” relationship between ICT and carbon emissions in the sample countries [46]. In addition, some scholars believe that R&D investment plays a positive role in regulating the relationship between digital economy development and carbon emissions reduction [47,48,49]. It can be seen that the development of the digital economy has a positive contribution to the promotion of China’s green development process [50]. Therefore, it is necessary to study the impact of the digital economy on the climate and environment, which has reference significance for the formulation and improvement of economic and environmental policies [51]. Different from this conclusion, other scholars believe that the development of the digital economy is unfavorable to carbon emission reduction [52,53]. The other is the impact of low-carbon development on the digital economy. Currently, there are relatively few studies on this aspect. Low carbon development of an economy can improve the level of digital transformation of enterprises. Specifically, low-carbon development-oriented environmental laws and regulations can compensate for driving enterprises’ digital transformation and innovation. It can be considered that low-carbon development can help reverse enterprises’ digital transformation [54].

By sorting out relevant domestic and foreign studies, this paper finds that most of the existing literature focuses on the unilateral relationship between digital economy development and carbon emissions based on the measurement of digital economy and carbon emissions. At present, there is still a gap in the discussion of the interaction effect of the digital economy and carbon emissions reduction. It can be noted that the relationship between the digital economy and carbon emissions reduction is a very topical issue. Therefore, the purpose of this study is to identify the interaction and coordination effect of the digital economy and carbon emissions reduction. The application of digital technology can optimize the end treatment technology of enterprise carbon emissions [55,56] and promote carbon emission reduction [57,58]. The promotion of the enterprise of the carbon emission reduction process brings about internal technology innovation and the strengthening of external digital supervision and management, which provide good technical environment support for the development of the digital economy. Therefore, it is necessary to study the coupling and coordination relationship between digital economy development and carbon emissions reduction. To achieve this, it is necessary to solve the following tasks: (1) it is necessary to measure and comprehensively grasp the status and space-time characteristics of the coupling and coordination between China’s digital economy development system and carbon emissions reduction system; (2) the source of the difference for the coupling and coordination between China’s digital economy and carbon emissions reduction needs to be explored; (3) the spatial “spillover” effect of the coupling and coordination of digital economy development and carbon emissions reduction among cities is tested.

Based on this, the marginal contributions of this paper are as follows: (1) from the perspective of system theory, this paper constructs a coupling coordination degree model of digital economy development and carbon emissions reduction and deeply identifies the interaction between digital economy and carbon emissions reduction, making up for the lack of one-way perspective on the relationship between the two. (2) Starting from eight regions, different provinces and different city types, the paper compares and analyzes the space-time difference of the coupling and coordination degree between the digital economy development system and the carbon emissions reduction system and identifies the source of the difference based on the Dagum Gini coefficient, which systematically presents the coordination characteristics of the interaction between the digital economy development and carbon emission reduction in China’s local areas, unifying the analysis from the national and local perspectives. At the same time, it also identifies the sources of differences and provides a useful reference for improving the development layout of various elements of the digital economy. (3) Moran’s I index is used to measure the spatial correlation of the coupling and coordination between digital economy development and carbon emissions reduction. From the perspective of spatial correlation, the existence of “spillover” effects of different regions’ coupling and coordination is identified, which complements the research on the spatial interaction of the coupling and coordination between digital economy and carbon emissions reduction in different regions, and provides decision-making basis for the dynamic linkage of policies in different regions’ digital economy to help green development.

## 2. Research Design

### 2.1. Index System Construction

#### 2.1.1. Digital Economy Development System and Carbon Emissions Reduction System

For the digital economy, scholars believe that there are three layers from the inside to the outside [59]. The core circle involves the improvement of digital infrastructure, the middle circle involves the construction of digital services and platform economy, and the outer circle involves the development of digital economic activities. The core circle of the digital economy provides basic support for the development of the digital economy and is the carrier of the development of the digital economy. The middle circle of the digital economy can maintain the stability of the development of the digital economy and provide an intermediary platform that is a guarantee of the development of the digital economy. The outer circle of the digital economy is the final embodiment of the activity and application level of the digital economy and the reflection of the development of the digital economy. The detailed relationship is shown in Figure 1.

In view of this, this paper builds a digital economy development system based on the three layers of the digital economy. Specifically, the indicator system of digital infrastructure includes: ① the government attaches importance to the development of the digital economy. The government’s regional development planning determines the direction and focus of urban development. Therefore, it is measured according to the frequency of big data, cloud computing, artificial intelligence, 5G, industrial Internet and other words related to the digital economy in the prefecture-level report on the work of the government. ② Emphasis on human capital. High-level human capital can provide knowledge and technical support for the improvement of digital infrastructure. Therefore, it is measured according to the proportion of urban education expenditure. ③ The degree of regional digitization. It is measured according to the digitalization index of prefecture-level cities. ④ The number of enterprises engaged in e-commerce transactions. The number of enterprises based on Internet development can represent the application level of digital infrastructure [60]. Therefore, it is measured according to the number of e-commerce transaction enterprises in each province. ⑤ The number of enterprises with digital economy characteristics. The total number of enterprises with digital economy label characteristics in prefecture-level cities is measured by adding.

The construction indicator system of digital services and platform economy includes ① the number of computer software employees. The number of Internet practitioners can reflect the scale of external services required for the development of the digital economy. Therefore, it is measured according to the number of computer software practitioners in prefecture-level cities. ② Software business income. Software business income can reflect the degree of regional demand for the Internet and is a direct reflection of digital services. Therefore, it is measured according to the amount of software business income of each province. ③ E-commerce sales. The development of e-commerce can directly reflect the sales level of the platform economy. Therefore, it is measured according to the e-commerce sales of each province. ④ The number of domain names. The number of domain names can measure the development of the Internet and is the basis for the development of digital services and platform economy. Therefore, it is measured according to the number of domain names in each province. ⑤ The number of websites. The number of websites can directly reflect the scale of digital services and is also the basis for the development of the platform economy. Therefore, it is measured according to the number of websites in each province.

The development indicator system of digital economic activities includes ① the telecommunication service level. Telecom business is an economic activity driven by e-commerce, so it is calculated according to the revenue of telecom businesses in prefecture-level cities. ② The number of mobile phone users. Mobile phone users are also activities driven by e-commerce. Therefore, the measurement is carried out according to the number of mobile phone users in prefecture-level cities. ③ Number of Internet users. The number of Internet users can reflect the development of digital economy-related businesses [61]. Therefore, it is measured according to the number of Internet users in prefecture-level cities. ④ Express delivery volume. The development of express delivery and e-commerce rely on each other, and they are all algorithmic-driven digital economic activities. Therefore, the measure is carried out according to the express delivery income of each province. ⑤ Digital inclusive financial index. The general index of inclusive digital finance can reflect the development trend of regional digital finance, which is different from traditional financial services. It is closely related to digital economic activities driven by algorithms. Therefore, it can be calculated according to the digital inclusive financial index of prefecture-level cities based on the compilation of indicators by [62].

The calculation of the carbon emissions reduction system mainly refers to the construction method of the green development system, and the sub-index construction is based on the input-output model [63]. Low carbon is one of the core concepts of green development [64], so it is reasonable to use the green development system for reference in the construction of carbon emissions reduction system. Specifically, among the input elements, this paper selects the capital element represented by the proportion of science and technology expenditure in financial expenditure, the technical element represented by the number of patent applications, the energy elements represented by renewable energy power generation and the social elements represented by green space area as the input elements of carbon emissions reduction. Output factors are measured by the economic benefits represented by carbon productivity. Carbon productivity can be understood as the GDP generated by unit carbon emissions. Therefore, the premise for obtaining carbon productivity data is to calculate the carbon emissions of each city. For the carbon dioxide emissions accounting of prefecture-level cities, this paper selects the carbon emissions of industrial consumption of natural gas, liquefied petroleum gas and industrial coal-fired electricity to sum up [65]. The specific calculation formula is as follows:(1)$$CO_{2}=gasCO_{2}+oilCO_{2}+electricCO_{2}=\vartheta \times gas+\tau \times oil+\omega \times \left(\varphi \times electric\right)$$

In Formula (1), $CO_{2}$ is the total carbon emissions of prefecture-level cities, $gasCO_{2}$ is the carbon emissions of natural gas consumed by industry, $oilCO_{2}$ is the carbon emissions of liquefied petroleum gas consumed by industry and $electricCO_{2}$ is the carbon emissions of electricity consumed by industry. ϑ is the carbon emission coefficient of natural gas, and the coefficient is 2.1622 kg/m^3^. τ is the carbon emissions coefficient of liquefied petroleum gas, and the coefficient is 3.1013 kg/kg, and gas and oil represent the consumption of natural gas and liquefied petroleum gas, respectively. ω is the greenhouse gas emission coefficient of the coal power fuel chain, and the coefficient can be expressed as 1.3023 kg/kWh in terms of equivalent carbon emissions φ represents the proportion of coal-fired power generation in the total power generation, which is dynamically adjusted according to the annual proportion change. electric refers to industrial power consumption. On this basis, this paper calculates the carbon emissions of each prefecture-level city from 2011 to 2018.

Based on the above analysis, the specific indicators selected for the development of the digital economy and carbon emissions reduction are shown in Table 1.

#### 2.1.2. Data Sources

This paper takes 283 prefecture-level cities in China from 2011 to 2018 as the research object, and the index data used are from the China Statistical Yearbook, China City Statistical Yearbook, China Statistical Yearbook on Environment, China Power Yearbook, the website of the National Bureau of Statistics, the national enterprise credit query website, the websites of municipal governments, the websites of Institute of digital Finance Peking University.

### 2.2. Research Method

#### 2.2.1. Measurement of the Level of Coupling and Coordination between Digital Economy Development and Carbon Emissions Reduction

Coordination means that two or more systems are interrelated and properly coordinated to form a mutually beneficial development situation [66,67]. In order to identify the coordinated relationship between China’s urban digital economy and carbon emissions reduction, this paper uses the coupling and coordination degree to measure the interaction between the digital economy development system and the carbon emissions reduction system in 283 prefecture-level cities in China. The specific measurement process of the coupling and coordination degree is as follows:(1)Calculation of comprehensive development level of the digital economy development system and carbon emission reduction system;(2)In order to ensure the consistency of the data set in terms of quantity and dimension;(3)this paper first gives priority to the standardized processing of data. The process is shown;(4)in Formula (2):
(2)$$Y_{ij}=\left\{\begin{array}{c} \frac{X_{ij}-min\left(X_{ij}\right)}{max\left(X_{ij}\right)-min\left(X_{ij}\right)},\;\;X_{ij}\;is\;a\;positive\;indicator\\ \frac{max\left(X_{ij}\right)-X_{ij}}{max\left(X_{ij}\right)-min\left(X_{ij}\right)},\;\;X_{ij}\;is\;a\;negative\;indicator \end{array}\right.$$

In Formula (2), *i* represents the city, *j* represents the specific indicator, $X_{ij}$ is the original index value, and $Y_{ij}$ is the value of the index after standardization. $min\left(X_{ij}\right)$ and $max\left(X_{ij}\right)$ represent the corresponding minimum and maximum values in the original indicator, respectively. $Y_{aij}$ and $Y_{bij}$ represent the standardized indicator values of the digital economic system and carbon emission reduction system, respectively. On the basis of the standardized data set, the information entropy of the digital economy development system and the carbon emission reduction system ($E_{j}$) are calculated as shown in Formulas (3) and (4):(3)$$P_{ij}=\frac{x_{ij}}{\sum_{i=1}^{m}x_{ij}}$$

In Formula (3), $P_{ij}$ represents the contribution of element *i* in indicator *j* to the indicator.
(4)$$E_{j}=-ln{\left(m\right)}^{-1}\sum_{i=1}^{m}P_{ij}ln\left(P_{ij}\right)$$

In Formula (4), $0\leq E_{j}\leq 1$. When the contribution of each element in the sub-index is the same, $E_{j}$ is 1. Based on the information entropy $E_{j}$ of each indicator, the corresponding weight of each indicator is calculated here. See Formula (5) for details:(5)$$W_{j}=\frac{1-E_{j}}{\sum_{j=1}^{m}\left(1-E_{j}\right)}$$

Formula (5) shows the weighting method of index *j* weight, $W_{aj}$ and $W_{bj}$ are the weight of the *j* index of the digital economy system and carbon emissions reduction system, on which the comprehensive development level of the digital economy system and carbon emissions reduction system is obtained. See Formula (6) for details:(6)$$Z_{i}=\sum_{i=1}^{n}W_{ij}\times Y_{ij},\;\;\sum_{i=1}^{n}W_{ij}=1$$

(5)Calculation of the coupling and coordination degree between the digital economy development system and carbon emissions reduction system;(6)The coupling and coordination degree model between multiple systems is con(7)structed as shown in Formula (7):


(7)
$$C(Z_{1},Z_{2},\cdots ,Z_{M})=n\times {\left[Z_{1}\times Z_{2}\times \cdots Z_{M}/{\left(Z_{1}+Z_{2}+\cdots Z_{M}\right)}^{M}\right]}^{1/M}$$


(8)In Formula (7), M represents the number of systems. In order to avoid the deviation

(9)of assessment results caused by the small measurement results of the digital economy development system, $Z_{a,}$ and carbon emissions reduction system, $Z_{b}$ [68], this paper further(10)builds a coupling and coordination model, and the specific model is established as shown(11)in Formula (8):


(8)
$$\;\;\;D_{ab}={\left(C_{ab}\times T_{ab}\right)}^{1/2}\;T_{ab}=\alpha_{1}Z_{a}+\alpha_{2}Z_{b}$$


In Formula (8),$\;D_{ab}$ represents the coupling and coordination of the digital economy development system and carbon emissions reduction system, and $T_{ab}$ represents the comprehensive development level of the digital economy development system and carbon emissions reduction system, and $\alpha_{1}$ and $\alpha_{2}$ are weight, respectively.

Furthermore, according to the coupling coordination level, $D_{ab,}$ of the digital economy development system and carbon emissions reduction system, this paper divides the coupling coordination into five types, namely, unbalance, light coupling and coordination, medium coupling and coordination, high coupling and coordination, and high-quality coupling and coordination. The specific division criteria are shown in Table 2:

#### 2.2.2. Spatial Difference Measure

In this paper, the Dagum Gini coefficient is used to conduct an in-depth analysis of the intra-group differences, inter-group differences, and sources of differences in the coupling and coordination of digital economy development and carbon emissions reduction in eight regions of China. The Dagum Gini coefficient and its decomposition method are used to decompose the overall Gini coefficient *G* into intra-group (intra-regional) differential contribution, $G_{w}$, inter-group (inter-regional) difference contribution, $G_{nb,}$ and hypervariable density contribution difference $G_{l}$, three parts [69], and their relationship is $G=G_{w}+G_{nb}+G_{l}$. The specific calculation method is as follows:(9)$$G=\frac{\sum_{i}^{k}\sum_{j}^{k}\sum_{r}^{n_{i}}\sum_{q}^{n_{j}}\left|y_{ir}-y_{jq}\right|}{2n^{2}\bar{y}}$$
(10)$$G_{ii}=\frac{\sum_{r=1}^{n_{i}}\sum_{q=1}^{n_{i}}\left|y_{ir}-y_{iq}\right|/2\bar{Y_{i}}}{n_{i}^{2}}$$
(11)$$G_{w}=\sum_{i=1}^{k}G_{ii}p_{i}s_{i}$$
(12)$$G_{ij}=\frac{\sum_{r}^{n_{i}}\sum_{q}^{n_{j}}\left|y_{ir}-y_{jq}\right|}{n_{i}n_{j}\left(\bar{Y_{i}}-\bar{Y_{j}}\right)}$$
(13)$$G_{nb}=\sum_{i=2}^{k}\sum_{j=1}^{i-1}G_{ij}\left(p_{i}s_{j}+p_{j}s_{i}\right)D_{ij}$$
(14)$$G_{l}=\sum_{i=2}^{k}\sum_{j=1}^{i-1}G_{ij}\left(p_{i}s_{j}+p_{j}s_{i}\right)\left(1-D_{ij}\right)$$
(15)$$D_{ij}=\frac{d_{ij}-p_{ij}}{d_{ij}+p_{ij}}$$
(16)$$d_{ij}=\int_{0}^{\infty }dF_{i}\left(y\right)\int_{0}^{y}\left(y-x\right)dF_{j}\left(x\right)$$
(17)$$d_{ij}=\int_{0}^{\infty }dF_{i}\left(y\right)\int_{0}^{y}\left(y-x\right)dF_{j}\left(x\right)$$

In the above formula, *i* and *j* represent different regions or different city types. *r* and *q* represent different prefecture-level cities. *k* represents the number of regions or city types, and $n_{i}$($n_{j}$) represents the number of prefecture-level cities in the region (city category). $y_{ir}$($y_{jq}$) is the coupling and coordination level of *r (q)* cities in the region (city category). Formulas (10) and (11) represent the Gini coefficient in region *i* (city category) and the contribution of difference in all prefecture-level city regions (city category), respectively. Formulas (14) and (15) represent the Gini coefficient between regions *i* and *j* (city category) and the contribution of regional (city category) differences in all prefecture-level cities, respectively. Formula (14) is the hypervariable density contribution. $D_{ij}$ is the relative impact of coupling and coordination between *i* and *j* regions (city category), and the calculation method is shown in Formula (15). $d_{ij}$ represents the mathematical expectation of the sum of all sample values of $y_{ir}-y_{jq}>0$ in *i* area (city category) and *j* area (city category). $p_{ij}$ is the mathematical expectation of the sum of all the sample values of $y_{jq}-y_{ir}>0$ in *i* region (city category) and *j* region (city category). The specific calculation method is shown in Formulas (16) and (17). $F_{i}$($F_{j}$) is the cumulative density distribution function of the *i (j)* region (city category).

#### 2.2.3. Spatial Correlation Measure

The first law of geography holds that everything has relevance, and the relevance of similar things is stronger. Therefore, the coupling and coordination degree of digital economy development and carbon emissions reduction in different cities may have a spatial correlation. In this paper, Moran’s I index is used to measure the overall spatial correlation of coupling and coordination. The specific calculation formula is as follows:(18)$$Moran^{\prime }s\;I=\frac{\sum_{i=1}^{n}\sum_{j=1}^{n}w_{ij}\left(D_{i}-\bar{D}\right)\left(D_{j}-\bar{D}\right)}{S^{2}\sum_{i=1}^{n}\sum_{j=1}^{n}w_{ij}}$$

In formula (18), $S^{2}=\frac{1}{n}\overset{n}{\underset{i=1}{\sum }}{\left(D-\bar{D}\right)}^{2},\;\bar{D}=\frac{1}{n}\overset{n}{\underset{i=1}{\sum }}D_{i}$*,*$D_{i}$ represents the coupling and coordination of region *i.* $w_{ij}$ is the spatial weight matrix, which is defined as 1 for adjacent regions and 0 for the opposite. Moran’s I-value range is between −1 and 1. The value is closer to 1, the more agglomeration occurs in the spatial areas with similar coupling coordination characteristics, and the value is closer to −1, the more agglomeration occurs in the spatial areas with different coupling coordination attributes. In addition, Moran’s I index is closer to 0, which means that the coupling and coordination degree belongs to a random distribution state, and the spatial autocorrelation is weak or even no spatial autocorrelation.

## 3. Spatial and Temporal Characteristics of the Coupling and Coordination between China’s Digital Economy Development and Carbon Emissions Reduction

### 3.1. Calculation of the Coupling and Coordination Degree of Two Systems

Based on Formula (8), this paper calculates the coupling and coordination of digital economy development and carbon emissions reduction in 283 prefecture-level cities in China from 2011 to 2018. In Table 3, *U_digital_* represents the comprehensive development level of the digital economy system, *U_carbon_* represents the comprehensive development level of the carbon emissions reduction system, and *U_carbon_/U_digital_* represents the ratio of carbon emissions reduction to the development of the digital economy, which is used to measure the leading or lagging degree of the carbon emissions reduction system relative to the digital economy system. If the ratio is greater than 1, it means that the carbon emissions reduction system is ahead of the digital economy system; If the ratio is less than 1, it means that the carbon emissions reduction system lags behind the digital economy system; The ratio is equal to 1, indicating that they develop simultaneously. *D_digital-carbon_* refers to the coupling and coordination between the digital economy system and the carbon emissions reduction system. The higher the value, the higher the coordination level between the two systems.

From the overall perspective of Table 3 and Figure 2, the coupling and coordination degree of China’s digital economy development and carbon emissions reduction shows a growing trend, which increased from 0.4301 in 2011 to 0.5418 in 2018, with a growth rate of 25.97% compared with 2011, indicating that China’s digital economy development system and carbon emissions reduction system have significantly promoted each other. In the sample period, the level of coupling and coordination has evolved from light coupling and coordination to medium coupling and coordination, reflecting a leap forward transformation in the mutual promotion between the digital economy system and the carbon emissions reduction system. From the ratio of the two systems, the results in the sample period are less than 1, and the ratio shows a trend of decreasing first and then increasing, which indicates that China’s carbon emissions reduction system has been lagging behind the digital economy development system for a long time in the past, but there is a positive phenomenon of carbon emissions reduction promoting digitalization in the later part of the sample period. The unbalanced development of the two reflects that the interaction between China’s digital economy and carbon emissions reduction has not achieved good results.

In order to more intuitively present the development of the coupling and coordination between the digital economy and carbon emissions reduction, this paper presents the data information in Table 3 through diagrams. See Figure 2 for details.

### 3.2. Calculation of the Coupling and Coordination Degree of Two Systems in Eight Regions

From the perspective of the coupling and coordination of digital economy development and carbon emissions reduction in eight regions, there are obvious differences between regions. The eight regions are the Northeast Comprehensive Economic Zone, Northern Coastal Comprehensive Economic Zone, Eastern Coastal Comprehensive Economic Zone, Southern Coastal Economic Zone, Middle Reaches of the Yellow River Comprehensive Economic Zone, Middle Reaches of the Yangtze River Comprehensive Economic Zone, Southwest Comprehensive Economic Zone, and Northwest Comprehensive Economic Zone. In order to present more content information, this paper shortens the names of the eight regions in Table 4. The specific results are shown in Table 4 and Figure 3. It can be seen from Table 4 that the coupling and coordination of digital economy development and carbon emissions reduction in the eastern coastal comprehensive economic zone and the southern coastal economic zone are higher than that in other regions as a whole. The coupling and coordination degree of the eastern coastal comprehensive economic zone has increased from 0.5878 in 2011 to 0.6810 in 2018, which is in a medium coupling and coordination state and shows an overall growth trend. The southern coastal economic zone has increased from 0.5814 in 2011 to 0.6725 in 2018. As a whole, the region belongs to the medium coupling and coordination type. It is worth noting that the coupling and coordination of the digital economy development and carbon emissions reduction was 0.7036 in 2017, showing a high coupling and coordination state. It can be seen that the interaction between the digital economy and carbon emissions reduction in the southern coastal economic zone has seen a significant leap. 

From the perspective of other regions, the coupling and coordination degree of the middle reaches of the Yangtze River comprehensive economic zone, and northeast comprehensive economic zone belong to the light type. The coupling and coordination degree of the northern coastal comprehensive economic zone and southwest comprehensive economic zone increased from 0.4415 and 0.3915 in 2011 to 0.5614 and 0.5152 in 2018, respectively, and the coupling and coordination type also increased from light to medium. The middle reaches of the Yellow River comprehensive economic zone increased from 0.2668 in 2011 to 0.3765 in 2018, and the coupling coordination type increased from maladjustment to light. It can be seen that the interaction effect of the digital economy and carbon emissions reduction in the above three regions has significantly improved. Compared with other regions, the coupling and coordination level of the northwest comprehensive economic zone is always lower than 0.3 and at the bottom. From the perspective of coordination type, it is in a stage of maladjustment. To sum up, it can be considered that the driving force of the eight regional digital economy development on carbon emission reduction shows obvious dynamic development characteristics, which indicates that the imbalance between the digital economy development system and the carbon emissions reduction system has significantly weakened in terms of time, but the regional differences are large in terms of space, especially the lag speed of the development of the northwest comprehensive economic zone is obvious.

On the whole, the eastern coastal comprehensive economic zone and the southern coastal economic zone have strong coordination and interaction between digital economic development and carbon emission reduction, which belong to the first echelon level at the regional level and have a typical “demonstration effect” on other regions. The coordination and interaction of digital economy development and carbon emission reduction in the northern coastal comprehensive economic zone, Yangtze River comprehensive economic zone, southwest comprehensive economic zone, northeast comprehensive economic zone and the Yellow River comprehensive economic zone belong to the coupling and coordination type of light or light to middle transformation, which is at the second echelon level in China’s regional level. The coordination and interaction between digital economic development and carbon emission reduction in the northwest comprehensive economic zone, which is relatively backward in economic development and low in infrastructure perfection, is the weakest. It belongs to the type of unbalance in the coupling and coordination and is at the third echelon level at the regional level.

Therefore, from the perspective of regional sustainable development, the central and local governments should continue to pay attention to the results of the role of digital economy development in driving carbon emissions reduction in the eastern coastal comprehensive economic zone and the southern coastal economic zone, and promote the realization of the “demonstration effect” and “learning effect” of the above two regions on other regions. In other regions, especially the northwest comprehensive economic zone, there is still much room for improvement in the good interaction between the digital economy development system and the carbon emissions reduction system. So the central government’s deployment of digital economy development should balance the needs of regional development, and the local government should deeply tap the potential of regional digital economy development so as to narrow the level of coupling and coordination of the digital economic development system and carbon emission reduction system between regions and improve the level of coupling and coordination of digital economic t system and carbon emissions reduction system within regions, thus forming a good overall interaction effect between digital economic development and carbon emissions reduction.

In order to present the above data results more clearly, this paper describes the development trend of coupling and coordination between the digital economy and carbon emissions reduction through diagrams in eight regions. See Figure 3 for details.

### 3.3. Calculation of the Coupling and Coordination Degree of Two Systems in Each Province

From the ranking of each province in Table 5, Beijing, Shanghai and Guangdong ranked top in the coupling and coordination of digital economic development and carbon emissions reduction, with the corresponding mean values of 0.7560, 0.7181, and 0.7086, respectively. The average coupling and coordination degree of the two major systems in the above provinces exceeded 0.7. The provinces at the bottom of the list are Ningxia, Qinghai, Hainan, Tibet, Shanxi, Inner Mongolia, Gansu and Jiangxi, with the corresponding mean values of 0.2291, 0.2298, 0.2441, 0.2507, 0.2660, 0.2703, 0.2758, and 0.2901 respectively. The average coupling and coordination degree of the two major systems in the above provinces is lower than 0.3. For other provinces, Jiangsu, Chongqing, Zhejiang, Tianjin and Sichuan have relatively high coupling and coordination degrees, and their ranking is high. Compared with other provinces, their coupling and coordination degree is between 0.5 and 0.7. The coupling and coordination degree of the remaining provinces is between 0.3 and 0.5. From the type of coupling coordination degree of each province, the three provinces with the highest coupling coordination degree represented by Beijing have reached the stage of highly coordinated development, while the seven provinces with the lowest coupling and coordination degree represented by Ningxia are still in the stage of unbalance. For other provinces, the rest provinces represented by Jiangsu are in the stage of medium coupling and coordination. Among them, Sichuan Province has risen from the type of light to the type of medium. Other provinces represented by Hubei Province belong to the type of light, and Hebei, Heilongjiang, Anhui, Henan, Guangxi, Guizhou and Xinjiang have all realized the transformation from unbalanced to light coupling and coordination.

In general, the coupling and coordination of digital economy development and carbon emissions reduction in most Chinese provinces are not high, and the two systems have not formed a strong mutual promotion situation. Developed provinces ranked higher in the coupling and coordination of digital economy development and carbon emissions reduction, while economically underdeveloped provinces ranked lower. The development of the digital economy is based on a high tech level, providing technical support for provincial carbon emissions reduction. The realization of carbon emissions reduction needs to be based on the improvement of resource or energy utilization efficiency. The core of this goal is technology upgrading, which provides a good external technical environment for the development of the digital economy in all provinces. The low degree of coupling and coordination between the two systems in most provinces of China means that the power of carbon emissions reduction driven by the development of the digital economy is insufficient. At the same time, the external impact of carbon emissions reduction on the development of the digital economy has not formed a good help. Therefore, the coupling and coordination of the digital economy development system and carbon emission reduction system in most provinces of China have a large space for improvement.

In addition, from the rank and type of provincial coupling and coordination degree, the provinces that rank high and belong to the type of high coupling and coordination degree are all located in the east, and the provinces that rank low and belong to the type of unbalance are mostly located in the west. Due to the unbalance of the economic development level in the eastern and western regions, there are large differences in the degree of technological innovation, infrastructure perfection, human capital and economic agglomeration that drive economic development. So the coupling and coordination of the digital economic development system and carbon emissions reduction system presents a spatial unbalance. In view of this, in order to promote the transformation of the type of unbalanced and light coupling and coordination degree to the type of medium and high coupling and coordination degree, provinces of the former type need to be closely integrated into the current economic form dominated by the digital economy, seize new development opportunities, stimulate regional technological innovation through financial subsidies, financial credit support and policy guidance, and promote the construction of new digital infrastructure under the background of carbon neutrality promotion, realize the goal of digital economy development leading regional economic change, so as to promote the coupling and coordination of digital economy development and carbon emissions reduction.

### 3.4. Calculation of the Coupling and Coordination Degree of Two Systems in Different Types of Cities

From the perspective of city classification, there are also differences in the coupling and coordination of the digital economy development system and carbon emissions reduction system in different types of cities. According to the carbon emissions per unit GDP, cities are divided into three types: resource-based cities, low-carbon transitional cities and service-oriented cities. The specific results are shown in Table 6 and Figure 4. For resource-based cities, the coupling and coordination degree of the digital economy development system and carbon emissions reduction system has not formed an ideal trend of continuous growth. Compared with 2011, the growth rate of coupling and coordination degree was −4.36% in 2018, indicating that the mutual promotion effect between the digital economy development system and carbon emissions reduction system of resource-based cities with the highest carbon emissions is weak, and the coupling and coordination degree still have huge growth potential. For low-carbon transitional cities, the degree of coupling and coordination has increased from 0.4543 in 2011 to 0.5016 in 2018, with a growth rate of 10.41%, indicating that the mutual promotion between the digital economy development system and the carbon emissions reduction system of low carbon transitional cities with medium carbon emissions has achieved positive results.

For service-oriented cities, the degree of coupling and coordination has increased from 0.3442 in 2011 to 0.6169 in 2018, with a growth rate of 79.22%, indicating that the digital economy development system and carbon emissions reduction system of service-oriented cities with the lowest carbon emissions have the best mutual promotion effect. It is worth noting that the coordinated and interactive promotion effect of digital economy development and carbon emissions reduction in service-oriented cities (0.6200) has begun to exceed that of low-carbon transitional cities (0.4477) and resource-based cities (0.3937) since 2017, indicating that the growth of the coupling and coordination degree of service-oriented cities has achieved remarkable results. To sum up, the development of the digital economy in resource-based cities has not achieved ideal results in driving carbon emissions reduction, and the growth rate of coupling and coordination has slowed down and is at the end. The coupling and coordination degree and its growth rate of low-carbon transitional cities are steadily increasing. The coupling and coordination degree of service-oriented cities show a rapidly-increasing trend, and the growth rate is at the top. The above phenomenon shows that the gap between the three types of cities is widening in China for the coupling and coordination of the digital economy and carbon emissions reduction. In particular, resource-based cities and service-oriented cities are more obvious.

Resource-based cities mainly rely on the massive input of resources to drive economic development. Such cities have a low level of technological foundation, which is difficult to play a technical support role in the development of the digital economy. Against the background that China attaches great importance to and develops the digital economy, resource-based cities lack the technical resources to promote the development of the digital economy, which leads to a weak driving role in urban carbon emission reduction. However, it also reflects that resource-based cities have the largest growth potential for the coordinated development of the digital economy and carbon emissions reduction. In view of this, resource-based cities should seize the historical opportunity of vigorously promoting the development of the digital economy in China, attract high-tech human capital through policy incentives and financial subsidies, encourage the development of cloud computing, big data, artificial intelligence and other related industries, stimulate urban technological innovation, thus promoting the overall increase of urban technology level to drive carbon emission reduction to achieve significant results.

Low-carbon transitional cities have begun to promote the transformation of economic development mode. Therefore, the development of the digital economy in low-carbon transitional cities can obtain technical support within the city, and the development of the digital economy and carbon emission reduction can promote each other. In view of this, it is necessary to continue to vigorously promote the transformation of the development mode of low-carbon transitional cities, increase the deep integration of digital and industrial development, and release the dividend of digital productivity to promote the realization of urban carbon emission reduction goals. Because of their strong, sustainable economic development and good technical foundation, service-oriented cities can provide adequate financial and technical support for the development of the digital economy. Therefore, under the strong promotion of the digital economy by the country, the digital economy development of service-oriented cities drives carbon emissions reduction more significantly, and the mutual promotion effect between urban digital economy development and carbon emission reduction increases significantly.

Based on the data information in Table 6, this paper presents the development trend of coupling and coordination of digital economy and carbon emission reduction in three types of cities. See Figure 4 for details.

## 4. Spatial Difference Analysis of the Coupling and Coordination between Digital Economy Development and Carbon Emissions Reduction

In order to further demonstrate the overall difference in the coupling and coordination of digital economy development and carbon emissions reduction, the differences within and differences among eight regions, as well as the main contribution sources, based on Matlab2017 software calculation, this section conducts a systematic analysis through the presentation of Dagum Gini coefficient. In order to present more result information, the names of the eight regions in the table are abbreviated here. The specific results are shown in Table 7.

Table 7 shows the Gini coefficient of the coupling and coordination of digital economy development and carbon emissions reduction in the whole country and eight regions. From the national perspective, the overall difference in coupling and coordination between the digital economy development and carbon emissions reduction shows an upward trend, with the Gini coefficient ranging from 0.1974 to 0.2200. Specifically, it decreased from 0.1985 in 2011 to 0.1974 in 2015 and then increased to 0.2106 in 2018. In the sample period, the overall increase rate was 6.10%, and the overall difference showed an expanding trend, indicating that the gap between the coupling and coordination of digital economy development and carbon emissions reduction in China’s cities is growing.

From the intra-group differences of the eight regions, as far as the changing trend is concerned, the Gini coefficient of the coupling and coordination between digital economic development and carbon emissions reduction in the northeast comprehensive economic zone, the northern coastal comprehensive economic zone, the eastern coastal comprehensive economic zone, the middle reaches of the Yellow River comprehensive economic zone, the southwest comprehensive economic zone and the northwest comprehensive economic zone shows an upward trend, The Gini coefficient increased from 0.1277, 0.1222, 0.0588, 0.1013, 0.1409 and 0.1389 in 2011 to 0.1426, 0.1438, 0.0706, 0.1359, 0.1603, and 0.1664 in 2018, with an increase of 11.67%, 17.68%, 20.07%, 34.16%, 13.77%, and 19.80% respectively, indicating that the gap within the region is widening. The Gini coefficient of the coupling and coordination between digital economic development and carbon emissions reduction in the southern coastal economic zone and middle reaches of the Yangtze River comprehensive economic zone shows a downward trend. The Gini coefficient decreased from 0.1418 and 0.1301 in 2011 to 0.1304 and 0.1004 in 2018, respectively, with a decrease of 8.09% and 22.83%, indicating that the gap within the region is narrowing. From the absolute value of the change, the Gini coefficient of the northeast comprehensive economic zone, northern coastal comprehensive economic zone, southwest comprehensive economic zone, and northwest comprehensive economic zone is greater than that of other regions, indicating that the unbalance of the coupling and coordination of digital economic development and carbon emissions reduction in the above regions is severe than that in other regions.

From the perspective of the differences among the eight regional groups, the regional differences between the northwest comprehensive economic zone, the middle reaches of the Yangtze River comprehensive economic zone, the northern coastal comprehensive economic zone, the middle reaches of the Yellow River comprehensive economic zone, the southwest comprehensive economic zone and the northeast comprehensive economic zone show a significant trend of expansion, with the expansion rates of 88.71%, 78.67%, 76.27%, 64.52%, and 48.00% respectively. However, the regional differences between the northern coastal comprehensive economic zone, the eastern coastal comprehensive economic zone and the southern coastal economic zone have a significant narrowing trend, with a narrowing range of 18.94% and 16.71%, respectively. At the same time, the regional differences between the middle reaches of the Yangtze River comprehensive economic zone, the southern coastal economic zone and the eastern coastal comprehensive economic zone also showed a significant downward trend, with a decline of 20.75% and 19.38%, respectively. To sum up, the overall regional difference in coupling and coordination of digital economy development and carbon emissions reduction between moderately developed regions and developed regions in China, but the gap between economically backward regions and moderately developed regions is widening.

From the contribution and source of regional differences in coupling and coordination degree of eight regions, the contribution rate of inter-group differences has always remained in the range of 69.82% to 75.91%, indicating that inter-regional differences are the main reason for the overall regional difference of coupling and coordination. The contribution rate of hypervariable density increased from 17.73% in 2011 to 17.78% in 2018, indicating that the hypervariable density contribution is the second cause affecting the overall regional difference in coupling and coordination. The contribution rate of intra-group differences is always lower than 8.5%, indicating that the contribution to the overall regional differences is small.

## 5. Spatial Correlation Analysis of the Coupling and Coordination between Digital Economy Development and Carbon Emissions Reduction in China

Regional development will not only have a direct impact on the region but also have an indirect impact on the development of surrounding areas. This “spillover” impact makes the economic activities between regions closely related in space. In view of this, it is necessary to deeply analyze whether the coupling and coordination of digital economy development and carbon emissions reduction in the target region will be affected by the changes of coupling and coordination in the surrounding regions, which will help to provide a practical reference for the policy implementation in the interaction and coordination of economic activities among regions in the future. It can be seen from Table 8 that Moran’s I index of coupling and coordination between the digital economy and carbon emissions reduction in China is greater than zero and has passed the 1% significance level test from 2011 to 2018. It can be considered that there is a positive spatial correlation in the coupling and coordination between China’s digital economy development and carbon emissions reduction in the sample period; that is, high or low values of the same type of coupling coordination will become more significant with the aggregation of spatial distribution distance. Agglomeration brings a positive “promotion effect” and deepens the spatial connection between relevant regions.

This result fully shows that the good situation of mutual promotion between the development of the digital economy and carbon emissions reduction will have a positive “spillover” impact, which will help to enhance the positive interaction between the digital economy and carbon emissions reduction in the surrounding areas in space. It is worth noting that, compared with the more positive space radiation impact of the same type of high-value aggregation, the radiation impact of low-value aggregation in the context of positive Moran’s I index will make it possible for the coordinated development of the digital economy and carbon emissions reduction in relevant regions to fall into a “low-locked” phenomenon

## 6. Conclusions 

This paper measures the coupling and coordination relationship between China’s digital economy development and carbon emission reduction by using the entropy weight method, building a coupling and coordination degree model, measuring the Dagum Gini coefficient and Moran’s I index, and analyzing the spatial-temporal features, spatial difference sources and spatial correlation of the coupling and coordination of digital economy and carbon emissions reduction in detail. Due to the limitation of data disclosure, this paper focuses on the measurement of digital economy development from 2011 to 2018. In the long term, this may produce certain restrictions on the identification of more significant features of the evaluation results and the capture of development stability. However, according to the research in this paper, China’s digital economy is developing rapidly, which makes it possible to supplement the data in the future and improve the long-term dynamic research in this paper. Through systematic research, the research conclusions are as follows:(1)The interactive promotion effect and interdependence between China’s digital economy development and carbon emissions reduction have gradually increased, which means that the coupling and coordination have shown an increasing trend. The mutual promotion effect of the two systems between regions shows the characteristics of “the east is better than the west” and “the coast is better than inland.” The coupling and coordination degree of different provinces and city types is unbalanced. The above phenomenon reflects the mutual promotion effect of China’s digital economy, and carbon emission reduction still has great potential for improvement;(2)It was found that the overall difference in the coupling and coordination between China’s digital economy development and carbon emissions reduction shows an expanding trend, which is reflected within and between regions. Especially among regions, the gap between developing regions and developed regions has widened significantly. The regional difference is the main source of the overall difference in the coupling and coordination;(3)On the whole, the spatial “spillover” effect of the interaction between China’s digital economy development and carbon emission reduction has been obtained. The improvement of the coupling and coordination degree of the two systems in various regions will have an indirect impact on the surrounding regions. The development of the coupling and coordination degree of the two systems has a significant spatial correlation.

## 7. Recommendations

In view of the above research conclusions, this paper proposes the following policy recommendations:(1)China should vigorously improve the growth potential of the coupling and coordination between digital economy development and carbon emissions reduction, accelerate the process of digital economy development and carbon emissions reduction, and form a sustainable development situation in which the digital economy and carbon emissions reduction can promote each other at a high level. The central and local governments should increase the policy support for the development of the digital economy in the western inland areas, economically underdeveloped provinces and resource-based cities, and create good external conditions such as policy preference and financial capital support to attract high-quality human capital and technologies. At the same time, the eastern coastal areas and other areas with superior financial and technological bases should form financial and technological support for the western inland areas, promote the efficient flow of key elements, and stimulate the development vitality of the digital economy in various regions. The western inland region itself should be closely integrated into a new round of economic development, seize the new opportunities of digital assistance for green development, combine the advantages of local resource elements, formulate appropriate digital economic development policies, and cultivate new drivers of the digital economy to drive the carbon emissions reduction process;(2)The government should attach importance to and improve the coordinated development mechanism of digital elements driving carbon emissions reduction in relatively backward regions represented by the northwest comprehensive economic zone, narrow the gap of the interaction effect between the digital economy and carbon emissions reduction in various regions, and promote the balanced development of the coupling and coordination of digital economy and carbon emissions reduction between relatively backward regions and developed regions. Local governments such as the northwest comprehensive economic zone should formulate a systematic plan for the development of the digital economy and make overall arrangements for the coordinated development between various elements and carbon emissions reduction. Economically underdeveloped provinces need to vigorously promote the development of new technologies such as big data, cloud computing, artificial intelligence, 5G communication, etc., deeply tap high-quality human capital and make full use of its advantages, support the improvement of various elements of the digital economy with high-quality human capital and science and technology, and improve the quality of digital development carriers. In addition, regions should rely on data flow to lead urban development, create new advantages such as digital elements to promote efficient resource allocation and external supervision and build a positive driving effect of digital economy terminal applications on carbon emissions reduction. Meanwhile, the northwest comprehensive economic zone should seize the development opportunity of carbon peaking and carbon neutrality, accelerate the sustainable development of cities enabled by technology, and boost cities to achieve digital development and change;(3)The government should coordinate the balanced development of the coupling and coordination of the digital economy and carbon emissions reduction among regions, break the barriers of spatial correlation for the digital economy and carbon emissions reduction to promote each other in various regions and strengthen the coordinated promotion of digital economy development policies and carbon emissions reduction policies among regions. Regions should give full play to the advantages of data element flow so that the digital economy and carbon emission reduction in each region can form a high level of mutual promotion of sustainability. Within the region, it should encourage policy linkage among regions, accelerate the formation of a unified digital economy development market, promote the emergence of digital economies of scale, and then expand and strengthen the digital industry in the region to help achieve the carbon emissions reduction goal.

## Figures and Tables

**Figure 1 ijerph-20-00872-f001:**
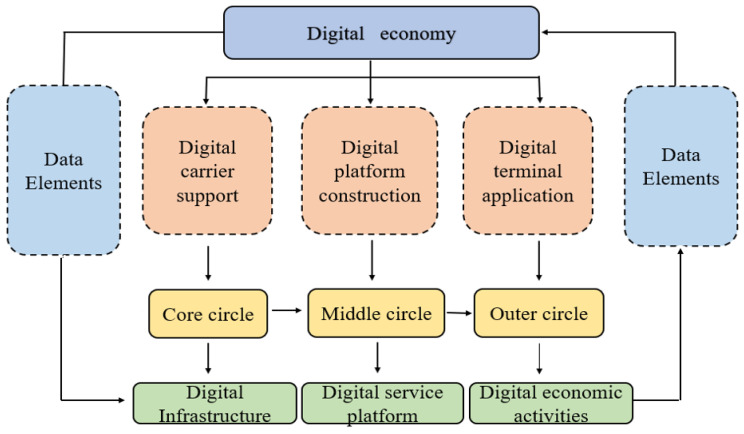
Logic framework of digital economy measurement. Source: Figure 1 made by the author.

**Figure 2 ijerph-20-00872-f002:**
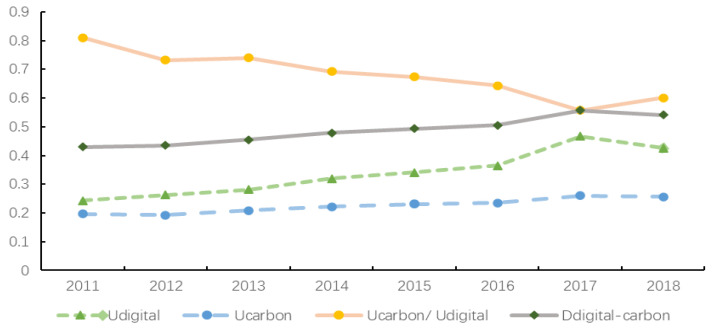
Status and trend of the coupling and coordination between digital economy development and carbon emissions reduction in China.

**Figure 3 ijerph-20-00872-f003:**
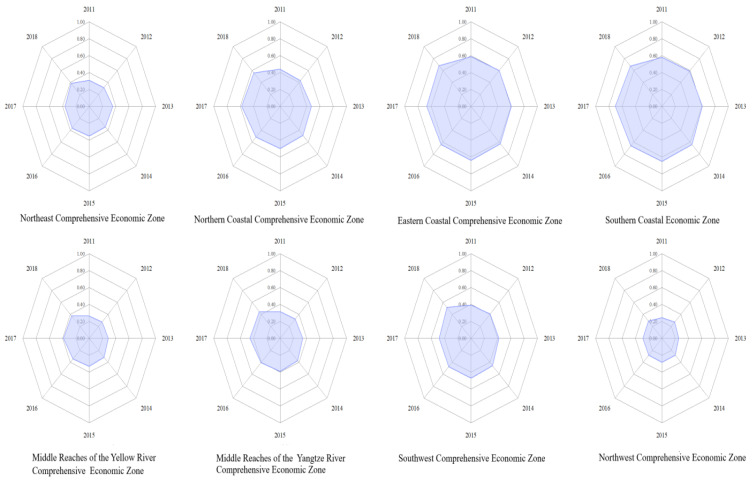
Status and characteristics of coupling and coordination of digital economy development and carbon emissions reduction in eight regions.

**Figure 4 ijerph-20-00872-f004:**
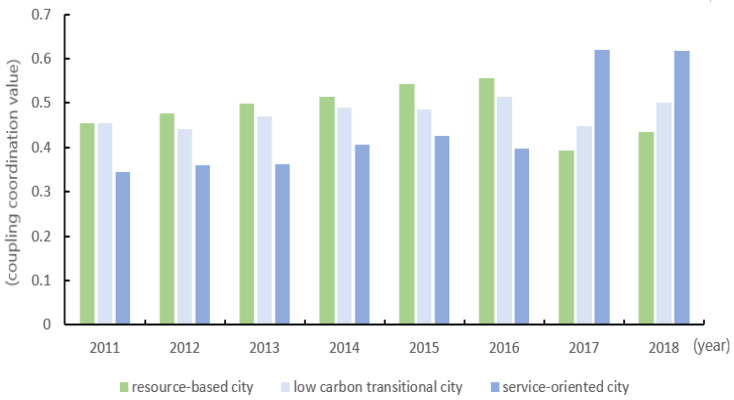
Status and trend of coupling and coordination of digital economy development and carbon emissions reduction in three types of cities.

**Table 1 ijerph-20-00872-t001:** Indicator system of digital economy system and carbon emissions reduction system.

	System	Subsystem	Indicators
coupling and coordination system of the digital economy development and carbon emissions reduction	digital economy development system	elements of digital infrastructure	(a1) the government’s attention to the development of the digital economy: the frequency of digital economy-related words in the government work report
			(a2) emphasis on human capital: education expenditure/total financial expenditure
			(a3) regional digitization degree: digitization degree index
			(a4) application level of digital infrastructure: number of enterprises engaged in e-commerce transactions
			(a5) application level of digital infrastructure: number of enterprises with digital economy characteristics
		elements of digital services and platform economy	(a6) computer software practitioners
			(a7) software business income
			(a8) e-commerce sales revenue
			(a9) number of domain names
			(a10) number of websites
		elements of digital economic activities	(a11) telecom business income
			(a12) number of mobile phone users
			(a13) number of Internet users
			(a14) express delivery income
			(a15) total digital inclusive finance index
			(b1) capital factor: proportion of science and technology expenditure in financial expenditure
			(b2) technical factor: number of patent applications
	carbon emissions reduction system	input factors	(b3) energy factor: renewable energy power generation
			(b4) social factor: green area
		output factors	(b5) carbon productivity: GDP per unit carbon emissions

Note: (a1) data and (a5) data in elements of digital infrastructure are obtained through python and manual collection, respectively. (b5) Data in the carbon emissions reduction system is obtained through manual collection.

**Table 2 ijerph-20-00872-t002:** Types and division criteria of coupling coordination degree.

Coupling and Coordination Scope	(0, 0.3]	(0.3, 0.5]	(0.5, 0.7]	(0.7, 0.9]	(0.9, 1.0]
Coupling and coordination type	unbalance	light coupling and coordination	medium coupling and coordination	high coupling and coordination	high-quality coupling and coordination

**Table 3 ijerph-20-00872-t003:** Characteristics of the coupling and coordination between digital economy development and carbon emissions reduction in China.

Year	U_digital_	U_carbon_	U_carbon_/U_digital_	D_digital-carbon_	Type of Coupling and Coordination
2011	0.2433	0.1972	0.8105	0.4301	light
2012	0.2625	0.1923	0.7325	0.4353	light
2013	0.2817	0.2086	0.7405	0.4558	light
2014	0.3196	0.2213	0.6924	0.4789	light
2015	0.3412	0.2301	0.6744	0.4931	light
2016	0.3651	0.2349	0.6434	0.5054	medium
2017	0.4673	0.2602	0.5568	0.5575	medium
2018	0.4261	0.2563	0.6015	0.5418	medium

Table 3 calculated by the authors.

**Table 4 ijerph-20-00872-t004:** Status and types of coupling and coordination between digital economy and carbon emissions reduction in eight regions.

Year	2011	2012	2013	2014	2015	2016	2017	2018
Northeast Economic Zone	0.3087	0.3147	0.3541	0.3458	0.3500	0.3638	0.3655	0.3837
Type	light	light	light	light	light	light	light	light
Northern Coastal Economic Zone	0.4415	0.4334	0.4725	0.4872	0.4976	0.5111	0.5776	0.5614
Type	light	light	light	light	light	medium	medium	medium
Eastern Coastal Economic Zone	0.5878	0.5992	0.6085	0.6247	0.6370	0.6384	0.6652	0.6810
Type	medium	medium	medium	medium	medium	medium	medium	medium
Southern Coastal Economic Zone	0.5814	0.5884	0.6076	0.6391	0.6518	0.6628	0.7036	0.6725
Type	medium	medium	medium	medium	medium	medium	high	medium
Yellow River Economic Zone	0.2668	0.2730	0.2867	0.3139	0.3295	0.3454	0.3895	0.3765
Type	unbalance	unbalance	unbalance	light	light	light	light	light
Yangtze River Economic Zone	0.3161	0.3226	0.3424	0.3721	0.3901	0.4034	0.4565	0.4431
Type	light	light	light	light	light	light	light	light
Southwest Economic Zone	0.3915	0.4108	0.4203	0.4586	0.4711	0.4750	0.4775	0.5152
Type	light	light	light	light	light	light	light	medium
Northwest Economic Zone	0.2462	0.2727	0.2541	0.2826	0.2831	0.2831	0.2876	0.2892
Type	unbalance	unbalance	unbalance	unbalance	unbalance	unbalance	unbalance	unbalance

Table 4 calculated by the authors.

**Table 5 ijerph-20-00872-t005:** Status and types of coupling and coordination between digital economy and carbon emissions reduction in various provinces.

Province	2011	2012	2013	2014	2015	2016	2017	2018	Mean	Ranking	Type
Beijing	0.7504	0.7246	0.7369	0.7602	0.7602	0.7652	0.7630	0.7876	0.7560	1	high
Tianjin	0.5021	0.4885	0.5067	0.5490	0.5915	0.5976	0.6082	0.6564	0.5625	7	medium
Hebei	0.2993	0.2781	0.2975	0.3042	0.3159	0.3289	0.3501	0.3630	0.3171	20	light
Shanxi	0.2281	0.2106	0.2430	0.2599	0.2432	0.2897	0.2936	0.2799	0.2560	27	unbalance
Inner Mongolia	0.2381	0.2330	0.2677	0.2683	0.2794	0.2873	0.2983	0.2899	0.2703	26	unbalance
Liaoning	0.3560	0.3686	0.4051	0.4005	0.3928	0.4160	0.4023	0.4320	0.3967	12	light
Jilin	0.2376	0.2459	0.2613	0.2700	0.3033	0.3190	0.3448	0.3590	0.2926	23	unbalance
Heilongjiang	0.2804	0.2725	0.3376	0.3106	0.3126	0.3304	0.3183	0.3297	0.3115	21	light
Shanghai	0.6778	0.6621	0.6934	0.7250	0.7429	0.7419	0.7370	0.7643	0.7181	2	high
Jiangsu	0.5804	0.5915	0.6056	0.6116	0.6171	0.6151	0.6297	0.6635	0.6143	4	medium
Zhejiang	0.5448	0.5750	0.5642	0.5879	0.6085	0.6155	0.6522	0.6621	0.6013	6	medium
Anhui	0.2795	0.2850	0.3161	0.3400	0.3730	0.4030	0.4222	0.4372	0.3570	16	light
Fujian	0.3831	0.3944	0.4080	0.4202	0.4512	0.4707	0.4978	0.5050	0.4413	11	light
Jiangxi	0.2179	0.2342	0.2509	0.2804	0.3123	0.3035	0.3471	0.3741	0.2901	24	unbalance
Shandong	0.3931	0.3985	0.4600	0.4637	0.4644	0.4801	0.5109	0.5398	0.4638	10	light
Henan	0.2769	0.2930	0.3098	0.3441	0.3718	0.3776	0.4035	0.4147	0.3489	17	light
Hubei	0.4129	0.4157	0.4247	0.4665	0.4720	0.4872	0.5599	0.5252	0.4705	9	light
Hunan	0.3118	0.3139	0.3357	0.3576	0.3682	0.3799	0.4054	0.4050	0.3596	15	light
Guangdong	0.6508	0.6588	0.6798	0.7191	0.7258	0.7352	0.7596	0.7400	0.7086	3	high
Guangxi	0.2765	0.2882	0.2842	0.3109	0.3248	0.3018	0.3240	0.3281	0.3048	22	light
Hainan	0.2206	0.2237	0.2161	0.2439	0.2612	0.2365	0.2675	0.2833	0.2441	29	unbalance
Chongqing	0.5188	0.5440	0.5774	0.6163	0.6336	0.6340	0.6565	0.6790	0.6075	5	medium
Sichuan	0.4336	0.4433	0.4709	0.5223	0.5276	0.5475	0.5951	0.5828	0.5154	8	medium
Guizhou	0.2641	0.3117	0.2814	0.3072	0.3540	0.3528	0.4009	0.3989	0.3334	18	light
Yunnan	0.3549	0.3882	0.3667	0.4000	0.3925	0.3971	0.4280	0.4196	0.3934	13	light
Tibet	0.2422	0.3248	0.1114	0.3392	0.3224	0.3428	0.1652	0.1574	0.2507	28	unbalance
Shaanxi	0.3121	0.3311	0.2954	0.3405	0.3557	0.3749	0.4841	0.4253	0.3649	14	light
Gansu	0.2743	0.2811	0.2779	0.2876	0.2695	0.2673	0.2699	0.2786	0.2758	25	unbalance
Qinghai	0.2208	0.2450	0.2266	0.2147	0.2406	0.2121	0.2322	0.2466	0.2298	30	unbalance
Ningxia	0.1743	0.2485	0.2025	0.2374	0.2377	0.2327	0.2431	0.2564	0.2291	31	unbalance
Xinjiang	0.2588	0.2796	0.2765	0.3337	0.3657	0.3962	0.3650	0.3922	0.3334	19	light

Table 5 calculated by the authors.

**Table 6 ijerph-20-00872-t006:** Status and types of coupling and coordination between digital economy and carbon emissions reduction in different types of cities.

Year	2011	2012	2013	2014	2015	2016	2017	2018	Growth Rate (%)
Resource-based cities	0.4541	0.4754	0.4981	0.5138	0.5426	0.5566	0.3937	0.4343	−4.36
Low-carbon transitional cities	0.4543	0.4404	0.4693	0.4904	0.4858	0.5150	0.4477	0.5016	10.41
Service-oriented cities	0.3442	0.3587	0.3618	0.4052	0.4254	0.3961	0.6200	0.6169	79.22

Table 6 calculated by the authors.

**Table 7 ijerph-20-00872-t007:** Analysis of national and regional differences in the coupling and coordination of digital economy development and carbon emissions reduction.

Year		2011	2012	2013	2014	2015	2016	2017	2018
Gaps inside region	Overall	0.1985	0.2026	0.2029	0.2000	0.1974	0.1992	0.2200	0.2106
	the northeast economic zone	0.1277	0.1439	0.1468	0.1386	0.1542	0.1268	0.1646	0.1426
	the northern economic zone	0.1222	0.1491	0.1547	0.1520	0.1507	0.1482	0.1383	0.1438
	the eastern economic zone	0.0588	0.0516	0.0637	0.0693	0.0711	0.0732	0.0787	0.0706
	the southern economic zone	0.1418	0.1384	0.1464	0.1536	0.1388	0.1418	0.1353	0.1304
	middle reaches of the Yellow River zone	0.1013	0.1206	0.0987	0.1109	0.1433	0.1319	0.1276	0.1359
	middle reaches of the Yangtze River zone	0.1301	0.1279	0.1175	0.1115	0.0956	0.1056	0.1121	0.1004
	the southwest economic zone	0.1409	0.1322	0.1571	0.1448	0.1304	0.1537	0.1609	0.1603
	the northwest economic zone	0.1389	0.0970	0.1472	0.1445	0.1534	0.1602	0.1528	0.1664
Gaps between regions	the northeast zone and the northwest zone	0.1398	0.1314	0.1697	0.1456	0.1747	0.2080	0.1793	0.2069
	the northern zone and the eastern zone	0.2144	0.2391	0.1926	0.1949	0.1995	0.1870	0.1755	0.1738
	the northern zone and the southern zone	0.2029	0.2121	0.1963	0.2127	0.2025	0.2030	0.1823	0.1690
	the northern zone and the northwest zone	0.2002	0.1690	0.1697	0.2301	0.2499	0.2862	0.3480	0.3577
	the eastern zone and Yangtze River zone	0.3163	0.1625	0.2990	0.2772	0.2659	0.2606	0.2583	0.2550
	the southern zone and Yangtze River zone	0.2858	0.2779	0.2737	0.2734	0.2542	0.2622	0.2471	0.2265
	Yellow River zone and the northwest zone	0.1252	0.1161	0.1293	0.1357	0.1610	0.1862	0.2024	0.2207
	Yangtze River zone and the northwest zone	0.1444	0.1204	0.1566	0.1540	0.1775	0.2090	0.2611	0.2725
	the southwest zone and the northwest zone	0.1807	0.1598	0.2031	0.2047	0.2280	0.2470	0.2942	0.2973
Contribution rate	Within the region	7.9200	8.0105	8.2580	8.3493	8.3617	8.4053	7.8096	7.9497
	Between regions	74.3517	73.3009	71.7543	71.9802	71.7502	69.8273	75.9021	74.2676
	Hypervariable density	17.7283	18.6886	19.9877	19.6705	19.8881	21.7674	16.2883	17.7827

Table 7 calculated by the authors. Table 7 only presents the core content. If readers need information about the gaps between other regions, please ask the author for it.

**Table 8 ijerph-20-00872-t008:** Spatial correlation of the coupling and coordination between digital economy development and carbon emissions reduction.

Year	2011	2012	2013	2014	2015	2016	2017	2018
Morans’I	0.148 *** (28.003)	0.149 *** (28.003)	0.139 *** (26.399)	0.142 *** (26.914)	0.140 *** (26.677)	0.131 *** (24.907)	0.143 *** (27.156)	0.138 *** (26.116)
Type	positive autocorrelation	positive autocorrelation	positive autocorrelation	positive autocorrelation	positive autocorrelation	positive autocorrelation	positive autocorrelation	positive autocorrelation

Table 8 calculated by the authors. *** is significant at the level of 1%.

## Data Availability

Not applicable.

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
