# Peer review of "Analysis of the Coupling Effect and Space-Time Difference between China’s Digital Economy Development and Carbon Emissions Reduction"

_ijerph, 2023, doi:10.3390/ijerph20010872_

Round 1
Reviewer 1 Report
The manuscript "Measurement and Evaluation of the Coupling and Coordination of China's Digital Economy Development and Carbon Emissions Reduction" by Nan Li, Beibei Shi, Rong Kang was submitted for peer review.
I read the submitted manuscript with great interest. The author turned to a very urgent problem: Carbon Emissions Reduction.
A great deal of research has been done by the authors. But the manuscript has several significant flaws that need to be corrected. Correction of the shortcomings listed below must be done to improve the quality of the manuscript, enhance the ease of perception of the presented material and increase the interest of a readers.
1.) From my point of view the title of the manuscript does not reflect the essence of the research. The authors are solving rather urgent task: Carbon Emissions Reduction. But the title of the manuscript has two unrelated parts. One part is 'digital technology' and the other part is 'Carbon Emissions Reduction'. From my point of view, a more succinct and concrete title is needed. This comment is advisory in nature.
2.) From my point of view, there are very few keywords. Keywords enable the reader to quickly search for the necessary material and enable the author to popularise their research and increase interest and citations. But if this number of keywords satisfies the requirement of the journal, this comment is advisory.
3.) The abstract is not formed correctly. It is very blurry and framed incorrectly. Abstract is a short and concise presentation of a complex study. It seems that the authors have taken certain phrases from the text and thus formed the abstract. The abstract should clearly indicate the purpose of the study, its importance for society (i.e. to characterize the problem), identify the methods and materials of the study, and the conclusions should be clearly and briefly formulated. There is no "starting point" in the abstract, that is, information about previous studies (one sentence is enough). From my point of view, in the abstract, such information begins with the statement: "Previously conducted studies have established that ...".
3.1) It is desirable to avoid narrative text in the abstract.
3.2) Try to use words and phrases: an analysis has been carried out; studied; developed; proposed; established and so on. It is advisable to start sentences in the abstract with these words and phrases.
3.3) At the end of the abstract, it is necessary to indicate the final result obtained by the authors, for example: A model has been developed that allows ...; A dependence has been established which is...; A pattern has been revealed...; An efficient system (technology) has been proposed, and so on.
The abstract should be revised.
4.) The manuscript has a sufficient reference list (43 references in total). At the same time there is no comprehensive coverage of studies in terms of geography of citations. There are not enough references to international studies in the field. There is reference to the work of foreign researchers, which dates to 1997. The list of references is intended to demonstrate the depth of the author's study of the material, the relevance and interest of their research.
4.1.) The depth of study is demonstrated with the number of references - is not enough.
4.2.) Relevance – with the availability of research in recent years – is enough.
4.3.) Interest – with the availability of research by scientists from different countries - is not enough (practically absent).
Since you are publishing your manuscript in an international publication, it is necessary to demonstrate the international relevance and interest of this issue. This can be done by analyzing the studies of scientists from different countries. It is imperative to supplement the list of references with studies of scientists from different countries over the past 3-5 years to show geographical (general/global) interest and relevance.
The List of References needs to be revised.
5.) It is necessary to avoid group references, for example: [17-20]. From my point of view, allowed up to three; more than three references are not acceptable and must be deciphered. Each paper you refer is unique and the studies you refer deserve more proper and careful review to demonstrate (and prove) its importance for the current research. It is necessary to demonstrate in detail the essence of each study and their need for your work.
6.) From my point of view, the authors abuse the names of scientists when mentioning the study. A reference [1] is sufficient. If the reader is interested in the name of the researcher, then it is easy to refer to the references list. It is important for the reader to know the essence (main idea) of the disclosed issue, not the name of the researcher.
7.) From my point of view, at the end of the introduction the authors did not quite correctly formulate a brief conclusion of the analytical study of previously performed works. The authors did not summarize their analysis and did not identify unresolved issues. This conclusion should make it possible to characterize the actual question posed, the purpose of the study and the tasks to be solved to achieve this goal. For example: Analyzing the above, it can be noted that ... is a very topical issue. Therefore, the purpose of this study is ... and to achieve this, it is necessary to solve the following tasks: 1); 2); ... Such a conclusion allows the reader to understand the vector of the study, and the authors to correctly formulate the conclusions. It needs to be improved.
8.) When analyzing previous studies, the authors make a number of unforced mistakes or make statements that are not supported by evidence (references). Some statements are very broad and difficult to understand. From my point of view, it is necessary to form more compact sentences, this way you avoid group references.
8.1.) From my point of view, lines 46-60 are redundant in the manuscript. The regulations that are not relevant to the study are listed.
8.2.) Lines 61-78. The authors try to draw parallels between the digital economy and carbon emissions reduction. But this parallel is not obvious to me. In addition, in these lines, the authors make a lot of statements unsupported by evidence (references). This shortcoming needs to be corrected.
8.3.) Lines 83-113. The authors make a lot of statements unsupported by evidence (references). This shortcoming needs to be corrected.
8.4.) Lines 83-113. The authors make a series of repetitive statements, stating the same idea in different words.
All of these things overload the manuscript and make it difficult for the reader to understand the material.
9.) Considering the comments (3), (4) and (8), I would like to note that the authors have very poorly disclosed the main subject of the study. In recent years, a lot of work has been carried out to study the role of digitalization in the economy or its individual areas. Since almost any production is accompanied, which has a significant impact on the environment, the issues of Carbon Emissions Reduction are very relevant and scientists around the world are trying to minimize it.
For example,
9.1.) Tcvetkov, P. Engagement of resource-based economies in the fight against rising carbon emissions. Energy Reports 2022, 8, 874-883. https://doi.org/10.1016/j.egyr.2022.05.259
9.2.) Tcvetkov, P. Climate policy imbalance in the energy sector: Time to focus on the value of CO2 utilization. Energies 2021, 14(2), 411. https://doi.org/10.3390/en14020411
The use of digital technologies to analyze the current situation, formulate a problem and find the right solution is carried out in all areas of the economy.
For example,
9.3) Khayrutdinov, M.M.; Golik, V.I.; Aleksakhin, A.V.; Trushina, E.V.; Lazareva, N.V.; Aleksakhina, Y.V. Proposal of an Algorithm for Choice of a Development System for Operational and Environmental Safety in Mining. Resources 2022, 11, 88. https://doi.org/10.3390/resources11100088
9.4) Kongar-Syuryun Ch., Ubysz A., Faradzhov V. Models and algorithms of choice of development technology of deposits when selecting the composition of the backfilling mixture. IOP Conf. Ser.: Earth Environ. Sci. 2021, 684(1), 012008. https://doi.org/10.1088/1755-1315/684/1/012008
As follows from the presented works (9.1) - (9.4) the authors of the manuscript submitted for review missed a large layer of research related to the impact of mining on the environment. If the authors become familiar with the works presented in (9.1), (9.2), (9.3), (9.4) they will be able to properly form the introduction, enrich their manuscript with international research by scientists from Poland, Czech Republic, Slovenia, Slovakia, Russia, Germany and demonstrate the depth of their material, as well as eliminate the remark (3).
10.) It is necessary to indicate who made figure 1. If this is the author's merit, then it is necessary to indicate: done by the authors; if this is a borrowed drawing, then it is necessary to indicate the source.
11.) Conclusion section is formatted incorrectly. Conclusion – brief summary of the study without repeating the wording given earlier in the manuscript. The authors abuse repetition throughout the manuscript, and Conclusion section is no exception. Such a presentation of the material reduces the ease of perception by the reader of the information presented. Some of the information provided by the authors in Conclusion section has already been reported in Materials and Methods section or is related to Results and Discussion section. This information should be placed in the relevant sections. This information is superfluous for the Conclusion. The mistake of incorrectly forming conclusions is a consequence of the incorrect presentation of the introduction noted by me in remark (7) due to the fact that when writing the introduction, the aims and objectives are not formulated.
Conclusions should briefly characterize the result of the study, for example:
As a result of the study
(1) the dependence of … was obtained.
(2) it was found that ...
(3) and so on.
The conclusion needs to be revised.
Summary: The manuscript is a finished research work. But the corrections are needed. The chosen research topic is relevant. From my point of view, the authors failed to present their research correctly and clearly, which reduced its value and worsened the ease of perception of the material presented.
From my point of view, the manuscript cannot be published in the open press without correction in accordance with my suggestions.
Author Response
Analysis on the Coupling Effect and Space-Time Difference between China's Digital Economy Development and Carbon Emissions Reduction
Response to Reviewer 1 Comments
We are grateful for the reviewer’s thoughtful comments and suggestions. We have thoroughly revised our manuscript according to the comments. Our responses (in blue) and the reviewer's original comments (in black) are given below:
(The specific modification content can also be seen in the Word version uploaded by the author)
Point 1:
English language and style:I don't feel qualified to judge about the English language and style
Response 1:
Thanks for your comments. Thank you for taking your precious time to review our paper. All the authors have made two language modifications when submitting the first draft. Because we have improved this article based on expert advice, we sent the revised version to professional language experts to improve the language quality of this article. We sincerely hope that we have not caused any inconvenience to your review. Thank you very much!
Point 2:
Does the introduction provide sufficient background and include all relevant references?——Must be improved.
Response 2:
Thank you very much for your comments. According to your comments, we checked the introduction and referred to the articles published in International Journal of Environmental Research and Public Health. Indeed, as you said, it is necessary to modify the introduction. We mainly made the following modifications:
First, we revised the research background. The policy background with low relevance to the theme of this paper is deleted, and the background content of digital economy and carbon emissions is re-presented, making the research background more consistent with the theme of this paper.
Second, in the introduction, we re-elaborated the connotation of coordinated development of digital economy and carbon emission reduction, so that readers can better understand the content of the coupling and coordination degree of the two systems. At the same time, it is also helpful for readers to understand the content of this study.
Third, we have made a lot of supplements to the literature in the field of digital economy and carbon emissions, presenting as much international research as possible, especially research from different countries. On the one hand, it reflects the comprehensiveness of the literature review of this paper, on the other hand, it also makes this paper more in line with the requirements of international journals. In addition, while supplementing references, we also added relevant documents to the arguments in the introduction to better support the establishment of the arguments, making the views in the article more persuasive.
Finally, in the introduction part, we clarified the purpose and task of this study, and corresponded it with the research contents and conclusions of this paper. In the introduction, the purpose and task of the study are presented, which will help readers to better understand the research content of the article and improve the logic of this study.
The specific modifications are as follows:
Line 42-54: Digital economy is an economic system with technology as its core driving force, and digitalization of economic activities is a typical feature of digital economic system. the new development mode led by technological innovation driven digital economy development has laid the foundation for the coordinated promotion of digital economy and green development. The digital economy relies on technological innovation to continuously promote industrial integration [1] and economic restructuring, while the transformation of industrial and economic development mode will directly cause changes in carbon dioxide emissions [2]. At the same time, under background of " emission peak" and "carbon neutrality", the digital regulation on which the carbon emissions reduction process is based will also have a certain impact on the development of the digital economy. Therefore, it is urgent to systematically answer the interactive development relationship between digital economy and carbon emissions reduction, so that the development of digital economy can support the advancement of carbon emissions reduction.
Line 71-83: The analysis of the interaction between digital economy and carbon emissions reduction is the analysis of the coordinated development between digital economy and carbon emissions reduction. Coordinated development includes two explanations. On the one hand, from the perspective of coordination, it emphasizes the positive interaction between digital economy development and carbon emission reduction and the formation of a virtuous circle situation that promotes each other. Coordination enables the two systems to work together effectively and sustainably [4,5]. On the other hand, from the perspective of development, it emphasizes the promotion of digital economy and carbon emissions reduction. The coordinated development between digital economy and carbon emissions reduction requires that digital economy elements can help promote the process of carbon emissions reduction through technology, which in turn can continue to provide technical and environmental support for digital development. This reflection of the connection between the two systems is coupling [6].
Line 93-97: There is a long way to go to promote the carbon emissions reduction process [7]. Therefore, promoting the research on the status and trend of coordinated development of digital economy and carbon emissions reduction is of practical significance for im-proving the layout of digital economy elements and promoting green development through digitalization.
Line 100-108: It is believed that the real digital economy refers to the part of economic output completely determined by digital technology, with a business framework based on digital services and goods [8].. With the integration of communication and computing technologies in the network and the flow of technology and data, it is strengthening the transformation of e-commerce and large-scale business [9]. Computer aided data flow has enabled digitization to transform various parts of the current economy [10]. Therefore, the digital economy is also divided into four different parts: digital services and goods; Mixed digital services and goods; IT based intensive production services; And IT industry [11].
Line 114-124: Most of the existing literatures measure the digital economy based on the method of summarizing the digital industry output, digital product classification, and digital related comprehensive indicators. Scholar first discussed the boundary problem of digital economy, which provides a useful reference for measuring digital economy [16]. The digital economy is also divided into five parts for measurement, namely infrastructure, e-commerce, industry structure, digital labor and digital price [17]. Some organizations have proposed an accounting framework with intelligent infrastructure investment, social promotion, innovation release, growth and employment as the main body to measure the digital economy [18]. In addition, there are also scholars who calculate the digital economy by building a comprehensive indicator system [19-21].
Line 131-138: A study took a telecommunications company in Slovenia as the research object, calculated the greenhouse gas emissions within the enterprise scope 3, and took the purchase of electricity, the commuting of employees, the use of vehicles owned by the company and heating as the measured objects [24]. In addition, the research calculates the grid emission factors, deduces the emission factors of specific European countries, and calculates the carbon intensity based on the power generation of power plants [25]. The other is the exploration of carbon emissions reduction path. It is generally believed that economic development is closely related to carbon emissions [26].
Line 140: The exploration of carbon emissions reduction path mainly involves the
improvement of carbon emissions trading market [27,28], the analysis of the policy effect of carbon emissions reduction pilot [29,30] and the research of other paths [31].
Line 145-152: Meanwhile, there is also study carried out from the perspective of energy, believing that the development of renewable energy is of great significance for carbon emissions reduction in resource-based countries [34]. Another study assessed the existing climate policy portfolio and believed that the energy technology portfolio needs to be adjusted [35]. Some scholars discussed the environmental impact of mineral mining based on the mining industry [36,37]. The adjustment of savings also contributes to the development of green economy [38].
Line 155-166: One is the impact of digital economy development on carbon emissions. Some scholars believe that the digital economy can inhibit carbon emissions [40-42]. A study have proved the role of Taiwan's digital development in the circular economy. It is found that through digital waste management, the consumption of raw materials can be reduced by 25% by 2030, and half of greenhouse gas emissions can be avoided [43]. There is study believed that ICT can contribute to the construction of low-carbon cities [44], mainly through governance capacity of the government [45] and energy systems [46]. Study have also targeted African countries and found an inverted "U" relationship between ICT and carbon emissions in the sample countries [47]. In addition, some scholars believe that R&D investment played a positive role in regulating the relationship between digital economy development and carbon emissions reduction [48-50].
Line 170-171: Different from this conclusion, other scholars believe that the development of digital economy is unfavorable to carbon emission reduction [53,54].
Line 178-185: By sorting out relevant domestic and foreign studies, this paper finds that most of the existing literatures focus on the unilateral relationship between digital economy development and carbon emissions based on the measurement of digital economy and carbon emissions. At present, there is still a gap in the discussion of the interaction effect of digital economy and carbon emissions reduction. It can be noted that the relationship between digital economy and carbon emissions reduction is a very topical issue. Therefore, the purpose of this study is to identify the interaction and coordination effect of digital economy and carbon emissions reduction.
Line 190-199: Therefore, it is necessary to study the coupling and coordination relationship between digital economy development and carbon emissions reduction. And to achieve this, it is necessary to solve the following tasks: (1) It is necessary to measure and comprehensively grasp the status and space-time characteristics of the coupling and coordination between China's digital economy development system and carbon emissions reduction system; (2) The source of the difference for the coupling and coordination between China's digital economy and carbon emissions reduction needs to be explored; (3) The spatial "spillover" effect of the coupling and coordination of digital economy development and carbon emissions reduction among cities is tested.
Point 3:
Are the conclusions supported by the results?——Must be improved
Response 3:
Thank you for your patient review and suggestions. All the authors checked the correspondence between the conclusion and the research results, and found that the conclusion part can also present the research content more concisely, so as to avoid duplication of expression with the research content part and improve the reading interest of readers. In addition, in order to make the whole article more enlightening, we also add the limitations of this study and the areas that can be broken through in the future in the conclusion part. Therefore, according to your suggestions and in combination with the research content of the article, we restated the conclusion. The specific modifications are as follows:
Line 840-866: Due to the limitation of data disclosure, this paper focuses on
the measurement of digital economy development from 2011 to 2018. In the long-term, this may produce certain restrictions on the identification of more significant features of the evaluation results and the capture of development stability. However, according to the research in this paper, China's digital economy is developing rapidly, which makes it possible to supplement the data in the future and improve the long-term dynamic research in this paper. Through systematic research, the research conclusions are as follows:
- The interactive promotion effect and interdependence between China's digital
economy development and carbon emissions reduction have gradually increased, which means that the coupling and coordinated has shown an increasing trend. The mutual promotion effect of the two systems between regions shows the characteristics of "the east is better than the west" and "the coast is better than the inland". The coupling and coordination degree of different provinces and city types is unbalanced. The above phenomenon reflects the mutual promotion effect of China's digital economy and car-bon emissions reduction still have great potential for improvement.
- It was found that the overall difference of the coupling and coordination be-
tween China's digital economy development and carbon emissions reduction shows an expanding trend, which is reflected within and between regions. Especially among regions, the gap between developing regions and developed regions has widened significantly. And the regional difference is the main source of the overall difference of the coupling and coordination.
- On the whole, the spatial "spillover" effect of the interaction between China's
digital economy development and carbon emission reduction has been obtained. The improvement of the coupling and coordination degree of the two systems in various regions will have an indirect impact on the surrounding regions. The development of the coupling and coordination degree of the two systems has a significant spatial correlation.
Point 4:
Comments and Suggestions for Authors:The manuscript "Measurement and Evaluation of the Coupling and Coordination of China's Digital Economy Development and Carbon Emissions Reduction" by Nan Li, Beibei Shi, Rong Kang was submitted for peer review. I read the submitted manuscript with great interest. The author turned to a very urgent problem: Carbon Emissions Reduction. A great deal of research has been done by the authors. But the manuscript has several significant flaws that need to be corrected. Correction of the shortcomings listed below must be done to improve the quality of the manuscript, enhance the ease of perception of the presented material and increase the interest of a readers.
Response 4:
Thank you very much for your affirmation of our article. It is your patient review that has made it possible to improve the quality of our article, which has greatly helped to improve our research and inspire future paper writing. All authors will carefully read and try their best to correct the shortcomings you proposed in this article to improve the quality of the manuscript. Thanks again for your help and support!
Point 5:
From my point of view the title of the manuscript does not reflect the essence of the research. The authors are solving rather urgent task: Carbon Emissions Reduction. But the title of the manuscript has two unrelated parts. One part is 'digital technology' and the other part is 'Carbon Emissions Reduction'. From my point of view, a more succinct and concrete title is needed. This comment is advisory in nature.
Response 5:
Thank you for your suggestion. The content of this paper focuses on the measurement and analysis of the degree of interaction and coordination between the digital economy development system and the carbon emissions reduction system, and "coupling and coordination" is just to enable the topic to highly summarize the research content. In combination with your suggestions, we have revised the title. The new title gives a brief summary of the specific content of this paper as much as possible, so that readers can directly capture the structure and key information of the paper. The specific modifications are as follows:
Line 1-4: Analysis on the Coupling Effect and Space-Time Difference between China's Digital Economy Development and Carbon Emissions Reduction.
Point 6:
From my point of view, there are very few keywords. Keywords enable the reader to quickly search for the necessary material and enable the author to popularise their research and increase interest and citations. But if this number of keywords satisfies the requirement of the journal, this comment is advisory.
Response 6:
Thank you very much for your advice. We checked the requirements of International Journal of Environmental Research and Public Health, and the key words can be expanded within 10. Therefore, based on your suggestion, we have added 3 keywords from the content, and the number of keywords in this article is 8. The setting of more keywords helps readers to quickly search for necessary materials, thus attracting readers. Thank you very much for your suggestions. The specific modifications are as follows:
Line 31-32: space-time difference; source of difference; unbalanced development.
Point 7:
The abstract is not formed correctly. It is very blurry and framed incorrectly. Abstract is a short and concise presentation of a complex study. It seems that the authors have taken certain phrases from the text and thus formed the abstract. The abstract should clearly indicate the purpose of the study, its importance for society (i.e. to characterize the problem), identify the methods and materials of the study, and the conclusions should be clearly and briefly formulated. There is no "starting point" in the abstract, that is, information about previous studies (one sentence is enough). From my point of view, in the abstract, such information begins with the statement: "Previously conducted studies have established that ...".
Response 7:
Thank you for your patient review and guidance. Indeed, as you said, the summary of this article needs to clarify the research purpose, content and methods, and briefly present the results. According to your suggestion, we have revised the summary, mainly including adding the previous research information, the purpose of this study and the method of use. At the same time, the research results were highly condensed and re- presented in the abstract. The number of words in the revised abstract was controlled within 216 words. The specific modifications are as follows:
Line 14-30:Abstract: Previously conducted studies have established that digital economy has a one-way inhibition effect on carbon emissions. Under this background, this paper aims to analyze the coordinated development effect of the interaction between digital economy and carbon emissions reduction. The entropy weight method, coupling and coordination degree model, Dagum Gini coefficient and Moran's I index have been carried out as research methods in this paper. The results showed that: (1) The coupling and coordination of China's digital economy and carbon emissions reduction shows an overall growth trend, but the coupling and coordination among regions, provinces and cities show a large imbalance. (2) In the sample period, the overall difference of the coupling and coordination between digital economy development and carbon emissions reduction shows an expanding trend, and the overall difference results are attributed to regional differences. (3)There is a significant spatial correlation in the coupling and coordination degree of digital economy development and carbon emissions reduction among cities. The paper systematically grasps the status of coupling and coordination development, the source of difference and spatial correlation between digital economy and carbon reduction in Chinese cities. A dependence relationship has been established which is digital economy development and carbon emissions reduction, and an interactive promotion pattern has been revealed between digital economy system and carbon emissions reduction system.
Point 8:
3.1) It is desirable to avoid narrative text in the abstract.
Response 8:
Thank you for your suggestion. We re examined the abstract and found that the narrative content in the original version was mainly focused on the description of the research results, which was too lengthy and repeated with the text. Therefore, in order to revise this place, we have re summarized the research results, mainly through highly concise language, and presented the results concisely, so that readers can quickly capture the research focus of this paper.
Line 18-25: The results showed that: (1) The coupling and coordination of China's digital economy and carbon emissions reduction shows an overall growth trend, but the coupling and coordination among regions, provinces and cities show a large imbalance. (2) In the sample period, the overall difference of the coupling and coordination between digital economy development and carbon emissions reduction shows an expanding trend, and the overall difference results are attributed to regional differences. (3)There is a significant spatial correlation in the coupling and coordination degree of digital economy development and carbon emissions reduction among cities.
Point 9:
3.2) Try to use words and phrases: an analysis has been carried out; studied; developed; proposed; established and so on. It is advisable to start sentences in the abstract with these words and phrases.
Response 9:
Thank you very much for your advice. We very much agree that using the words and phrases you recommend in the summary will make this part more logical and clear. Therefore, we adopted the words and phrases you recommended when revising the summary. The specific modifications are as follows:
Line 17-18: The entropy weight method, coupling and coordination degree model, Dagum Gini coefficient and Moran's I index have been carried out as research methods in this paper.
Line 27-30: A dependence relationship has been established which is digital economy development and carbon emissions reduction, and an interactive promotion pattern has been revealed between digital economy system and carbon emissions reduction system.
Point 10:
3.3) At the end of the abstract, it is necessary to indicate the final result obtained by the authors, for example: A model has been developed that allows ...; A dependence has been established which is...; A pattern has been revealed...; An efficient system (technology) has been proposed, and so on. The abstract should be revised.
Response 10:
Thank you for your suggestion. We have studied the articles published in International Journal of Environmental Research and Public Health and found that, as you said, it is necessary to give the final findings of good articles at the end. Therefore, we refer to your tips and give the final results at the end of the summary. The specific modifications are as follows:
Line 25-30: The paper systematically grasps the status of coupling and coordination development, the source of difference and spatial correlation between digital economy and carbon reduction in Chinese cities. A dependence relationship has been established which is digital economy development and carbon emissions reduction, and an interactive promotion pattern has been revealed between digital economy system and carbon emissions reduction system.
Point 11:
The manuscript has a sufficient reference list (43 references in total). At the same time there is no comprehensive coverage of studies in terms of geography of citations. There are not enough references to international studies in the field. There is reference to the work of foreign researchers, which dates to 1997. The list of references is intended to demonstrate the depth of the author's study of the material, the relevance and interest of their research.
Response 11:
Thank you for your patient review and comments. We checked the number and content of article references. Compared with articles published in International Journal of Environmental Research and Public Health, we really need to expand the number of references cited to improve the comprehensiveness of existing research. From the content point of view, it is necessary to sort out the research contents of different countries in the world, which is conducive to reflecting the breadth and depth of this study, so as to attract more international readers' interest in this paper. In the process of revision, we added 31 new articles, of which about 29 were studied by international scholars. The specific modifications are as follows:
Line 46-49: The digital economy relies on technological innovation to continuously promote industrial integration [1] and economic restructuring, while the transformation of industrial and economic development mode will directly cause changes in carbon dioxide emissions [2].
Line 76-77: Coordination enables the two systems to work together effectively and sustainably [4,5].
Line 82-83: This reflection of the connection between the two systems is coupling [6].
Line 93-94: There is a long way to go to promote the carbon emissions reduction process [7].
Line 100-108: It is believed that the real digital economy refers to the part of economic output completely determined by digital technology, with a business framework based on digital services and goods [8].. With the integration of communication and computing technologies in the network and the flow of technology and data, it is strengthening the transformation of e-commerce and large-scale business [9]. Computer aided data flow has enabled digitization to transform various parts of the current economy [10]. Therefore, the digital economy is also divided into four different parts: digital services and goods; Mixed digital services and goods; IT based intensive production services; And IT industry [11].
Line 117-124: Scholar first discussed the boundary problem of digital economy, which provides a useful reference for measuring digital economy [16]. The digital economy is also divided into five parts for measurement, namely infrastructure, e-commerce, industry structure, digital labor and digital price [17]. Some organizations have proposed an ac-counting framework with intelligent infrastructure investment, social promotion, in-novation release, growth and employment as the main body to measure the digital economy [18]. In addition, there are also scholars who calculate the digital economy by building a comprehensive indicator system [19-21].
Line 131-138: A study took a telecommunications company in Slovenia as the research object, calculated the greenhouse gas emissions within the enterprise scope 3, and took the purchase of electricity, the commuting of employees, the use of vehicles owned by the company and heating as the measured objects [24]. In addition, the research calculates the grid emission factors, deduces the emission factors of specific European countries, and calculates the carbon intensity based on the power generation of power plants [25]. The other is the exploration of carbon emissions reduction path. It is generally believed that economic development is closely related to carbon emissions [26].
Line 140-141: the analysis of the policy effect of carbon emissions reduction pilot [29,30].
Line 145-152: Meanwhile, there is also study carried out from the perspective of energy, believing that the development of renewable energy is of great significance for carbon emissions reduction in resource-based countries [34]. Another study assessed the existing climate policy portfolio and believed that the energy technology portfolio needs to be adjusted [35]. Some scholars discussed the environmental impact of mineral mining based on the mining industry [36,37]. The adjustment of savings also contributes to the development of green economy [38].
Added References:
- Zuo, P.; Jiang, Q.; Chen, J. Internet development, urbanization and the upgrading of China's industrial structure. Quantitative Economics and Technical Economics. 2020, 7: 71-91.
- Hosseinzadeh, B.H.; Nabavi, P. A.; Khanali, M.; Ghahderijani, M.; Chau, K. Application of data envelopment analysis ap-proach for optimization of energy use and reduction of greenhouse gas emission in peanut production of Iran. Journal of Cleaner Production. 2018,172:1327-1335.
- Kunz, N.C.; Moran, C.J.; Kastelle, T. Conceptualising “coupling” for sustainability implementation in the industrial sector: a review of the field and projection of future research opportunities. Journal of Cleaner Production. 2013, 53:69-80.
- Carl, F. Resilience: the emergence of a perspective for social-ecological systems analyses. Global Environmental Change. 2006, 16 (3):253-267.
- Solymar, L., Webb, D., Grunnet, J, A. The Physics and Applications of Photorefractive Materials. 1996, 11,Clarendon Press.
- Yan, B.; Erda, W.; Dominik, M.; Martin, L. How population migration affects carbon emissions in China: Factual and coun-terfactual scenario analysis. Technological Forecasting and Social Change. 2022,184: 122023
- Luyanda, D.W. Concepts of digital economy and industry 4.0 in intelligent and information systems. International Journal of Intelligent Networks. 2021,2:122-129.
- Mahmod, S. A. 5G wireless technologies- future generation communication technologies. International Journal of Computing and Digital Systemss. 2017, 6 (3):139-147.
- Chouhan, N.; Rathore, D. Role of digitalization after demonetization in economy. International Journal of Computer Sciences and Engineering. 2018, 6(9): 88-90.
- Szeto, K. Keeping score, digitally. Music Reference Services Quarterly. 2018, 21 (2):98-100.
- Margheriol, D. H.; Cook, S.; Montes, S. The emerging digital economy. US Department of Commerce. 1998.
- Kahin, B.; Brynjolfsson, E. Understanding the digital economy. The MIT Press. 2000.
- OECD. Measuring the digital economy: an new perspective. Pairs: OECD Publishing. 2014.
- Shahzad, M.; Wang, J.; Dong, K.; Zhao, J. The impact of digital economy on energy transition across the globe: the mediating role of government governance. Renewable and Sustainable Energy Reviews. 2022, 166 (3) :112620.
- Gregor, R.; Saša, T. Carbon footprint calculation in telecommunications companies – The importance and relevance of scope 3 greenhouse gases emissions. Renewable and Sustainable Energy Reviews. 2018, 98: 361-375.
- Jan, F.U.; Anke, W.; Leonhard, G.; Mirko, S. Open-data based carbon emission intensity signals for electricity generation in European countries – top down vs. bottom up approach. Cleaner Energy Systems. 2022, 3: 100018
- Batrancea, I.; Rathnaswamy, M.K.; Batrancea, L.; Nichita, A.; Gaban, L.; Fatacean, G.; Tulai, H.; Bircea, I.; Rus, M. I. A panel data analysis on sustainable economic growth in India, Brazil, and Romania. Journal of Risk and Financial Management. 2020, 13(8): 170.
- Huo, W.D.; Qi, J.; Yang, T.; Liu, J.L.; Liu, M.M.; Zhou, Z.Q. Effects of China's pilot low-carbon city policy on carbon emission reduction: A quasi-natural experiment based on satellite data. Technological Forecasting and Social Change. 2022,175: 121422.
- Liu, X.; Li, Y.C.; Chen, X.H.; Liu, J. Evaluation of low carbon city pilot policy effect on carbon abatement in China: An em-pirical evidence based on time-varying DID model. Cities. 2022, 123: 103582.
- Tcvetkov, P. Engagement of resource-based economies in the fight against rising carbon emissions. Energy Reports. 2022, 8: 874-883.
- Tcvetkov, P. Climate policy imbalance in the energy sector: Time to focus on the value of CO2 utilization. Energies. 2021, 14(2): 411.
- Khayrutdinov, M.M.; Golik, V.I.; Aleksakhin, A.V.; Trushina, E.V.; Lazareva, N.V.; Aleksakhina, Y.V. Proposal of an Algo-rithm for Choice of a Development System for Operational and Environmental Safety in Mining. Resources. 2022, 11: 88.
- Kongar, S.C.; Ubysz, A.; Faradzhov, V. Models and algorithms of choice of development technology of deposits when se-lecting the composition of the backfilling mixture. IOP Conference Series Earth and Environmental Science. 2021, 684(1): 012008.
- Batrancea, L., Rathnaswamy, M.M., Batrancea, I., Nichita, A., Rus, M.-I., Tulai, H., Fatacean, G., Masca, E.S., Morar, I.D. Adjusted net savings of CEE and Baltic nations in the context of sustainable economic growth: A panel data analysis. Journal of Risk and Financial Management. 2020, 13(10): 234.
- Tonni, A.K.; Aleksandra, M.; Marina, K.; Elena,B.; Mohd, H.D.O.; Hui, H.G. Accelerating sustainability transition in St. Petersburg (Russia) through digitalization-based circular economy in waste recycling industry: A strategy to promote carbon neutrality in era of Industry 4.0. Journal of Cleaner Production. 2022,363: 132452
- Jacob, P. Information and communication technology in shaping urban low carbon development pathways.Current Opinion in Environmental Sustainability. 2018, 30:133-137
- Muhammad, S.; Wang, J.D.; Dong, K.Y.; Zhao, J. The impact of digital economy on energy transition across the globe: The mediating role of government governance. Renewable and Sustainable Energy Reviews. 2022, 166:112620,
- Moyer, J.D.; Hughes, B.B. ICTs: Do they contribute to increased carbon emissions?. Technological Forecasting & Social Change. 2012, 79 (5) :919-931
- Asongu, S.A.; Roux, S. L.; Biekpe, N. Enhancing ICT for environmental sustainability in sub-Saharan Africa. Technological Forecasting & Social Change. 2018, 127: 209-21.
- Ilhan, O.; Sana, U. Does digital financial inclusion matter for economic growth and environmental sustainability in OBRI economies? An empirical analysis. Resources, Conservation and Recycling. 2022, 185:106489.
- Salahuddin, M.; Alam, K. Internet usage, electricity consumption and economic growth in Australia: a time series evidence. Telematics and Informatics. 2015, 32 (4): 862-878.
Point 12:
4.1.) The depth of study is demonstrated with the number of references - is not enough.
Response 12:
Thank you very much for your advice. Indeed, rich references are necessary for a high-quality paper. On the one hand, sufficient references can reflect the completeness of literature sorting workload in the early stage, which can provide reliable support for the problems that have not been solved in the research. On the other hand, sufficient references show that research in various countries around the world has been analyzed, which is helpful to reflect the depth of research. Therefore, according to your suggestion, we have added 31 articles, especially 29 international articles, to better enrich the research content of this paper. After improving the reference part, the number of references cited in this paper is 70. The specific newly added references are as follows:
- Zuo, P.; Jiang, Q.; Chen, J. Internet development, urbanization and the upgrading of China's industrial structure. Quantitative Economics and Technical Economics. 2020, 7: 71-91.
- Hosseinzadeh, B.H.; Nabavi, P. A.; Khanali, M.; Ghahderijani, M.; Chau, K. Application of data envelopment analysis ap-proach for optimization of energy use and reduction of greenhouse gas emission in peanut production of Iran. Journal of Cleaner Production. 2018,172:1327-1335.
- Kunz, N.C.; Moran, C.J.; Kastelle, T. Conceptualising “coupling” for sustainability implementation in the industrial sector: a review of the field and projection of future research opportunities. Journal of Cleaner Production. 2013, 53:69-80.
- Carl, F. Resilience: the emergence of a perspective for social-ecological systems analyses. Global Environmental Change. 2006, 16 (3):253-267.
- Solymar, L., Webb, D., Grunnet, J, A. The Physics and Applications of Photorefractive Materials. 1996, 11,Clarendon Press.
- Yan, B.; Erda, W.; Dominik, M.; Martin, L. How population migration affects carbon emissions in China: Factual and coun-terfactual scenario analysis. Technological Forecasting and Social Change. 2022,184: 122023
- Luyanda, D.W. Concepts of digital economy and industry 4.0 in intelligent and information systems. International Journal of Intelligent Networks. 2021,2:122-129.
- Mahmod, S. A. 5G wireless technologies- future generation communication technologies. International Journal of Computing and Digital Systemss. 2017, 6 (3):139-147.
- Chouhan, N.; Rathore, D. Role of digitalization after demonetization in economy. International Journal of Computer Sciences and Engineering. 2018, 6(9): 88-90.
- Szeto, K. Keeping score, digitally. Music Reference Services Quarterly. 2018, 21 (2):98-100.
- Margheriol, D. H.; Cook, S.; Montes, S. The emerging digital economy. US Department of Commerce. 1998.
- Kahin, B.; Brynjolfsson, E. Understanding the digital economy. The MIT Press. 2000.
- OECD. Measuring the digital economy: an new perspective. Pairs: OECD Publishing. 2014.
- Shahzad, M.; Wang, J.; Dong, K.; Zhao, J. The impact of digital economy on energy transition across the globe: the mediating role of government governance. Renewable and Sustainable Energy Reviews. 2022, 166 (3) :112620.
- Gregor, R.; Saša, T. Carbon footprint calculation in telecommunications companies – The importance and relevance of scope 3 greenhouse gases emissions. Renewable and Sustainable Energy Reviews. 2018, 98: 361-375.
- Jan, F.U.; Anke, W.; Leonhard, G.; Mirko, S. Open-data based carbon emission intensity signals for electricity generation in European countries – top down vs. bottom up approach. Cleaner Energy Systems. 2022, 3: 100018
- Batrancea, I.; Rathnaswamy, M.K.; Batrancea, L.; Nichita, A.; Gaban, L.; Fatacean, G.; Tulai, H.; Bircea, I.; Rus, M. I. A panel data analysis on sustainable economic growth in India, Brazil, and Romania. Journal of Risk and Financial Management. 2020, 13(8): 170.
- Huo, W.D.; Qi, J.; Yang, T.; Liu, J.L.; Liu, M.M.; Zhou, Z.Q. Effects of China's pilot low-carbon city policy on carbon emission reduction: A quasi-natural experiment based on satellite data. Technological Forecasting and Social Change. 2022,175: 121422.
- Liu, X.; Li, Y.C.; Chen, X.H.; Liu, J. Evaluation of low carbon city pilot policy effect on carbon abatement in China: An em-pirical evidence based on time-varying DID model. Cities. 2022, 123: 103582.
- Tcvetkov, P. Engagement of resource-based economies in the fight against rising carbon emissions. Energy Reports. 2022, 8: 874-883.
- Tcvetkov, P. Climate policy imbalance in the energy sector: Time to focus on the value of CO2 utilization. Energies. 2021, 14(2): 411.
- Khayrutdinov, M.M.; Golik, V.I.; Aleksakhin, A.V.; Trushina, E.V.; Lazareva, N.V.; Aleksakhina, Y.V. Proposal of an Algo-rithm for Choice of a Development System for Operational and Environmental Safety in Mining. Resources. 2022, 11: 88.
- Kongar, S.C.; Ubysz, A.; Faradzhov, V. Models and algorithms of choice of development technology of deposits when se-lecting the composition of the backfilling mixture. IOP Conference Series Earth and Environmental Science. 2021, 684(1): 012008.
- Batrancea, L., Rathnaswamy, M.M., Batrancea, I., Nichita, A., Rus, M.-I., Tulai, H., Fatacean, G., Masca, E.S., Morar, I.D. Adjusted net savings of CEE and Baltic nations in the context of sustainable economic growth: A panel data analysis. Journal of Risk and Financial Management. 2020, 13(10): 234.
- Tonni, A.K.; Aleksandra, M.; Marina, K.; Elena,B.; Mohd, H.D.O.; Hui, H.G. Accelerating sustainability transition in St. Petersburg (Russia) through digitalization-based circular economy in waste recycling industry: A strategy to promote carbon neutrality in era of Industry 4.0. Journal of Cleaner Production. 2022,363: 132452
- Jacob, P. Information and communication technology in shaping urban low carbon development pathways.Current Opinion in Environmental Sustainability. 2018, 30:133-137
- Muhammad, S.; Wang, J.D.; Dong, K.Y.; Zhao, J. The impact of digital economy on energy transition across the globe: The mediating role of government governance. Renewable and Sustainable Energy Reviews. 2022, 166:112620,
- Moyer, J.D.; Hughes, B.B. ICTs: Do they contribute to increased carbon emissions?. Technological Forecasting & Social Change. 2012, 79 (5) :919-931
- Asongu, S.A.; Roux, S. L.; Biekpe, N. Enhancing ICT for environmental sustainability in sub-Saharan Africa. Technological Forecasting & Social Change. 2018, 127: 209-21.
- Ilhan, O.; Sana, U. Does digital financial inclusion matter for economic growth and environmental sustainability in OBRI economies? An empirical analysis. Resources, Conservation and Recycling. 2022, 185:106489.
- Salahuddin, M.; Alam, K. Internet usage, electricity consumption and economic growth in Australia: a time series evidence. Telematics and Informatics. 2015, 32 (4): 862-878.
Point 13:
4.2.) Relevance – with the availability of research in recent years – is enough.
Response 13:
Thank you for your affirmation. We pay great attention to the relevance and timeliness of the literature when sorting out the literature. We have sorted out the relevant literature based on the theme of digital economy and carbon emissions in this paper, especially the research and discussion over the past three years, which we will cite as an important literature. Thank you again for your affirmation. We will keep this way in the future paper writing. Thank you very much.
Point 14:
4.3.) Interest – with the availability of research by scientists from different countries - is not enough (practically absent). Since you are publishing your manuscript in an international publication, it is necessary to demonstrate the international relevance and interest of this issue. This can be done by analyzing the studies of scientists from different countries. It is imperative to supplement the list of references with studies of scientists from different countries over the past 3-5 years to show geographical (general/global) interest and relevance. The List of References needs to be revised.
Response 14:
Thank you very much for your advice. Indeed, as you said, it is necessary to publish articles in international journals and quote international papers, which will help attract more international scholars and readers to view this paper. At the same time, the research achievements of scholars from different countries have been extensively understood and cited, which will help to improve the internationalization of this paper and the breadth and depth of research. Therefore, at your suggestion and in combination with the content of the paper, we have added 31 new articles in the past 3-5 years, including 29 international articles, including Iran, Slovenia, the United Kingdom and African countries, to highlight the interesting and comprehensive nature of this discussion. In addition, we also expanded the literature from multiple perspectives, including adding research from different perspectives such as economic development and mining industry. The supplement to the multi angle discussion of carbon emission reduction path can enrich the carding boundary of the existing literature in this paper, thus increasing the comprehensiveness of this study. Finally, we updated the reference list synchronously after adding all the literatures. The specific newly added documents are as follows:
- Zuo, P.; Jiang, Q.; Chen, J. Internet development, urbanization and the upgrading of China's industrial structure. Quantitative Economics and Technical Economics. 2020, 7: 71-91.
- Hosseinzadeh, B.H.; Nabavi, P. A.; Khanali, M.; Ghahderijani, M.; Chau, K. Application of data envelopment analysis ap-proach for optimization of energy use and reduction of greenhouse gas emission in peanut production of Iran. Journal of Cleaner Production. 2018,172:1327-1335.
- Kunz, N.C.; Moran, C.J.; Kastelle, T. Conceptualising “coupling” for sustainability implementation in the industrial sector: a review of the field and projection of future research opportunities. Journal of Cleaner Production. 2013, 53:69-80.
- Carl, F. Resilience: the emergence of a perspective for social-ecological systems analyses. Global Environmental Change. 2006, 16 (3):253-267.
- Solymar, L., Webb, D., Grunnet, J, A. The Physics and Applications of Photorefractive Materials. 1996, 11,Clarendon Press.
- Yan, B.; Erda, W.; Dominik, M.; Martin, L. How population migration affects carbon emissions in China: Factual and coun-terfactual scenario analysis. Technological Forecasting and Social Change. 2022,184: 122023
- Luyanda, D.W. Concepts of digital economy and industry 4.0 in intelligent and information systems. International Journal of Intelligent Networks. 2021,2:122-129.
- Mahmod, S. A. 5G wireless technologies- future generation communication technologies. International Journal of Computing and Digital Systemss. 2017, 6 (3):139-147.
- Chouhan, N.; Rathore, D. Role of digitalization after demonetization in economy. International Journal of Computer Sciences and Engineering. 2018, 6(9): 88-90.
- Szeto, K. Keeping score, digitally. Music Reference Services Quarterly. 2018, 21 (2):98-100.
- Margheriol, D. H.; Cook, S.; Montes, S. The emerging digital economy. US Department of Commerce. 1998.
- Kahin, B.; Brynjolfsson, E. Understanding the digital economy. The MIT Press. 2000.
- OECD. Measuring the digital economy: an new perspective. Pairs: OECD Publishing. 2014.
- Shahzad, M.; Wang, J.; Dong, K.; Zhao, J. The impact of digital economy on energy transition across the globe: the mediating role of government governance. Renewable and Sustainable Energy Reviews. 2022, 166 (3) :112620.
- Gregor, R.; Saša, T. Carbon footprint calculation in telecommunications companies – The importance and relevance of scope 3 greenhouse gases emissions. Renewable and Sustainable Energy Reviews. 2018, 98: 361-375.
- Jan, F.U.; Anke, W.; Leonhard, G.; Mirko, S. Open-data based carbon emission intensity signals for electricity generation in European countries – top down vs. bottom up approach. Cleaner Energy Systems. 2022, 3: 100018
- Batrancea, I.; Rathnaswamy, M.K.; Batrancea, L.; Nichita, A.; Gaban, L.; Fatacean, G.; Tulai, H.; Bircea, I.; Rus, M. I. A panel data analysis on sustainable economic growth in India, Brazil, and Romania. Journal of Risk and Financial Management. 2020, 13(8): 170.
- Huo, W.D.; Qi, J.; Yang, T.; Liu, J.L.; Liu, M.M.; Zhou, Z.Q. Effects of China's pilot low-carbon city policy on carbon emission reduction: A quasi-natural experiment based on satellite data. Technological Forecasting and Social Change. 2022,175: 121422.
- Liu, X.; Li, Y.C.; Chen, X.H.; Liu, J. Evaluation of low carbon city pilot policy effect on carbon abatement in China: An em-pirical evidence based on time-varying DID model. Cities. 2022, 123: 103582.
- Tcvetkov, P. Engagement of resource-based economies in the fight against rising carbon emissions. Energy Reports. 2022, 8: 874-883.
- Tcvetkov, P. Climate policy imbalance in the energy sector: Time to focus on the value of CO2 utilization. Energies. 2021, 14(2): 411.
- Khayrutdinov, M.M.; Golik, V.I.; Aleksakhin, A.V.; Trushina, E.V.; Lazareva, N.V.; Aleksakhina, Y.V. Proposal of an Algo-rithm for Choice of a Development System for Operational and Environmental Safety in Mining. Resources. 2022, 11: 88.
- Kongar, S.C.; Ubysz, A.; Faradzhov, V. Models and algorithms of choice of development technology of deposits when se-lecting the composition of the backfilling mixture. IOP Conference Series Earth and Environmental Science. 2021, 684(1): 012008.
- Batrancea, L., Rathnaswamy, M.M., Batrancea, I., Nichita, A., Rus, M.-I., Tulai, H., Fatacean, G., Masca, E.S., Morar, I.D. Adjusted net savings of CEE and Baltic nations in the context of sustainable economic growth: A panel data analysis. Journal of Risk and Financial Management. 2020, 13(10): 234.
- Tonni, A.K.; Aleksandra, M.; Marina, K.; Elena,B.; Mohd, H.D.O.; Hui, H.G. Accelerating sustainability transition in St. Petersburg (Russia) through digitalization-based circular economy in waste recycling industry: A strategy to promote carbon neutrality in era of Industry 4.0. Journal of Cleaner Production. 2022,363: 132452
- Jacob, P. Information and communication technology in shaping urban low carbon development pathways. Current Opinion in Environmental Sustainability. 2018, 30:133-137
- Muhammad, S.; Wang, J.D.; Dong, K.Y.; Zhao, J. The impact of digital economy on energy transition across the globe: The mediating role of government governance. Renewable and Sustainable Energy Reviews. 2022, 166:112620,
- Moyer, J.D.; Hughes, B.B. ICTs: Do they contribute to increased carbon emissions?. Technological Forecasting & Social Change. 2012, 79 (5) :919-931
- Asongu, S.A.; Roux, S. L.; Biekpe, N. Enhancing ICT for environmental sustainability in sub-Saharan Africa. Technological Forecasting & Social Change. 2018, 127: 209-21.
- Ilhan, O.; Sana, U. Does digital financial inclusion matter for economic growth and environmental sustainability in OBRI economies? An empirical analysis. Resources, Conservation and Recycling. 2022, 185:106489.
- Salahuddin, M.; Alam, K. Internet usage, electricity consumption and economic growth in Australia: a time series evidence. Telematics and Informatics. 2015, 32 (4): 862-878.
Point 15:
It is necessary to avoid group references, for example: [17-20]. From my point of view, allowed up to three; more than three references are not acceptable and must be deciphered. Each paper you refer is unique and the studies you refer deserve more proper and careful review to demonstrate (and prove) its importance for the current research. It is necessary to demonstrate in detail the essence of each study and their need for your work.
Response 15:
Thank you very much for your guidance and suggestions. We agree with you. Indeed, the citation of each document needs to reflect its unique research value. The combination and citation of too many documents may reduce its own value. Thank you for your reminding, which will be of great help to us in the high-quality presentation of literature in the future paper writing. Therefore, according to your suggestions, we have revised the literature citation. First, we checked the literature citations of the full text and deleted more than three citations to improve the quality of literature citations. Secondly, we checked the placement of each document to ensure that the citation of each document conforms to the corresponding content of the article. Finally, we present the essence of each study to ensure that each reference can reflect its own value. The specific modifications are as follows:
Line 76-77: Coordination enables the two systems to work together effectively and sustainably [4,5].
Line 82-83: This reflection of the connection between the two systems is coupling [6].
Line 93-94: There is a long way to go to promote the carbon emissions reduction process [7].
Line 100-107: It is believed that the real digital economy refers to the part of economic output completely determined by digital technology, with a business framework based on digital services and goods [8]. With the integration of communication and computing technologies in the network and the flow of technology and data, it is strengthening the transformation of e-commerce and large-scale business [9]. Computer aided data flow has enabled digitization to transform various parts of the current economy [10]. Therefore, the digital economy is also divided into four different parts: digital services and goods; Mixed digital services and goods; IT based intensive production services; And IT industry [11].
Line 117-124: Scholar first discussed the boundary problem of digital economy, which provides a useful reference for measuring digital economy [16]. The digital economy is also divided into five parts for measurement, namely infrastructure, e-commerce, industry structure, digital labor and digital price [17]. Some organizations have proposed an ac-counting framework with intelligent infrastructure investment, social promotion, in-novation release, growth and employment as the main body to measure the digital economy [18]. In addition, there are also scholars who calculate the digital economy by building a comprehensive indicator system [19-21].
Line 131-138: A study took a telecommunications company in Slovenia as the research object, calculated the greenhouse gas emissions within the enterprise scope 3, and took the purchase of electricity, the commuting of employees, the use of vehicles owned by the company and heating as the measured objects [24]. In addition, the research calculates the grid emission factors, deduces the emission factors of specific European countries, and calculates the carbon intensity based on the power generation of power plants [25]. The other is the exploration of carbon emissions reduction path. It is generally believed that economic development is closely related to carbon emissions [26].
Line 138-141: The exploration of carbon emissions reduction path mainly involves the improvement of carbon emissions trading market [27,28], the analysis of the policy effect of carbon emissions reduction pilot [29,30] and the research of other paths [31].
Line 145-152: Meanwhile, there is also study carried out from the perspective of energy, believing that the development of renewable energy is of great significance for carbon emissions re-duction in resource-based countries [34]. Another study assessed the existing climate policy portfolio and believed that the energy technology portfolio needs to be adjusted [35]. Some scholars discussed the environmental impact of mineral mining based on the mining industry [36,37]. The adjustment of savings also contributes to the development of green economy [38].
Line 158-163: It is found that through digital waste management, the consumption of raw materials can be reduced by 25% by 2030, and half of greenhouse gas emissions can be avoided [43]. There is study believed that ICT can contribute to the construction of low-carbon cities [44], mainly through governance capacity of the government [45] and energy systems [46]. Study have also targeted African countries and found an inverted "U" relationship between ICT and carbon emissions in the sample countries [47].
Line 170-171: Different from this conclusion, other scholars believe that the development of digital economy is unfavorable to carbon emission reduction [53,54].
Added References:
- Zuo, P.; Jiang, Q.; Chen, J. Internet development, urbanization and the upgrading of China's industrial structure. Quantitative Economics and Technical Economics. 2020, 7: 71-91.
- Hosseinzadeh, B.H.; Nabavi, P. A.; Khanali, M.; Ghahderijani, M.; Chau, K. Application of data envelopment analysis ap-proach for optimization of energy use and reduction of greenhouse gas emission in peanut production of Iran. Journal of Cleaner Production. 2018,172:1327-1335.
- Kunz, N.C.; Moran, C.J.; Kastelle, T. Conceptualising “coupling” for sustainability implementation in the industrial sector: a review of the field and projection of future research opportunities. Journal of Cleaner Production. 2013, 53:69-80.
- Carl, F. Resilience: the emergence of a perspective for social-ecological systems analyses. Global Environmental Change. 2006, 16 (3):253-267.
- Solymar, L., Webb, D., Grunnet, J, A. The Physics and Applications of Photorefractive Materials. 1996, 11,Clarendon Press.
- Yan, B.; Erda, W.; Dominik, M.; Martin, L. How population migration affects carbon emissions in China: Factual and coun-terfactual scenario analysis. Technological Forecasting and Social Change. 2022,184: 122023
- Luyanda, D.W. Concepts of digital economy and industry 4.0 in intelligent and information systems. International Journal of Intelligent Networks. 2021,2:122-129.
- Mahmod, S. A. 5G wireless technologies- future generation communication technologies. International Journal of Computing and Digital Systemss. 2017, 6 (3):139-147.
- Chouhan, N.; Rathore, D. Role of digitalization after demonetization in economy. International Journal of Computer Sciences and Engineering. 2018, 6(9): 88-90.
- Szeto, K. Keeping score, digitally. Music Reference Services Quarterly. 2018, 21 (2):98-100.
- Margheriol, D. H.; Cook, S.; Montes, S. The emerging digital economy. US Department of Commerce. 1998.
- Kahin, B.; Brynjolfsson, E. Understanding the digital economy. The MIT Press. 2000.
- OECD. Measuring the digital economy: an new perspective. Pairs: OECD Publishing. 2014.
- Shahzad, M.; Wang, J.; Dong, K.; Zhao, J. The impact of digital economy on energy transition across the globe: the mediating role of government governance. Renewable and Sustainable Energy Reviews. 2022, 166 (3) :112620.
- Gregor, R.; Saša, T. Carbon footprint calculation in telecommunications companies – The importance and relevance of scope 3 greenhouse gases emissions. Renewable and Sustainable Energy Reviews. 2018, 98: 361-375.
- Jan, F.U.; Anke, W.; Leonhard, G.; Mirko, S. Open-data based carbon emission intensity signals for electricity generation in European countries – top down vs. bottom up approach. Cleaner Energy Systems. 2022, 3: 100018
- Batrancea, I.; Rathnaswamy, M.K.; Batrancea, L.; Nichita, A.; Gaban, L.; Fatacean, G.; Tulai, H.; Bircea, I.; Rus, M. I. A panel data analysis on sustainable economic growth in India, Brazil, and Romania. Journal of Risk and Financial Management. 2020, 13(8): 170.
- Huo, W.D.; Qi, J.; Yang, T.; Liu, J.L.; Liu, M.M.; Zhou, Z.Q. Effects of China's pilot low-carbon city policy on carbon emission reduction: A quasi-natural experiment based on satellite data. Technological Forecasting and Social Change. 2022,175: 121422.
- Liu, X.; Li, Y.C.; Chen, X.H.; Liu, J. Evaluation of low carbon city pilot policy effect on carbon abatement in China: An em-pirical evidence based on time-varying DID model. Cities. 2022, 123: 103582.
- Tcvetkov, P. Engagement of resource-based economies in the fight against rising carbon emissions. Energy Reports. 2022, 8: 874-883.
- Tcvetkov, P. Climate policy imbalance in the energy sector: Time to focus on the value of CO2 utilization. Energies. 2021, 14(2): 411.
- Khayrutdinov, M.M.; Golik, V.I.; Aleksakhin, A.V.; Trushina, E.V.; Lazareva, N.V.; Aleksakhina, Y.V. Proposal of an Algo-rithm for Choice of a Development System for Operational and Environmental Safety in Mining. Resources. 2022, 11: 88.
- Kongar, S.C.; Ubysz, A.; Faradzhov, V. Models and algorithms of choice of development technology of deposits when se-lecting the composition of the backfilling mixture. IOP Conference Series Earth and Environmental Science. 2021, 684(1): 012008.
- Batrancea, L., Rathnaswamy, M.M., Batrancea, I., Nichita, A., Rus, M.-I., Tulai, H., Fatacean, G., Masca, E.S., Morar, I.D. Adjusted net savings of CEE and Baltic nations in the context of sustainable economic growth: A panel data analysis. Journal of Risk and Financial Management. 2020, 13(10): 234.
- Tonni, A.K.; Aleksandra, M.; Marina, K.; Elena,B.; Mohd, H.D.O.; Hui, H.G. Accelerating sustainability transition in St. Petersburg (Russia) through digitalization-based circular economy in waste recycling industry: A strategy to promote carbon neutrality in era of Industry 4.0. Journal of Cleaner Production. 2022,363: 132452
- Jacob, P. Information and communication technology in shaping urban low carbon development pathways. Current Opinion in Environmental Sustainability. 2018, 30:133-137
- Muhammad, S.; Wang, J.D.; Dong, K.Y.; Zhao, J. The impact of digital economy on energy transition across the globe: The mediating role of government governance. Renewable and Sustainable Energy Reviews. 2022, 166:112620,
- Moyer, J.D.; Hughes, B.B. ICTs: Do they contribute to increased carbon emissions?. Technological Forecasting & Social Change. 2012, 79 (5) :919-931
- Asongu, S.A.; Roux, S. L.; Biekpe, N. Enhancing ICT for environmental sustainability in sub-Saharan Africa. Technological Forecasting & Social Change. 2018, 127: 209-21.
- Ilhan, O.; Sana, U. Does digital financial inclusion matter for economic growth and environmental sustainability in OBRI economies? An empirical analysis. Resources, Conservation and Recycling. 2022, 185:106489.
- Salahuddin, M.; Alam, K. Internet usage, electricity consumption and economic growth in Australia: a time series evidence. Telematics and Informatics. 2015, 32 (4): 862-878.
Point 16:
From my point of view, the authors abuse the names of scientists when mentioning the study. A reference [1] is sufficient. If the reader is interested in the name of the researcher, then it is easy to refer to the references list. It is important for the reader to know the essence (main idea) of the disclosed issue, not the name of the researcher.
Response 16:
Thank you very much. We checked the articles that have been published in International Journal of Environmental Research and Public Health. Indeed, as you said, the citation format of references in the articles needs to be revised. According to your suggestion, we removed the author information corresponding to the references in the article, and only retained the document serial number. At the same time, we carefully checked the correspondence between the serial number of the literature and the information in the references to ensure that the reader can correctly find the relevant author information in the last part. The specific modifications are as follows:
Line 46-49: The digital economy relies on technological innovation to continuously promote industrial integration [1] and economic restructuring, while the transformation of industrial and economic development mode will directly cause changes in carbon dioxide emissions [2].
Line 67-71: Relying on the new digital infrastructure, the digital economy takes knowledge and information as the key production factors, modern information network as the carrier, and the use of information technology as the key driving technology for development, which profoundly affects and fundamentally changes the economic development mode and social activity structure [3].
Line 76-77: Coordination enables the two systems to work together effectively and sustainably [4,5].
Line 82-83: This reflection of the connection between the two systems is coupling [6].
Line 93-94: There is a long way to go to promote the carbon emissions reduction process [7].
Line 100-113: It is believed that the real digital economy refers to the part of economic output completely determined by digital technology, with a business framework based on digital services and goods [8]. With the integration of communication and computing technologies in the network and the flow of technology and data, it is strengthening the transformation of e-commerce and large-scale business [9]. Computer aided data flow has enabled digitization to transform various parts of the current economy [10]. Therefore, the digital economy is also divided into four different parts: digital services and goods; Mixed digital services and goods; IT based intensive production services; And IT industry [11].It is believed that the digital economy is an economic activity based on data application and data technology innovation, with data as the core element [12]. Data technology is the basic support for the development of digital economy. It is a new general technology, mainly through the combination of digital information and the Internet [13], including hardware, software and network technology [14]. There is an opinion that digital economy is a more advanced economic form after industrial economy [15].
Line 114-124: Most of the existing literatures measure the digital economy based on the method of summarizing the digital industry output, digital product classification, and digital related comprehensive indicators. Scholar first discussed the boundary problem of digital economy, which provides a useful reference for measuring digital economy [16]. The digital economy is also divided into five parts for measurement, namely infrastructure, e-commerce, industry structure, digital labor and digital price [17]. Some organizations have proposed an accounting framework with intelligent infrastructure investment, social promotion, innovation release, growth and employment as the main body to measure the digital economy [18]. In addition, there are also scholars who calculate the digital economy by building a comprehensive indicator system [19-21].
Line 126-153: The calculation of carbon dioxide emissions is mainly based on different energy consumption. China's industrial energy carbon emissions have been calculated based on industrial energy consumption [22]. Based on the IPCC inventory method, a research uses the multiplication method of different fuel consumption and fuel carbon emission factors to calculate the regional power carbon emissions [23]. A study took a telecommunications company in Slovenia as the research object, calculated the greenhouse gas emissions within the enterprise scope 3, and took the purchase of electricity, the commuting of employees, the use of vehicles owned by the company and heating as the measured objects [24]. In addition, the research calculates the grid emission factors, deduces the emission factors of specific European countries, and calculates the carbon intensity based on the power generation of power plants [25]. The other is the exploration of carbon emissions reduction path. It is generally believed that economic development is closely related to carbon emissions [26]. The exploration of carbon emissions reduction path mainly involves the improvement of carbon emissions trading market [27,28], the analysis of the policy effect of carbon emissions reduction pilot [29,30] and the research of other paths [31]. In the study of other carbon emissions reduction paths, there is other scholar believed that carbon capture, utilization and storage technology (CCUS) is the only technological path to achieve net zero emissions [32]. And another research believed that the promotion of " Nature Based Solution" (NBS) can provide positive reference for China to solve climate and environmental problems [33]. Mean-while, there is also study carried out from the perspective of energy, believing that the development of renewable energy is of great significance for carbon emissions reduction in resource-based countries [34]. Another study assessed the existing climate policy portfolio and believed that the energy technology portfolio needs to be adjusted [35]. Some scholars discussed the environmental impact of mineral mining based on the mining industry [36,37]. The adjustment of savings also contributes to the development of green economy [38]. In addition, international climate cooperation and climate assistance can also effectively achieve carbon emissions reduction targets [39].
Line 155-171: One is the impact of digital economy development on carbon emissions. Some scholars believe that the digital economy can inhibit carbon emissions [40-42]. A study have proved the role of Taiwan's digital development in the circular economy. It is found that through digital waste management, the consumption of raw materials can be reduced by 25% by 2030, and half of greenhouse gas emissions can be avoided [43]. There is study believed that ICT can contribute to the construction of low-carbon cities [44], mainly through governance capacity of the government [45] and energy systems [46]. Study have also targeted African countries and found an inverted "U" relationship between ICT and carbon emissions in the sample countries [47]. In addition, some scholars believe that R&D investment played a positive role in regulating the relationship between digital economy development and carbon emissions reduction [48-50]. It can be seen that the development of the digital economy has a positive contribution to the promotion of China's green development process [51]. Therefore, it is necessary to study the impact of the digital economy on the climate and environment, which has reference significance for the formulation and improvement of economic and environmental policies [52]. Different from this conclusion, other scholars believe that the development of digital economy is unfavorable to carbon emission reduction [53,54].
Line 176-177: It can be considered that low carbon development can help reverse enterprises' digital transformation [55].
Line 185-187: The application of digital technology can optimize the end treatment technology of enterprise carbon emissions [56,57], and promote carbon emission reduction [58,59].
Line 224-225: For the digital economy, there is scholar believed that there are three layers from the inside to the outside [60].
Line 251-252: The number of enterprises based on Internet development can represent the application level of digital infrastructure [61].
Line 278-280: Number of Internet users. The number of Internet users can reflect the development of digital economy related businesses [62].
Line 287-288: Therefore, it can be calculated according to the digital inclusive financial index of prefecture level cities based on the compilation of indicators by [63].
Line 289-291: The calculation of carbon emissions reduction system mainly refers to the construction method of the green development system, and the sub index construction is based on the input-output model [64]. Low carbon is one of the core concepts of green development [65], so it is reasonable to use the green development system for reference in the construction of carbon emissions reduction system.
Line 301-304: For the carbon dioxide emissions accounting of prefecture level cities, this paper selects the carbon emissions of industrial consumption of natural gas, liquefied petroleum gas and industrial coal-fired electricity to sum up [66].
Line 338-339: Coordination means that two or more systems are interrelated and properly coordinated to form a mutually beneficial development situation [67,68].
Line 379: carbon emissions reduction system [69].
Line 398-401: Dagum Gini coefficient and its decomposition method is to decompose the overall Gini coefficient G into intra group (intra regional) differential contribution , inter group (inter regional) difference contribution and hypervariable density contribution difference , three parts [70].
Point 17:
From my point of view, at the end of the introduction the authors did not quite correctly formulate a brief conclusion of the analytical study of previously performed works. The authors did not summarize their analysis and did not identify unresolved issues. This conclusion should make it possible to characterize the actual question posed, the purpose of the study and the tasks to be solved to achieve this goal. For example: Analyzing the above, it can be noted that ... is a very topical issue. Therefore, the purpose of this study is ... and to achieve this, it is necessary to solve the following tasks: 1); 2); ... Such a conclusion allows the reader to understand the vector of the study, and the authors to correctly formulate the conclusions. It needs to be improved.
Response 17:
Thank you for your suggestion. Indeed, as you said, it is necessary to make a brief summary of all literature at the end of the introduction, so as to propose the boundaries of existing research and the problems to be solved. At the end of the introduction, we also agree with your point of view that the purpose and task of this study should be clarified. Therefore, we have modified the end of the introduction based on the phrases you recommended. The main modifications include three aspects. First, the existing research is summarized and the unsolved problems of the existing research are proposed. Secondly, the research purpose and significance of this paper are clearly put forward. Finally, three research tasks of this paper are proposed, and the research tasks are corresponding to the research contents and conclusions, so as to improve the logic of the structure of this paper. The specific modifications are as follows:
Line 178-199: By sorting out relevant domestic and foreign studies, this paper finds that most of the existing literatures focus on the unilateral relationship between digital economy development and carbon emissions based on the measurement of digital economy and carbon emissions. At present, there is still a gap in the discussion of the interaction effect of digital economy and carbon emissions reduction. It can be noted that the relationship between digital economy and carbon emissions reduction is a very topical issue. Therefore, the purpose of this study is to identify the interaction and coordination effect of digital economy and carbon emissions reduction. The application of digital technology can optimize the end treatment technology of enterprise carbon emissions [56,57], and promote carbon emission reduction [58,59]. The promotion of enterprise carbon emission reduction process brings about internal technology innovation and the strengthening of external digital supervision and management, which provide a good technical environment support for the development of digital economy. Therefore, it is necessary to study the coupling and coordination relationship between digital economy development and carbon emissions reduction. And to achieve this, it is necessary to solve the following tasks: (1) It is necessary to measure and comprehensively grasp the status and space-time characteristics of the coupling and coordination between China's digital economy development system and carbon emissions reduction system; (2) The source of the difference for the coupling and coordination between China's digital economy and carbon emissions reduction needs to be explored; (3) The spatial "spillover" effect of the coupling and coordination of digital economy development and carbon emissions reduction among cities is tested.
Point 18:
When analyzing previous studies, the authors make a number of unforced mistakes or make statements that are not supported by evidence (references). Some statements are very broad and difficult to understand. From my point of view, it is necessary to form more compact sentences, this way you avoid group references.
Response 18:
Thank you very much for your patient guidance. According to your hints, we re examined this article and found that some arguments are indeed not supported by references, especially in the introduction, which will weaken the credibility of the arguments to some extent. Therefore, in combination with your suggestions, we have checked the arguments in the introduction and added references to support them. At the same time, we also deleted group references to improve the quality of references. The specific modifications are as follows:
Line 46-49: The digital economy relies on technological innovation to continuously promote industrial integration [1] and economic restructuring, while the transformation of industrial and economic development mode will directly cause changes in carbon dioxide emissions [2].
Line 76-77: Coordination enables the two systems to work together effectively and sustainably [4,5].
Line 82-83: This reflection of the connection between the two systems is coupling [6].
Line 93-94: There is a long way to go to promote the carbon emissions reduction process [7].
Line 123-124: In addition, there are also scholars who calculate the digital economy by building a comprehensive indicator system [19-21].
Line 140-141: The exploration of carbon emissions reduction path mainly involves the improvement of carbon emissions trading market [27,28], the analysis of the policy effect of carbon emissions reduction pilot [29,30].
Added References:
- Zuo, P.; Jiang, Q.; Chen, J. Internet development, urbanization and the upgrading of China's industrial structure. Quantitative Economics and Technical Economics. 2020, 7: 71-91.
- Hosseinzadeh, B.H.; Nabavi, P. A.; Khanali, M.; Ghahderijani, M.; Chau, K. Application of data envelopment analysis ap-proach for optimization of energy use and reduction of greenhouse gas emission in peanut production of Iran. Journal of Cleaner Production. 2018,172:1327-1335.
- Kunz, N.C.; Moran, C.J.; Kastelle, T. Conceptualising “coupling” for sustainability implementation in the industrial sector: a review of the field and projection of future research opportunities. Journal of Cleaner Production. 2013, 53:69-80.
- Carl, F. Resilience: the emergence of a perspective for social-ecological systems analyses. Global Environmental Change. 2006, 16 (3):253-267.
- Solymar, L., Webb, D., Grunnet, J, A. The Physics and Applications of Photorefractive Materials. 1996, 11,Clarendon Press.
- Yan, B.; Erda, W.; Dominik, M.; Martin, L. How population migration affects carbon emissions in China: Factual and counterfactual scenario analysis. Technological Forecasting and Social Change. 2022,184: 122023.
- Xu, X.C.; Zhang, M.H. Research on the measurement of China's digital economy scale -- based on the perspective of inter-national comparison. China Industrial Economics. 2020, 5, 23-41.
- Yang, H.M.; Jiang, L. Digital economy, spatial effect and total factor productivity. Statistical Research. 2021,38,3-15.
- Shahzad, M.; Wang, J.; Dong, K.; Zhao, J. The impact of digital economy on energy transition across the globe: the mediating role of government governance. Renewable and Sustainable Energy Reviews. 2022, 166 (3) :112620.
- Feng, C.; Shi, B.B.; Kang, R. Does environmental policy reduce enterprise innovation?—evidence from China. Sustainability. 2017, 9,1-24.
- Wu, Y.Y.; Qi, J.; Xian, Q.; Chen, J.D. Research on carbon reduction effect of China's carbon market -- based on the synergistic perspective of market mechanism and administrative intervention. China Industrial Economics.2021, 8, 114-132.
- Huo, W.D.; Qi, J.; Yang, T.; Liu, J.L.; Liu, M.M.; Zhou, Z.Q. Effects of China's pilot low-carbon city policy on carbon emission reduction: A quasi-natural experiment based on satellite data. Technological Forecasting and Social Change. 2022,175: 121422.
- Liu, X.; Li, Y.C.; Chen, X.H.; Liu, J. Evaluation of low carbon city pilot policy effect on carbon abatement in China: An empirical evidence based on time-varying DID model. Cities. 2022, 123: 103582.
Point 19:
8.1.) From my point of view, lines 46-60 are redundant in the manuscript. The regulations that are not relevant to the study are listed.
Response 19:
Thank you for your suggestion. We reviewed line 46-60 again and found that the policy background of this article is mainly introduced here. We fully agree with you. The policy here is too macro and has a low relevance to the content of this article. Therefore, we have deleted this policy according to your suggestion. After deleting this part, in order to make the research background more abundant, we filled this part again, mainly focusing on the research theme of this paper to present the research background. The specific modifications are as follows:
Line 42-54: Digital economy is an economic system with technology as its core driving force, and digitalization of economic activities is a typical feature of digital economic system. the new development mode led by technological innovation driven digital economy development has laid the foundation for the coordinated promotion of digital economy and green development. The digital economy relies on technological innovation to continuously promote industrial integration [1] and economic restructuring, while the transformation of industrial and economic development mode will directly cause changes in carbon dioxide emissions [2]. At the same time, under background of " emission peak" and "carbon neutrality", the digital regulation on which the carbon emissions reduction process is based will also have a certain impact on the development of the digital economy. Therefore, it is urgent to systematically answer the interactive development relationship between digital economy and carbon emissions reduction, so that the development of digital economy can support the advancement of carbon emissions reduction.
Added References:
- Zuo, P.; Jiang, Q.; Chen, J. Internet development, urbanization and the
upgrading of China's industrial structure. Quantitative Economics and Technical Economics. 2020, 7: 71-91.
- Hosseinzadeh, B.H.; Nabavi, P. A.; Khanali, M.; Ghahderijani, M.; Chau,
- Application of data envelopment analysis ap-proach for optimization of energy use and reduction of greenhouse gas emission in peanut production of Iran. Journal of Cleaner Production. 2018,172:1327-1335.
Point 20:
8.2.) Lines 61-78. The authors try to draw parallels between the digital economy and carbon emissions reduction. But this parallel is not obvious to me. In addition, in these lines, the authors make a lot of statements unsupported by evidence (references). This shortcoming needs to be corrected.
Response 20:
Thank you for your suggestion. Since the purpose of this paper is to discuss the coordination effect of the interaction between the digital economy system and the carbon emission reduction system, it is necessary to explain the coordination and development relationship between the digital economy and carbon emission reduction, so as to clarify the internal logic of the coupling and coordination between the digital economy system and the carbon emission reduction system. At the same time, we also added new documents to support the argument here. The specific modifications are as follows:
Line 71-83: The analysis of the interaction between digital economy and carbon emissions reduction is the analysis of the coordinated development between digital economy and carbon emissions reduction. Coordinated development includes two explanations. On the one hand, from the perspective of coordination, it emphasizes the positive interaction between digital economy development and carbon emission reduction and the formation of a virtuous circle situation that promotes each other. Coordination enables the two systems to work together effectively and sustainably [4,5]. On the other hand, from the perspective of development, it emphasizes the promotion of digital economy and carbon emissions reduction. The coordinated development between digital economy and carbon emissions reduction requires that digital economy elements can help promote the process of carbon emissions reduction through technology, which in turn can continue to provide technical and environmental support for digital development. This reflection of the connection between the two systems is coupling [6].
Added References:
- Kunz, N.C.; Moran, C.J.; Kastelle, T. Conceptualising “coupling” for
sustainability implementation in the industrial sector: a review of the field and projection of future research opportunities. Journal of Cleaner Production. 2013, 53:69-80.
- Carl, F. Resilience: the emergence of a perspective for social-ecological
systems analyses. Global Environmental Change. 2006, 16 (3):253-267.
- Solymar, L., Webb, D., Grunnet, J, A. The Physics and Applications of
Photorefractive Materials. 1996, 11,Clarendon Press.
Point 21:
8.3.) Lines 83-113. The authors make a lot of statements unsupported by evidence
(references). This shortcoming needs to be corrected.
Response 21:
Thank you for your suggestion. We fully agree with your point of view. Indeed, the argument needs to be supported by references to be more convincing. Therefore, based on your suggestions, we have added relevant references to this part of the argument.
Line 71-83: The analysis of the interaction between digital economy and carbon emissions reduction is the analysis of the coordinated development between digital economy and carbon emissions reduction. Coordinated development includes two explanations. On the one hand, from the perspective of coordination, it emphasizes the positive interaction between digital economy development and carbon emission reduction and the formation of a virtuous circle situation that promotes each other. Coordination enables the two systems to work together effectively and sustainably [4,5]. On the other hand, from the perspective of development, it emphasizes the promotion of digital economy and carbon emissions reduction. The coordinated development between digital economy and carbon emissions reduction requires that digital economy elements can help promote the process of carbon emissions reduction through technology, which in turn can continue to provide technical and environmental support for digital development. This reflection of the connection between the two systems is coupling [6].
Line 93-94: There is a long way to go to promote the carbon emissions reduction process [7].
Added References:
- Kunz, N.C.; Moran, C.J.; Kastelle, T. Conceptualising “coupling” for
sustainability implementation in the industrial sector: a review of the field and projection of future research opportunities. Journal of Cleaner Production. 2013, 53:69-80.
- Carl, F. Resilience: the emergence of a perspective for social-ecological
systems analyses. Global Environmental Change. 2006, 16 (3):253-267.
- Solymar, L., Webb, D., Grunnet, J, A. The Physics and Applications of
Photorefractive Materials. 1996, 11,Clarendon Press.
- Yan, B.; Erda, W.; Dominik, M.; Martin, L. How population migration affects
carbon emissions in China: Factual and counterfactual scenario analysis. Technological Forecasting and Social Change. 2022,184: 122023.
Point 22:
8.4.) Lines 83-113. The authors make a series of repetitive statements, stating the same idea in different words. All of these things overload the manuscript and make it difficult for the reader to understand the material.
Response 22:
Thank you for your patience. The analysis of the coordinated development relationship between the digital economy system and the carbon emission reduction system is the basis for studying the interaction effect of the two systems. The purpose of this paper is to discuss the coordination effect of the interaction between the digital economy system and the carbon emission reduction system. Therefore, from the perspective of coordination and development, it is necessary to first explain the relationship between the digital economy and carbon emission reduction, so as to clarify the internal logic of the coupling and coordination between the digital economy system and the carbon emission reduction system. At the same time, in order to avoid repeated expression, we modified the content of this book, re expressed and presented it, so as to clarify the internal logical relationship between the two systems and the significance of this study. The specific modifications are as follows:
Line 71-97: The analysis of the interaction between digital economy and carbon emissions reduction is the analysis of the coordinated development between digital economy and carbon emissions reduction. Coordinated development includes two explanations. On the one hand, from the perspective of coordination, it emphasizes the positive interaction be-tween digital economy development and carbon emission reduction and the formation of a virtuous circle situation that promotes each other. Coordination enables the two systems to work together effectively and sustainably [4,5]. On the other hand, from the perspective of development, it emphasizes the promotion of digital economy and car-bon emissions reduction. The coordinated development between digital economy and carbon emissions reduction requires that digital economy elements can help promote the process of carbon emissions reduction through technology, which in turn can continue to provide technical and environmental support for digital development. This re-flection of the connection between the two systems is coupling [6].
Data shows that the market size of China's digital economy has increased from
31.3 trillion yuan in 2018 to 40.5 trillion yuan in 2020, with a growth rate of 29.39%. From the perspective of data center construction, the market size of China's data center has in-creased 195.8 billion yuan by 2020, an increase of 25.27% compared with the previous year. For the development of 5G, China has built 1.425 million 5G base stations by 2021, accounting for more than 60% of the world's total. Compared with the development of digital economy, China's carbon emissions reduction situation is not optimistic. Ac-cording to the data released by "Global Carbon Atlas", China's carbon emissions in-creased from 2421 tons in 1990 to 9751 tons in 2017, with a growth rate of 302.77%, its share in global total carbon emissions increased from 10.67% in 1990 to 27.32% in 2017. There is a long way to go to promote the carbon emissions reduction process [7]. Therefore, promoting the research on the status and trend of coordinated development of digital economy and carbon emissions reduction is of practical significance for im-proving the layout of digital economy elements and promoting green development through digitalization.
Point 23:
9.) Considering the comments (3), (4) and (8), I would like to note that the authors have
very poorly disclosed the main subject of the study. In recent years, a lot of work has been carried out to study the role of digitalization in the economy or its individual areas. Since almost any production is accompanied, which has a significant impact on the environment, the issues of Carbon Emissions Reduction are very relevant and scientists around the world are trying to minimize it. For example,
9.1.) Tcvetkov, P. Engagement of resource-based economies in the fight against rising carbon emissions. Energy Reports 2022, 8, 874-883. https://doi.org/10.1016/j.egyr.2022.05.259
9.2.) Tcvetkov, P. Climate policy imbalance in the energy sector: Time to focus on the value of CO2 utilization. Energies 2021, 14(2), 411. https://doi.org/10.3390/en14020411
Response 23:
Thank you for your suggestion. We fully agree with you. Indeed, there are a lot of researches on digital economy by scholars all over the world, and there are also many researches focusing on the field of environment, which are closely related to the research theme of this paper. Therefore, we reorganized the relevant literature, especially the international literature, to support the discussion of the existing results with more research. At the same time, we have carefully read the references you recommended. Tcvetkov (2022) mainly conducts research from the perspective of energy and believes that the development of renewable energy is of great significance to the carbon emission reduction of resource based countries. Tcvetkov (2021) assessed the existing climate policy portfolio and believed that it was necessary to adjust the existing energy technology portfolio. These documents have provided positive help for expanding international research in this field. Therefore, we cite these two articles in the references section. The specific modifications are as follows:
Line 145-149: Meanwhile, there is also study carried out from the perspective of
energy, believing that the development of renewable energy is of great significance for carbon emissions reduction in resource-based countries [34]. Another study assessed the existing climate policy portfolio and believed that the energy technology portfolio needs to be adjusted [35].
Added References:
- Tcvetkov, P. Engagement of resource-based economies in the fight against
rising carbon emissions. Energy Reports. 2022, 8: 874-883.
- Tcvetkov, P. Climate policy imbalance in the energy sector: Time to focus on
the value of CO2 utilization. Energies. 2021, 14(2): 411.
Point 24:
The use of digital technologies to analyze the current situation, formulate a problem and find the right solution is carried out in all areas of the economy. For example,
9.3) Khayrutdinov, M.M.; Golik, V.I.; Aleksakhin, A.V.; Trushina, E.V.; Lazareva,
N.V.; Aleksakhina, Y.V. Proposal of an Algorithm for Choice of a Development System for Operational and Environmental Safety in Mining. Resources 2022, 11, 88. https://doi.org/10.3390/resources11100088
9.4) Kongar-Syuryun Ch., Ubysz A., Faradzhov V. Models and algorithms of choice of
development technology of deposits when selecting the composition of the backfilling mixture. IOP Conf. Ser.: Earth Environ. Sci. 2021, 684(1), 012008. https://doi.org/10.1088/1755-1315/684/1/012008.
Response 24:
Thank you for your suggestion. Indeed, the research on the direction of digital economy is relatively rich. Therefore, we have added relevant research in the field of digital economy based on the theme of this paper to enrich the literature in the field of digital economy. We mainly supplement the literature from three perspectives: the connotation of digital economy, measurement methods, and the relationship between the development of digital economy and carbon emission reduction. In addition, thank you for recommending relevant research in the international field, which makes it possible for us to expand the boundaries of literature review and thus enhance the interest and internationality of the article. We carefully read your recommended articles Khayrutdinov et al. (2022) and Kongar er al. Therefore, we also added these two articles to the references section. The specific modifications are as follows:
Line 99-108: On the one hand, the connotation of digital economy has been intensively discussed. It is believed that the real digital economy refers to the part of economic output completely determined by digital technology, with a business framework based on digital services and goods [8]. With the integration of communication and computing technologies in the network and the flow of technology and data, it is strengthening the transformation of e-commerce and large-scale business [9]. Computer aided data flow has enabled digitization to transform various parts of the current economy [10]. Therefore, the digital economy is also divided into four different parts: digital services and goods; Mixed digital services and goods; IT based intensive production services; And IT industry [11].
Line 114-124: On the other hand, it focuses on the analysis of digital economy measurement. Most of the existing literatures measure the digital economy based on the method of summarizing the digital industry output, digital product classification, and digital related comprehensive indicators. Scholar first discussed the boundary problem of digital economy, which provides a useful reference for measuring digital economy [16]. The digital economy is also divided into five parts for measurement, namely infrastructure, e-commerce, industry structure, digital labor and digital price [17]. Some organizations have proposed an accounting framework with intelligent infrastructure investment, social promotion, innovation release, growth and employment as the main body to measure the digital economy [18]. In addition, there are also scholars who calculate the digital economy by building a comprehensive indicator system [19-21].
Line 149-151: . Some scholars discussed the environmental impact of mineral mining based on the mining industry [36,37].
Added References:
- Luyanda, D.W. Concepts of digital economy and industry 4.0 in intelligent and information systems. International Journal of Intelligent Networks. 2021,2:122-129.
- Mahmod, S. A. 5G wireless technologies- future generation communication
technologies. International Journal of Computing and Digital Systemss. 2017, 6 (3):139-147.
- Chouhan, N.; Rathore, D. Role of digitalization after demonetization in economy. International Journal of Computer Sciences and Engineering. 2018, 6(9): 88-90.
- Szeto, K. Keeping score, digitally. Music Reference Services Quarterly. 2018, 21 (2):98-100.
- Margheriol, D. H.; Cook, S.; Montes, S. The emerging digital economy. US Department of Commerce. 1998.
- Kahin, B.; Brynjolfsson, E. Understanding the digital economy. The MIT Press. 2000.
- OECD. Measuring the digital economy: an new perspective. Pairs: OECD Publishing. 2014.
- Shahzad, M.; Wang, J.; Dong, K.; Zhao, J. The impact of digital economy on energy transition across the globe: the mediating role of government governance. Renewable and Sustainable Energy Reviews. 2022, 166 (3) :112620.
- Khayrutdinov, M.M.; Golik, V.I.; Aleksakhin, A.V.; Trushina, E.V.; Lazareva, N.V.; Aleksakhina, Y.V. Proposal of an Algo-rithm for Choice of a Development System for Operational and Environmental Safety in Mining. Resources. 2022, 11: 88.
- Kongar, S.C.; Ubysz, A.; Faradzhov, V. Models and algorithms of choice of development technology of deposits when se-lecting the composition of the backfilling mixture. IOP Conference Series Earth and Environmental Science. 2021, 684(1): 012008.
Point 25:
As follows from the presented works (9.1) - (9.4) the authors of the manuscript submitted for review missed a large layer of research related to the impact of mining on the environment. If the authors become familiar with the works presented in (9.1), (9.2), (9.3), (9.4) they will be able to properly form the introduction, enrich their manuscript with international research by scientists from Poland, Czech Republic, Slovenia, Slovakia, Russia, Germany and demonstrate the depth of their material, as well as eliminate the remark (3).
Response 25:
Thank you very much for your advice. We fully agree with you. For the literature review part, it is really necessary to sort out from multiple perspectives and add as many international literature as possible to the article. We carefully read the literature you recommended and added it to the introduction. At the same time, in order to improve the richness of literature review and summarize as much as possible relevant literature in the field of digital economy and carbon emission reduction, we also supplemented about 31 papers, especially international literature, from the perspectives of the connotation and measurement methods of digital economy, the measurement and emission reduction path exploration of carbon emissions, and the relationship between the development of digital economy and carbon emission reduction. The newly added literature includes Iran, Slovenia, the United Kingdom and African countries to highlight the interesting and comprehensive nature of this discussion. The specific modifications are as follows:
Line 46-49: The digital economy relies on technological innovation to continuously promote industrial integration [1] and economic restructuring, while the transformation of industrial and economic development mode will directly cause changes in carbon dioxide emissions [2].
Line 76-77: Coordination enables the two systems to work together effectively and sustainably [4,5].
Line 82-83: This reflection of the connection between the two systems is coupling [6].
Line 93-94: There is a long way to go to promote the carbon emissions reduction process [7].
Line 100-108: It is believed that the real digital economy refers to the part of economic output completely determined by digital technology, with a business framework based on digital services and goods [8]. With the integration of communication and computing technologies in the network and the flow of technology and data, it is strengthening the transformation of e-commerce and large-scale business [9]. Computer aided data flow has enabled digitization to transform various parts of the current economy [10]. Therefore, the digital economy is also divided into four different parts: digital services and goods; Mixed digital services and goods; IT based intensive production services; And IT industry [11].
Line 114-124: Most of the existing literatures measure the digital economy based on the method of summarizing the digital industry output, digital product classification, and digital related comprehensive indicators. Scholar first discussed the boundary problem of digital economy, which provides a useful reference for measuring digital economy [16]. The digital economy is also divided into five parts for measurement, namely infrastructure, e-commerce, industry structure, digital labor and digital price [17]. Some organizations have proposed an accounting framework with intelligent infrastructure investment, social promotion, innovation release, growth and employment as the main body to measure the digital economy [18]. In addition, there are also scholars who calculate the digital economy by building a comprehensive indicator system [19-21].
Line 131-141: A study took a telecommunications company in Slovenia as the research object, calculated the greenhouse gas emissions within the enterprise scope 3, and took the purchase of electricity, the commuting of employees, the use of vehicles owned by the company and heating as the measured objects [24]. In addition, the research calculates the grid emission factors, deduces the emission factors of specific European countries, and calculates the carbon intensity based on the power generation of power plants [25]. The other is the exploration of carbon emissions reduction path. It is generally believed that economic development is closely related to carbon emissions [26]. The exploration of carbon emissions reduction path mainly involves the improvement of carbon emissions trading market [27,28], the analysis of the policy effect of carbon emissions reduction pilot [29,30]
Line 145-151: Meanwhile, there is also study carried out from the perspective of energy, believing that the development of renewable energy is of great significance for carbon emissions reduction in resource-based countries [34]. Another study assessed the existing climate policy portfolio and believed that the energy technology portfolio needs to be adjusted [35]. Some scholars discussed the environmental impact of mineral mining based on the mining industry [36,37]. The adjustment of savings also contributes to the development of green economy [38].
Line 155-163: One is the impact of digital economy development on carbon emissions. Some scholars believe that the digital economy can inhibit carbon emissions [40-42]. A study have proved the role of Taiwan's digital development in the circular economy. It is found that through digital waste management, the consumption of raw materials can be reduced by 25% by 2030, and half of greenhouse gas emissions can be avoided [43]. There is study believed that ICT can contribute to the construction of low-carbon cities [44], mainly through governance capacity of the government [45] and energy systems [46]. Study have also targeted African countries and found an inverted "U" relationship between ICT and carbon emissions in the sample countries [47].
Line 170-171: Different from this conclusion, other scholars believe that the development of digital economy is unfavorable to carbon emission reduction [53,54].
Added References:
- Zuo, P.; Jiang, Q.; Chen, J. Internet development, urbanization and the upgrading of China's industrial structure. Quantitative Economics and Technical Economics. 2020, 7: 71-91.
- Hosseinzadeh, B.H.; Nabavi, P. A.; Khanali, M.; Ghahderijani, M.; Chau, K. Application of data envelopment analysis ap-proach for optimization of energy use and reduction of greenhouse gas emission in peanut production of Iran. Journal of Cleaner Production. 2018,172:1327-1335.
- Kunz, N.C.; Moran, C.J.; Kastelle, T. Conceptualising “coupling” for sustainability implementation in the industrial sector: a review of the field and projection of future research opportunities. Journal of Cleaner Production. 2013, 53:69-80.
- Carl, F. Resilience: the emergence of a perspective for social-ecological systems analyses. Global Environmental Change. 2006, 16 (3):253-267.
- Solymar, L., Webb, D., Grunnet, J, A. The Physics and Applications of Photorefractive Materials. 1996, 11,Clarendon Press.
- Yan, B.; Erda, W.; Dominik, M.; Martin, L. How population migration affects carbon emissions in China: Factual and coun-terfactual scenario analysis. Technological Forecasting and Social Change. 2022,184: 122023
- Luyanda, D.W. Concepts of digital economy and industry 4.0 in intelligent and information systems. International Journal of Intelligent Networks. 2021,2:122-129.
- Mahmod, S. A. 5G wireless technologies- future generation communication technologies. International Journal of Computing and Digital Systemss. 2017, 6 (3):139-147.
- Chouhan, N.; Rathore, D. Role of digitalization after demonetization in economy. International Journal of Computer Sciences and Engineering. 2018, 6(9): 88-90.
- Szeto, K. Keeping score, digitally. Music Reference Services Quarterly. 2018, 21 (2):98-100.
- Margheriol, D. H.; Cook, S.; Montes, S. The emerging digital economy. US Department of Commerce. 1998.
- Kahin, B.; Brynjolfsson, E. Understanding the digital economy. The MIT Press. 2000.
- OECD. Measuring the digital economy: an new perspective. Pairs: OECD Publishing. 2014.
- Shahzad, M.; Wang, J.; Dong, K.; Zhao, J. The impact of digital economy on energy transition across the globe: the mediating role of government governance. Renewable and Sustainable Energy Reviews. 2022, 166 (3) :112620.
- Gregor, R.; Saša, T. Carbon footprint calculation in telecommunications companies – The importance and relevance of scope 3 greenhouse gases emissions. Renewable and Sustainable Energy Reviews. 2018, 98: 361-375.
- Jan, F.U.; Anke, W.; Leonhard, G.; Mirko, S. Open-data based carbon emission intensity signals for electricity generation in European countries – top down vs. bottom up approach. Cleaner Energy Systems. 2022, 3: 100018
- Batrancea, I.; Rathnaswamy, M.K.; Batrancea, L.; Nichita, A.; Gaban, L.; Fatacean, G.; Tulai, H.; Bircea, I.; Rus, M. I. A panel data analysis on sustainable economic growth in India, Brazil, and Romania. Journal of Risk and Financial Management. 2020, 13(8): 170.
- Huo, W.D.; Qi, J.; Yang, T.; Liu, J.L.; Liu, M.M.; Zhou, Z.Q. Effects of China's pilot low-carbon city policy on carbon emission reduction: A quasi-natural experiment based on satellite data. Technological Forecasting and Social Change. 2022,175: 121422.
- Liu, X.; Li, Y.C.; Chen, X.H.; Liu, J. Evaluation of low carbon city pilot policy effect on carbon abatement in China: An em-pirical evidence based on time-varying DID model. Cities. 2022, 123: 103582.
- Tcvetkov, P. Engagement of resource-based economies in the fight against rising carbon emissions. Energy Reports. 2022, 8: 874-883.
- Tcvetkov, P. Climate policy imbalance in the energy sector: Time to focus on the value of CO2 utilization. Energies. 2021, 14(2): 411.
- Khayrutdinov, M.M.; Golik, V.I.; Aleksakhin, A.V.; Trushina, E.V.; Lazareva, N.V.; Aleksakhina, Y.V. Proposal of an Algo-rithm for Choice of a Development System for Operational and Environmental Safety in Mining. Resources. 2022, 11: 88.
- Kongar, S.C.; Ubysz, A.; Faradzhov, V. Models and algorithms of choice of development technology of deposits when se-lecting the composition of the backfilling mixture. IOP Conference Series Earth and Environmental Science. 2021, 684(1): 012008.
- Batrancea, L., Rathnaswamy, M.M., Batrancea, I., Nichita, A., Rus, M.-I., Tulai, H., Fatacean, G., Masca, E.S., Morar, I.D. Adjusted net savings of CEE and Baltic nations in the context of sustainable economic growth: A panel data analysis. Journal of Risk and Financial Management. 2020, 13(10): 234.
- Tonni, A.K.; Aleksandra, M.; Marina, K.; Elena,B.; Mohd, H.D.O.; Hui, H.G. Accelerating sustainability transition in St. Petersburg (Russia) through digitalization-based circular economy in waste recycling industry: A strategy to promote carbon neutrality in era of Industry 4.0. Journal of Cleaner Production. 2022,363: 132452
- Jacob, P. Information and communication technology in shaping urban low carbon development pathways.Current Opinion in Environmental Sustainability. 2018, 30:133-137
- Muhammad, S.; Wang, J.D.; Dong, K.Y.; Zhao, J. The impact of digital economy on energy transition across the globe: The mediating role of government governance. Renewable and Sustainable Energy Reviews. 2022, 166:112620,
- Moyer, J.D.; Hughes, B.B. ICTs: Do they contribute to increased carbon emissions?. Technological Forecasting & Social Change. 2012, 79 (5) :919-931
- Asongu, S.A.; Roux, S. L.; Biekpe, N. Enhancing ICT for environmental sustainability in sub-Saharan Africa. Technological Forecasting & Social Change. 2018, 127: 209-21.
- Ilhan, O.; Sana, U. Does digital financial inclusion matter for economic growth and environmental sustainability in OBRI economies? An empirical analysis. Resources, Conservation and Recycling. 2022, 185:106489.
- Salahuddin, M.; Alam, K. Internet usage, electricity consumption and economic growth in Australia: a time series evidence. Telematics and Informatics. 2015, 32 (4): 862-878.
Point 26:
It is necessary to indicate who made figure 1. If this is the author's merit, then it is necessary to indicate: done by the authors; if this is a borrowed drawing, then it is necessary to indicate the source.
Response 26:
Thank you very much for your guidance. We fully agree that the image source you said needs to be clarified. In fact, Figure 1 is made by the author, mainly referring to the definition of digital economy in Bukht and Heeks (2019), and has been innovated and improved on the basis. At your prompt, we have explained the source of Figure 1. The specific modifications are as follows:
Line 237: Source: Figure 1 made by the author.
Reference:
Bukht, R.; Heeks, R. Defining, conceptualising and measuring the digital economy. Development Informatics Working Paper. 2019.
Point 27:
Conclusion section is formatted incorrectly. Conclusion – brief summary of the study without repeating the wording given earlier in the manuscript. The authors abuse repetition throughout the manuscript, and Conclusion section is no exception. Such a presentation of the material reduces the ease of perception by the reader of the information presented. Some of the information provided by the authors in Conclusion section has already been reported in Materials and Methods section or is related to Results and Discussion section. This information should be placed in the relevant sections. This information is superfluous for the Conclusion. The mistake of incorrectly forming conclusions is a consequence of the incorrect presentation of the introduction noted by me in remark (7) due to the fact that when writing the introduction, the aims and objectives are not formulated. Conclusions should briefly characterize the result of the study, for example: As a result of the study
(1) the dependence of … was obtained.
(2) it was found that ...
(3) and so on.
The conclusion needs to be revised.
Response 27:
Thank you very much for your patient guidance. According to your hints, we checked the conclusion part, and we fully agree with you. The conclusion part needs to make a high summary of the research results, and avoid repetition with the text part, so as to improve readers' reading interest and understanding of the full text research results. Therefore, we agree with your opinion that this part needs to be modified. In the conclusion part, we mainly made corrections from the following three aspects. First, we restated the conclusion with new wording to avoid repetition with the language expression in the text. We have highly condensed and summarized the research results, so that readers can better grasp the research results of this paper as a whole, and logically correspond to the research content. While improving readers' reading interest, we also make the conclusion independent from the results of the text report. Secondly, we supplement the current research limitations and potential research directions in the future, so that the article is more in line with a scientific paper. Finally, we used the sentence you recommended to modify and simplify the wording of the conclusion, so that readers can grasp the level of the conclusion and express the meaning more quickly. The specific modifications are as follows:
Line 836-867: This paper measures the coupling and coordination relationship
between China's digital economy development and carbon emission reduction by using entropy weight method, building a coupling and coordination degree model, measuring Dagum Gini coefficient and Moran's I index, and analyzes the spatial-temporal features, spatial difference sources and spatial correlation of the coupling and coordination of digital economy and carbon emissions reduction in detail. Due to the limitation of data dis-closure, this paper focuses on the measurement of digital economy development from 2011 to 2018. In the long-term, this may produce certain restrictions on the identification of more significant features of the evaluation results and the capture of development stability. However, according to the research in this paper, China's digital economy is developing rapidly, which makes it possible to supplement the data in the future and improve the long-term dynamic research in this paper. Through systematic research, the research conclusions are as follows:
- The interactive promotion effect and interdependence between China's
digital economy development and carbon emissions reduction have gradually increased, which means that the coupling and coordinated has shown an increasing trend. The mutual promotion effect of the two systems between regions shows the characteristics of "the east is better than the west" and "the coast is better than the inland". The coupling and coordination degree of different provinces and city types is unbalanced. The above phenomenon reflects the mutual promotion effect of China's digital economy and carbon emissions reduction still have great potential for improvement.
- It was found that the overall difference of the coupling and coordination be-
tween China's digital economy development and carbon emissions reduction shows an expanding trend, which is reflected within and between regions. Especially among regions, the gap between developing regions and developed regions has widened significantly. And the regional difference is the main source of the overall difference of the coupling and coordination.
- On the whole, the spatial "spillover" effect of the interaction between China's
digital economy development and carbon emission reduction has been obtained. The improvement of the coupling and coordination degree of the two systems in various regions will have an indirect impact on the surrounding regions. The development of the coupling and coordination degree of the two systems has a significant spatial correlation.
Point 28:
Summary: The manuscript is a finished research work. But the corrections are needed. The chosen research topic is relevant. From my point of view, the authors failed to present their research correctly and clearly, which reduced its value and worsened the ease of perception of the material presented. From my point of view, the manuscript cannot be published in the open press without correction in accordance with my suggestions.
Response 28:
Thank you very much for your patient guidance. We carefully reviewed this article, and learned the very targeted suggestions you gave together for several rounds, aiming to improve the quality of this paper so that it can meet the international journal publishing requirements. All the authors would like to express their sincere thanks to you for your patient review and detailed theoretical guidance for the improvement of this article. It is your careful guidance and help that we have the opportunity to improve this paper, and your suggestions will help us avoid premature ideas in the future paper writing. All the suggestions you put forward have given us great inspiration for this article and the writing of future papers, and let us understand the characteristics of a scientific and interesting high-quality international article. Such valuable guidance opportunities are very difficult to meet in our daily writing, and all authors cherish them very much. Finally, all the authors sincerely thank you for your valuable time to help and support us, so that we can make continuous progress. Thank you very much!
According to your suggestions, we have carefully revised the paper, mainly including the following aspects:
First, we have redefined the topic. The new topic highly summarizes the research content of this paper, which can help readers better understand the research focus of this paper.
Second, we rewrote the summary. The revised version of the abstract includes a brief review of the existing research, the purpose of this study, the use of methods, research results and research significance. At the same time, we have referred to the word prompts you have given to simplify the language expression, so that the abstract can more clearly convey information and meet the length requirements of XX journal publication.
Third, we have added three keywords. On the premise of meeting the requirements of International Journal of Environmental Research and Public Health, our final total number of keywords is 8, mainly including research content, research methods and important reference indicators, so as to facilitate readers to better search.
Fourth, we have added new content in the introduction, including the research background, the sorting of international literature supporting the argument by new references, and the research purpose and task of this paper. We have deleted the policy introduction that is less relevant to this article. At the same time, we have supplemented the reference basis for important arguments to make the arguments more convincing. In addition, we have also added international literature from different countries, including 31 new articles recommended by you, which expands the scope of this paper's sorting of previous studies, enriches the literature review part of this paper, and increases the interest and internationality of this paper. In addition, in order to make readers better understand the problems to be solved in this article, according to your suggestions, we have added the research purpose and research task of this article at the end of the introduction, making them correspond to the content and conclusion, which helps to improve the internal logic of the article.
Fifth, we will supplement the source of the figures made in the article. For example: Figure 1 is made by the author, which can be regarded as our unique contribution. We added this description at your prompt.
Sixth, we re-present the conclusion. We re summarized the conclusions, highly refined the research results, and re expressed them on the basis of your recommended vocabulary to avoid duplication with the text. At the same time, we also add the research limitations and research directions that can be expanded in the future, so that the article has more elements that a scientific paper should have.
Finally, we invited language experts to polish this article. As this article needs to be presented in international journals, professional language expression is necessary.
Thanks very much to the reviewers and editors for your valuable comments and suggestions, so that the quality of the manuscript can be improved. If there is also some improper expression in the revised manuscript, please let us know and we will further improve it.

Reviewer 2 Report
This article thoroughly analyzes China's digital economy development and carbon emissions reduction. I think the paper is very good, and I suggest the authors to describe the motivation and shortcomings in more detail.
Author Response
Analysis on the Coupling Effect and Space-Time Difference between China's Digital Economy Development and Carbon Emissions Reduction
Response to Reviewer 2 Comments
We are grateful for the reviewer’s thoughtful comments and suggestions. We have thoroughly revised our manuscript according to the comments. Our responses (in blue) and the reviewer's original comments (in black) are given below:
(The specific modification content can also be seen in the Word version uploaded by the author)
Point 1:
English language and style: English language and style are fine/minor spell check required.
Response 1:
Thank you very much for your advice. All authors fully agree with you. We carefully checked the article and found syntax errors in the model design part, so we corrected them. In order to ensure the professionalism of the full text language, we submitted the article to the language experts for modification to avoid language errors in this article. The specific modifications made by the authors are as follows:
Line 306-309: In Formula (1), is the total carbon emissions of prefecture level cities, is the carbon emissions of natural gas consumed by industry, is the carbon emissions of liquefied petroleum gas consumed by industry and is the carbon emissions of electricity consumed by industry.
Point 2:
Is the research design appropriate?——Can be improved.
Response 2:
Thank you very much for your patient guidance. We checked the full text and agreed with your opinion that it is necessary to improve the research design. A good research design can make the article more logical and help readers better understand the content of the article. Therefore, according to your suggestions, we have made design modifications to the article. It mainly includes two aspects: one is to modify the sub title of the paper to make the content more consistent with the title. The other is to modify the content, mainly including introduction and conclusion. In the introduction, we have added new contents, including research background, new references to the support of the argument and the sorting of international literature, as well as the research purpose and task of this paper. We have deleted the policy introduction that is less relevant to this article. At the same time, it has supplemented the reference basis for the important argument to make the argument more convincing. In addition, we have also added 31 international documents from different countries, which expands the scope of this paper's sorting of previous studies, enriches the literature review part of this paper, and increases the interest and internationality of this paper. In addition, in order to make readers better understand the problems to be solved in this article, we added the research purpose and task of this article at the end of the introduction, making them correspond to the content and conclusion, which helps to improve the internal logic of the article. In the conclusion, we made a new summary, highly refined the research results, and avoided the repetition of its expression and the text. At the same time, we also add the research limitations and research directions that can be expanded in the future, so that the article has more elements that a scientific paper should have. To sum up, we have revised the title design and research content. The specific modifications are as follows:
Line 42-54: Digital economy is an economic system with technology as its core driving force, and digitalization of economic activities is a typical feature of digital economic system. the new development mode led by technological innovation driven digital economy development has laid the foundation for the coordinated promotion of digital economy and green development. The digital economy relies on technological innovation to continuously promote industrial integration [1] and economic restructuring, while the transformation of industrial and economic development mode will directly cause changes in carbon dioxide emissions [2]. At the same time, under background of " emission peak" and "carbon neutrality", the digital regulation on which the carbon emissions reduction process is based will also have a certain impact on the development of the digital economy. Therefore, it is urgent to systematically answer the interactive development relationship between digital economy and carbon emissions reduction, so that the development of digital economy can support the advancement of carbon emissions reduction.
Line 71-83: The analysis of the interaction between digital economy and carbon emissions reduction is the analysis of the coordinated development between digital economy and carbon emissions reduction. Coordinated development includes two explanations. On the one hand, from the perspective of coordination, it emphasizes the positive interaction between digital economy development and carbon emission reduction and the formation of a virtuous circle situation that promotes each other. Coordination enables the two systems to work together effectively and sustainably [4,5]. On the other hand, from the perspective of development, it emphasizes the promotion of digital economy and carbon emissions reduction. The coordinated development between digital economy and carbon emissions reduction requires that digital economy elements can help promote the process of carbon emissions reduction through technology, which in turn can continue to provide technical and environmental support for digital development. This reflection of the connection between the two systems is coupling [6].
Line 93-97: There is a long way to go to promote the carbon emissions reduction process [7]. Therefore, promoting the research on the status and trend of coordinated development of digital economy and carbon emissions reduction is of practical significance for improving the layout of digital economy elements and promoting green development through digitalization.
Line 100-108: It is believed that the real digital economy refers to the part of economic output completely determined by digital technology, with a business framework based on digital services and goods [8]. With the integration of communication and computing technologies in the network and the flow of technology and data, it is strengthening the transformation of e-commerce and large-scale business [9]. Computer aided data flow has enabled digitization to transform various parts of the current economy [10]. Therefore, the digital economy is also divided into four different parts: digital services and goods; Mixed digital services and goods; IT based intensive production services; And IT industry [11].
Line 114-124: Most of the existing literatures measure the digital economy based on the method of summarizing the digital industry output, digital product classification, and digital related comprehensive indicators. Scholar first discussed the boundary problem of digital economy, which provides a useful reference for measuring digital economy [16]. The digital economy is also divided into five parts for measurement, namely infrastructure, e-commerce, industry structure, digital labor and digital price [17]. Some organizations have proposed an accounting framework with intelligent infrastructure investment, social promotion, innovation release, growth and employment as the main body to measure the digital economy [18]. In addition, there are also scholars who calculate the digital economy by building a comprehensive indicator system [19-21].
Line 131-141: A study took a telecommunications company in Slovenia as the research object, calculated the greenhouse gas emissions within the enterprise scope 3, and took the purchase of electricity, the commuting of employees, the use of vehicles owned by the company and heating as the measured objects [24]. In addition, the research calculates the grid emission factors, deduces the emission factors of specific European countries, and calculates the carbon intensity based on the power generation of power plants [25]. The other is the exploration of carbon emissions reduction path. It is generally believed that economic development is closely related to carbon emissions [26]. The exploration of carbon emissions reduction path mainly involves the improvement of carbon emissions trading market [27,28], the analysis of the policy effect of carbon emissions reduction pilot [29,30].
Line 145-152: Meanwhile, there is also study carried out from the perspective of energy, believing that the development of renewable energy is of great significance for carbon emissions reduction in resource-based countries [34]. Another study assessed the existing climate policy portfolio and believed that the energy technology portfolio needs to be adjusted [35]. Some scholars discussed the environmental impact of mineral mining based on the mining industry [36,37]. The adjustment of savings also contributes to the development of green economy [38].
Line 155-166: One is the impact of digital economy development on carbon emissions. Some scholars believe that the digital economy can inhibit carbon emissions [40-42]. A study have proved the role of Taiwan's digital development in the circular economy. It is found that through digital waste management, the consumption of raw materials can be reduced by 25% by 2030, and half of greenhouse gas emissions can be avoided [43]. There is study believed that ICT can contribute to the construction of low-carbon cities [44], mainly through governance capacity of the government [45] and energy systems [46]. Study have also targeted African countries and found an inverted "U" relationship between ICT and carbon emissions in the sample countries [47]. In addition, some scholars believe that R&D investment played a positive role in regulating the relationship between digital economy development and carbon emissions reduction [48-50].
Line 170-171: Different from this conclusion, other scholars believe that the development of digital economy is unfavorable to carbon emission reduction [53,54].
Line 178-185: By sorting out relevant domestic and foreign studies, this paper finds that most of the existing literatures focus on the unilateral relationship between digital economy development and carbon emissions based on the measurement of digital economy and carbon emissions. At present, there is still a gap in the discussion of the interaction effect of digital economy and carbon emissions reduction. It can be noted that the relationship between digital economy and carbon emissions reduction is a very topical issue. Therefore, the purpose of this study is to identify the interaction and coordination effect of digital economy and carbon emissions reduction.
Line 190-199: Therefore, it is necessary to study the coupling and coordination relationship between digital economy development and carbon emissions reduction. And to achieve this, it is necessary to solve the following tasks: (1) It is necessary to measure and comprehensively grasp the status and space-time characteristics of the coupling and coordination between China's digital economy development system and carbon emissions reduction system; (2) The source of the difference for the coupling and coordination between China's digital economy and carbon emissions reduction needs to be explored; (3) The spatial "spillover" effect of the coupling and coordination of digital economy development and carbon emissions reduction among cities is tested.
Line 450: 3.1. Calculation of the Coupling and Coordination Degree of Two Systems.
Line 492: 3.2. Calculation of the Coupling and Coordination Degree of Two Systems in Eight Regions.
Line 574: 3.3. Calculation of the Coupling and Coordination Degree of Two Systems in Each Province.
Line 652: 3.4. Calculation of the Coupling and Coordination Degree of Two Systems in Different Types of Cities.
Added References:
- Zuo, P.; Jiang, Q.; Chen, J. Internet development, urbanization and the upgrading of China's industrial structure. Quantitative Economics and Technical Economics. 2020, 7: 71-91.
- Hosseinzadeh, B.H.; Nabavi, P. A.; Khanali, M.; Ghahderijani, M.; Chau, K. Application of data envelopment analysis ap-proach for optimization of energy use and reduction of greenhouse gas emission in peanut production of Iran. Journal of Cleaner Production. 2018,172:1327-1335.
- Kunz, N.C.; Moran, C.J.; Kastelle, T. Conceptualising “coupling” for sustainability implementation in the industrial sector: a review of the field and projection of future research opportunities. Journal of Cleaner Production. 2013, 53:69-80.
- Carl, F. Resilience: the emergence of a perspective for social-ecological systems analyses. Global Environmental Change. 2006, 16 (3):253-267.
- Solymar, L., Webb, D., Grunnet, J, A. The Physics and Applications of Photorefractive Materials. 1996, 11,Clarendon Press.
- Yan, B.; Erda, W.; Dominik, M.; Martin, L. How population migration affects carbon emissions in China: Factual and coun-terfactual scenario analysis. Technological Forecasting and Social Change. 2022,184: 122023
- Luyanda, D.W. Concepts of digital economy and industry 4.0 in intelligent and information systems. International Journal of Intelligent Networks. 2021,2:122-129.
- Mahmod, S. A. 5G wireless technologies- future generation communication technologies. International Journal of Computing and Digital Systemss. 2017, 6 (3):139-147.
- Chouhan, N.; Rathore, D. Role of digitalization after demonetization in economy. International Journal of Computer Sciences and Engineering. 2018, 6(9): 88-90.
- Szeto, K. Keeping score, digitally. Music Reference Services Quarterly. 2018, 21 (2):98-100.
- Margheriol, D. H.; Cook, S.; Montes, S. The emerging digital economy. US Department of Commerce. 1998.
- Kahin, B.; Brynjolfsson, E. Understanding the digital economy. The MIT Press. 2000.
- OECD. Measuring the digital economy: an new perspective. Pairs: OECD Publishing. 2014.
- Shahzad, M.; Wang, J.; Dong, K.; Zhao, J. The impact of digital economy on energy transition across the globe: the mediating role of government governance. Renewable and Sustainable Energy Reviews. 2022, 166 (3) :112620.
- Gregor, R.; Saša, T. Carbon footprint calculation in telecommunications companies – The importance and relevance of scope 3 greenhouse gases emissions. Renewable and Sustainable Energy Reviews. 2018, 98: 361-375.
- Jan, F.U.; Anke, W.; Leonhard, G.; Mirko, S. Open-data based carbon emission intensity signals for electricity generation in European countries – top down vs. bottom up approach. Cleaner Energy Systems. 2022, 3: 100018
- Batrancea, I.; Rathnaswamy, M.K.; Batrancea, L.; Nichita, A.; Gaban, L.; Fatacean, G.; Tulai, H.; Bircea, I.; Rus, M. I. A panel data analysis on sustainable economic growth in India, Brazil, and Romania. Journal of Risk and Financial Management. 2020, 13(8): 170.
- Huo, W.D.; Qi, J.; Yang, T.; Liu, J.L.; Liu, M.M.; Zhou, Z.Q. Effects of China's pilot low-carbon city policy on carbon emission reduction: A quasi-natural experiment based on satellite data. Technological Forecasting and Social Change. 2022,175: 121422.
- Liu, X.; Li, Y.C.; Chen, X.H.; Liu, J. Evaluation of low carbon city pilot policy effect on carbon abatement in China: An em-pirical evidence based on time-varying DID model. Cities. 2022, 123: 103582.
- Tcvetkov, P. Engagement of resource-based economies in the fight against rising carbon emissions. Energy Reports. 2022, 8: 874-883.
- Tcvetkov, P. Climate policy imbalance in the energy sector: Time to focus on the value of CO2 utilization. Energies. 2021, 14(2): 411.
- Khayrutdinov, M.M.; Golik, V.I.; Aleksakhin, A.V.; Trushina, E.V.; Lazareva, N.V.; Aleksakhina, Y.V. Proposal of an Algo-rithm for Choice of a Development System for Operational and Environmental Safety in Mining. Resources. 2022, 11: 88.
- Kongar, S.C.; Ubysz, A.; Faradzhov, V. Models and algorithms of choice of development technology of deposits when se-lecting the composition of the backfilling mixture. IOP Conference Series Earth and Environmental Science. 2021, 684(1): 012008.
- Batrancea, L., Rathnaswamy, M.M., Batrancea, I., Nichita, A., Rus, M.-I., Tulai, H., Fatacean, G., Masca, E.S., Morar, I.D. Adjusted net savings of CEE and Baltic nations in the context of sustainable economic growth: A panel data analysis. Journal of Risk and Financial Management. 2020, 13(10): 234.
- Tonni, A.K.; Aleksandra, M.; Marina, K.; Elena,B.; Mohd, H.D.O.; Hui, H.G. Accelerating sustainability transition in St. Petersburg (Russia) through digitalization-based circular economy in waste recycling industry: A strategy to promote carbon neutrality in era of Industry 4.0. Journal of Cleaner Production. 2022,363: 132452
- Jacob, P. Information and communication technology in shaping urban low carbon development pathways.Current Opinion in Environmental Sustainability. 2018, 30:133-137
- Muhammad, S.; Wang, J.D.; Dong, K.Y.; Zhao, J. The impact of digital economy on energy transition across the globe: The mediating role of government governance. Renewable and Sustainable Energy Reviews. 2022, 166:112620,
- Moyer, J.D.; Hughes, B.B. ICTs: Do they contribute to increased carbon emissions?. Technological Forecasting & Social Change. 2012, 79 (5) :919-931
- Asongu, S.A.; Roux, S. L.; Biekpe, N. Enhancing ICT for environmental sustainability in sub-Saharan Africa. Technological Forecasting & Social Change. 2018, 127: 209-21.
- Ilhan, O.; Sana, U. Does digital financial inclusion matter for economic growth and environmental sustainability in OBRI economies? An empirical analysis. Resources, Conservation and Recycling. 2022, 185:106489.
- Salahuddin, M.; Alam, K. Internet usage, electricity consumption and economic growth in Australia: a time series evidence. Telematics and Informatics. 2015, 32 (4): 862-878.
Point 3:
Comments and Suggestions for Authors:This article thoroughly analyzes China's digital economy development and carbon emissions reduction. I think the paper is very good, and I suggest the authors to describe the motivation and shortcomings in more detail.
Response 3:
First, thank you very much for your affirmation of our article. It is your patient review that has made it possible to improve the quality of our article, which has greatly helped to improve our research and inspire future paper writing. For your suggestions, all authors will carefully and carefully read them and try their best to correct them to improve the quality of the manuscript. Thanks again for your help and support!
Second, in view of your motivation and shortcomings, we agree that it is really necessary to clarify the research purpose and current research limitations of this paper, which is an essential element of a scientific paper. Therefore, based on your suggestions, we have supplemented the research purpose and task of this paper at the end of the introduction. At the same time, in the conclusion part, we also added the research limitations and research directions that can be expanded in the future to improve the standardization and integrity of this article. The specific modifications are as follows:
Line 178-185: By sorting out relevant domestic and foreign studies, this paper finds that most of the existing literatures focus on the unilateral relationship between digital economy development and carbon emissions based on the measurement of digital economy and carbon emissions. At present, there is still a gap in the discussion of the interaction effect of digital economy and carbon emissions reduction. It can be noted that the relationship between digital economy and carbon emissions reduction is a very topical issue. Therefore, the purpose of this study is to identify the interaction and coordination effect of digital economy and carbon emissions reduction.
Line 190-199: Therefore, it is necessary to study the coupling and coordination relationship between digital economy development and carbon emissions reduction. And to achieve this, it is necessary to solve the following tasks: (1) It is necessary to measure and comprehensively grasp the status and space-time characteristics of the coupling and coordination between China's digital economy development system and carbon emissions reduction system; (2) The source of the difference for the coupling and coordination between China's digital economy and carbon emissions reduction needs to be explored; (3) The spatial "spillover" effect of the coupling and coordination of digital economy development and carbon emissions reduction among cities is tested.
Line 835-846: This paper measures the coupling and coordination relationship between China's digital economy development and carbon emission reduction by using entropy weight method, building a coupling and coordination degree model, measuring Dagum Gini coefficient and Moran's I index, and analyzes the spatial-temporal features, spatial difference sources and spatial correlation of the coupling and coordination of digital economy and carbon emissions reduction in detail. Due to the limitation of data disclosure, this paper focuses on the measurement of digital economy development from 2011 to 2018. In the long-term, this may produce certain restrictions on the identification of more significant features of the evaluation results and the capture of development stability. However, according to the research in this paper, China's digital economy is developing rapidly, which makes it possible to supplement the data in the future and improve the long-term dynamic research in this paper.
Thank you very much for your patient guidance. We carefully reviewed this article, and learned the very targeted suggestions you gave together for several rounds, aiming to improve the quality of this paper so that it can meet the international journal publishing requirements. All the authors would like to express their sincere thanks to you for your patient review and detailed theoretical guidance for the improvement of this article. It is your careful guidance and help that we have the opportunity to improve this paper, and your suggestions will help us avoid premature ideas in the future paper writing. All the suggestions you put forward have given us great inspiration for this article and the writing of future papers, and let us understand the characteristics of a scientific and interesting high-quality international article. Such valuable guidance opportunities are very difficult to meet in our daily writing, and all authors cherish them very much. Finally, all the authors sincerely thank you for your valuable time to help and support us, so that we can make continuous progress. Thank you very much!
Thanks very much to the reviewers and editors for your valuable comments and suggestions, so that the quality of the manuscript can be improved. If there is also some improper expression in the revised manuscript, please let us know and we will further improve it.

Reviewer 3 Report
The interaction between digital economy and carbon emissions is a frontier issue. This study uses econometric models to explore the coupling and co scheduling of China's digital economy system and carbon emission reduction system over the years, and obtains some valuable empirical results. My suggestions are as follows.
1、Figure 1 should be optimized.
2、It is suggested that the author refine the most valuable findings.
3、Misuse of two reference methods. It is suggested that the author choose one of the two reference methods, either name plus year or serial number.
4、some subtitles are too miscellaneous, it is recommended that the author refine the title to avoid repetition.
5、Table 7 is too redundant and unclear.
6、There are too many self-references.
Author Response
Analysis on the Coupling Effect and Space-Time Difference between China's Digital Economy Development and Carbon Emissions Reduction
Response to Reviewer 3 Comments
We are grateful for the reviewer’s thoughtful comments and suggestions. We have thoroughly revised our manuscript according to the comments. Our responses (in blue) and the reviewer's original comments (in black) are given below:
(The specific modification content can also be seen in the word version uploaded by the author)
Point 1:
English language and style: Extensive editing of English language and style required.
Response 1:
Thank you very much for your advice. All authors fully agree with you. We carefully checked the article and found syntax errors in the model design part, so we corrected them. In order to ensure the professionalism of the full text language, we submitted the article to the language experts for modification to avoid language errors in this article. The specific modifications made by the authors are as follows:
Line 306-309: In Formula (1), is the total carbon emissions of prefecture level cities, is the carbon emissions of natural gas consumed by industry, is the carbon emissions of liquefied petroleum gas consumed by industry and is the carbon emissions of electricity consumed by industry.
Point 2:
Are all the cited references relevant to the research?——Must be improved.
Response 2:
Thank you very much for your patient guidance. We checked the citation of references and found that the citation of some references was indeed unnecessary. Therefore, we fully agree with your point of view, and need to modify the references. We mainly made the following three changes: First, we added references to the important arguments in the introduction to support them, so as to improve the scientific expression of the arguments. In the process of adding, we mainly choose to add relevant documents in the field of digital economy and environment represented by carbon emission reduction to ensure that the references are highly consistent with the theme of this paper. Secondly, we have added 31 relevant articles, especially international articles. After re searching, we found that there have been a lot of research on digital economy and carbon emission reduction in the world. Therefore, we added more literature in this field to sort out the existing research under this theme as completely as possible. Finally, we deleted marginal literature, mainly focusing on the positive impact of digital economy on carbon emission reduction. The specific modifications are as follows:
Line 42-54: Digital economy is an economic system with technology as its core driving force, and digitalization of economic activities is a typical feature of digital economic system. the new development mode led by technological innovation driven digital economy development has laid the foundation for the coordinated promotion of digital economy and green development. The digital economy relies on technological innovation to continuously promote industrial integration [1] and economic restructuring, while the transformation of industrial and economic development mode will directly cause changes in carbon dioxide emissions [2]. At the same time, under background of " emission peak" and "carbon neutrality", the digital regulation on which the carbon emissions reduction process is based will also have a certain impact on the development of the digital economy. Therefore, it is urgent to systematically answer the interactive development relationship between digital economy and carbon emissions reduction, so that the development of digital economy can support the advancement of carbon emissions reduction.
Line 71-83: The analysis of the interaction between digital economy and carbon emissions reduction is the analysis of the coordinated development between digital economy and carbon emissions reduction. Coordinated development includes two explanations. On the one hand, from the perspective of coordination, it emphasizes the positive interaction between digital economy development and carbon emission reduction and the formation of a virtuous circle situation that promotes each other. Coordination enables the two systems to work together effectively and sustainably [4,5]. On the other hand, from the perspective of development, it emphasizes the promotion of digital economy and carbon emissions reduction. The coordinated development between digital economy and carbon emissions reduction requires that digital economy elements can help promote the process of carbon emissions reduction through technology, which in turn can continue to provide technical and environmental support for digital development. This reflection of the connection between the two systems is coupling [6].
Line 93-97: There is a long way to go to promote the carbon emissions reduction process [7]. Therefore, promoting the research on the status and trend of coordinated development of digital economy and carbon emissions reduction is of practical significance for improving the layout of digital economy elements and promoting green development through digitalization.
Line 100-108: It is believed that the real digital economy refers to the part of economic output completely determined by digital technology, with a business framework based on digital services and goods [8]. With the integration of communication and computing technologies in the network and the flow of technology and data, it is strengthening the transformation of e-commerce and large-scale business [9]. Computer aided data flow has enabled digitization to transform various parts of the current economy [10]. Therefore, the digital economy is also divided into four different parts: digital services and goods; Mixed digital services and goods; IT based intensive production services; And IT industry [11].
Line 114-124: Most of the existing literatures measure the digital economy based on the method of summarizing the digital industry output, digital product classification, and digital related comprehensive indicators. Scholar first discussed the boundary problem of digital economy, which provides a useful reference for measuring digital economy [16]. The digital economy is also divided into five parts for measurement, namely infrastructure, e-commerce, industry structure, digital labor and digital price [17]. Some organizations have proposed an accounting framework with intelligent infrastructure investment, social promotion, innovation release, growth and employment as the main body to measure the digital economy [18]. In addition, there are also scholars who calculate the digital economy by building a comprehensive indicator system [19-21].
Line 131-141: A study took a telecommunications company in Slovenia as the research object, calculated the greenhouse gas emissions within the enterprise scope 3, and took the purchase of electricity, the commuting of employees, the use of vehicles owned by the company and heating as the measured objects [24]. In addition, the research calculates the grid emission factors, deduces the emission factors of specific European countries, and calculates the carbon intensity based on the power generation of power plants [25]. The other is the exploration of carbon emissions reduction path. It is generally believed that economic development is closely related to carbon emissions [26]. The exploration of carbon emissions reduction path mainly involves the improvement of carbon emissions trading market [27,28], the analysis of the policy effect of carbon emissions reduction pilot [29,30].
Line 145-152: Meanwhile, there is also study carried out from the perspective of energy, believing that the development of renewable energy is of great significance for carbon emissions reduction in resource-based countries [34]. Another study assessed the existing climate policy portfolio and believed that the energy technology portfolio needs to be adjusted [35]. Some scholars discussed the environmental impact of mineral mining based on the mining industry [36,37]. The adjustment of savings also contributes to the development of green economy [38].
Line 155-166: One is the impact of digital economy development on carbon emissions. Some scholars believe that the digital economy can inhibit carbon emissions [40-42]. A study have proved the role of Taiwan's digital development in the circular economy. It is found that through digital waste management, the consumption of raw materials can be reduced by 25% by 2030, and half of greenhouse gas emissions can be avoided [43]. There is study believed that ICT can contribute to the construction of low-carbon cities [44], mainly through governance capacity of the government [45] and energy systems [46]. Study have also targeted African countries and found an inverted "U" relationship between ICT and carbon emissions in the sample countries [47]. In addition, some scholars believe that R&D investment played a positive role in regulating the relationship between digital economy development and carbon emissions reduction [48-50].
Line 170-171: Different from this conclusion, other scholars believe that the development of digital economy is unfavorable to carbon emission reduction [53,54].
Line 178-185: By sorting out relevant domestic and foreign studies, this paper finds that most of the existing literatures focus on the unilateral relationship between digital economy development and carbon emissions based on the measurement of digital economy and carbon emissions. At present, there is still a gap in the discussion of the interaction effect of digital economy and carbon emissions reduction. It can be noted that the relationship between digital economy and carbon emissions reduction is a very topical issue. Therefore, the purpose of this study is to identify the interaction and coordination effect of digital economy and carbon emissions reduction.
Line 190-199: Therefore, it is necessary to study the coupling and coordination relationship between digital economy development and carbon emissions reduction. And to achieve this, it is necessary to solve the following tasks: (1) It is necessary to measure and comprehensively grasp the status and space-time characteristics of the coupling and coordination between China's digital economy development system and carbon emissions reduction system; (2) The source of the difference for the coupling and coordination between China's digital economy and carbon emissions reduction needs to be explored; (3) The spatial "spillover" effect of the coupling and coordination of digital economy development and carbon emissions reduction among cities is tested.
Line 450: 3.1. Calculation of the Coupling and Coordination Degree of Two Systems.
Line 492: 3.2. Calculation of the Coupling and Coordination Degree of Two Systems in Eight Regions.
Line 574: 3.3. Calculation of the Coupling and Coordination Degree of Two Systems in Each Province.
Line 652: 3.4. Calculation of the Coupling and Coordination Degree of Two Systems in Different Types of Cities.
Added References:
- Zuo, P.; Jiang, Q.; Chen, J. Internet development, urbanization and the upgrading of China's industrial structure. Quantitative Economics and Technical Economics. 2020, 7: 71-91.
- Hosseinzadeh, B.H.; Nabavi, P. A.; Khanali, M.; Ghahderijani, M.; Chau, K. Application of data envelopment analysis ap-proach for optimization of energy use and reduction of greenhouse gas emission in peanut production of Iran. Journal of Cleaner Production. 2018,172:1327-1335.
- Kunz, N.C.; Moran, C.J.; Kastelle, T. Conceptualising “coupling” for sustainability implementation in the industrial sector: a review of the field and projection of future research opportunities. Journal of Cleaner Production. 2013, 53:69-80.
- Carl, F. Resilience: the emergence of a perspective for social-ecological systems analyses. Global Environmental Change. 2006, 16 (3):253-267.
- Solymar, L., Webb, D., Grunnet, J, A. The Physics and Applications of Photorefractive Materials. 1996, 11,Clarendon Press.
- Yan, B.; Erda, W.; Dominik, M.; Martin, L. How population migration affects carbon emissions in China: Factual and coun-terfactual scenario analysis. Technological Forecasting and Social Change. 2022,184: 122023
- Luyanda, D.W. Concepts of digital economy and industry 4.0 in intelligent and information systems. International Journal of Intelligent Networks. 2021,2:122-129.
- Mahmod, S. A. 5G wireless technologies- future generation communication technologies. International Journal of Computing and Digital Systemss. 2017, 6 (3):139-147.
- Chouhan, N.; Rathore, D. Role of digitalization after demonetization in economy. International Journal of Computer Sciences and Engineering. 2018, 6(9): 88-90.
- Szeto, K. Keeping score, digitally. Music Reference Services Quarterly. 2018, 21 (2):98-100.
- Margheriol, D. H.; Cook, S.; Montes, S. The emerging digital economy. US Department of Commerce. 1998.
- Kahin, B.; Brynjolfsson, E. Understanding the digital economy. The MIT Press. 2000.
- OECD. Measuring the digital economy: an new perspective. Pairs: OECD Publishing. 2014.
- Shahzad, M.; Wang, J.; Dong, K.; Zhao, J. The impact of digital economy on energy transition across the globe: the mediating role of government governance. Renewable and Sustainable Energy Reviews. 2022, 166 (3) :112620.
- Gregor, R.; Saša, T. Carbon footprint calculation in telecommunications companies – The importance and relevance of scope 3 greenhouse gases emissions. Renewable and Sustainable Energy Reviews. 2018, 98: 361-375.
- Jan, F.U.; Anke, W.; Leonhard, G.; Mirko, S. Open-data based carbon emission intensity signals for electricity generation in European countries – top down vs. bottom up approach. Cleaner Energy Systems. 2022, 3: 100018
- Batrancea, I.; Rathnaswamy, M.K.; Batrancea, L.; Nichita, A.; Gaban, L.; Fatacean, G.; Tulai, H.; Bircea, I.; Rus, M. I. A panel data analysis on sustainable economic growth in India, Brazil, and Romania. Journal of Risk and Financial Management. 2020, 13(8): 170.
- Huo, W.D.; Qi, J.; Yang, T.; Liu, J.L.; Liu, M.M.; Zhou, Z.Q. Effects of China's pilot low-carbon city policy on carbon emission reduction: A quasi-natural experiment based on satellite data. Technological Forecasting and Social Change. 2022,175: 121422.
- Liu, X.; Li, Y.C.; Chen, X.H.; Liu, J. Evaluation of low carbon city pilot policy effect on carbon abatement in China: An em-pirical evidence based on time-varying DID model. Cities. 2022, 123: 103582.
- Tcvetkov, P. Engagement of resource-based economies in the fight against rising carbon emissions. Energy Reports. 2022, 8: 874-883.
- Tcvetkov, P. Climate policy imbalance in the energy sector: Time to focus on the value of CO2 utilization. Energies. 2021, 14(2): 411.
- Khayrutdinov, M.M.; Golik, V.I.; Aleksakhin, A.V.; Trushina, E.V.; Lazareva, N.V.; Aleksakhina, Y.V. Proposal of an Algo-rithm for Choice of a Development System for Operational and Environmental Safety in Mining. Resources. 2022, 11: 88.
- Kongar, S.C.; Ubysz, A.; Faradzhov, V. Models and algorithms of choice of development technology of deposits when se-lecting the composition of the backfilling mixture. IOP Conference Series Earth and Environmental Science. 2021, 684(1): 012008.
- Batrancea, L., Rathnaswamy, M.M., Batrancea, I., Nichita, A., Rus, M.-I., Tulai, H., Fatacean, G., Masca, E.S., Morar, I.D. Adjusted net savings of CEE and Baltic nations in the context of sustainable economic growth: A panel data analysis. Journal of Risk and Financial Management. 2020, 13(10): 234.
- Tonni, A.K.; Aleksandra, M.; Marina, K.; Elena,B.; Mohd, H.D.O.; Hui, H.G. Accelerating sustainability transition in St. Petersburg (Russia) through digitalization-based circular economy in waste recycling industry: A strategy to promote carbon neutrality in era of Industry 4.0. Journal of Cleaner Production. 2022,363: 132452
- Jacob, P. Information and communication technology in shaping urban low carbon development pathways.Current Opinion in Environmental Sustainability. 2018, 30:133-137
- Muhammad, S.; Wang, J.D.; Dong, K.Y.; Zhao, J. The impact of digital economy on energy transition across the globe: The mediating role of government governance. Renewable and Sustainable Energy Reviews. 2022, 166:112620,
- Moyer, J.D.; Hughes, B.B. ICTs: Do they contribute to increased carbon emissions?. Technological Forecasting & Social Change. 2012, 79 (5) :919-931
- Asongu, S.A.; Roux, S. L.; Biekpe, N. Enhancing ICT for environmental sustainability in sub-Saharan Africa. Technological Forecasting & Social Change. 2018, 127: 209-21.
- Ilhan, O.; Sana, U. Does digital financial inclusion matter for economic growth and environmental sustainability in OBRI economies? An empirical analysis. Resources, Conservation and Recycling. 2022, 185:106489.
- Salahuddin, M.; Alam, K. Internet usage, electricity consumption and economic growth in Australia: a time series evidence. Telematics and Informatics. 2015, 32 (4): 862-878.
Point 3:
Are the conclusions supported by the results?——Can be improved.
Response 3:
Thank you for your patient review and suggestions. All the authors checked the correspondence between the conclusion and the research results, and found that the conclusion part can also present the research content more concisely, so as to avoid duplication of expression with the research content part and improve the reading interest of readers. In addition, in order to make the whole article more enlightening, we also add the limitations of this study and the areas that can be broken through in the future in the conclusion part. Therefore, according to your suggestions and in combination with the research content of the article, we restated the conclusion. The specific modifications are as follows:
Line 840-866: Due to the limitation of data disclosure, this paper focuses on
the measurement of digital economy development from 2011 to 2018. In the long-term, this may produce certain restrictions on the identification of more significant features of the evaluation results and the capture of development stability. However, according to the research in this paper, China's digital economy is developing rapidly, which makes it possible to supplement the data in the future and improve the long-term dynamic research in this paper. Through systematic research, the research conclusions are as follows:
- The interactive promotion effect and interdependence between China's digital
economy development and carbon emissions reduction have gradually increased, which means that the coupling and coordinated has shown an increasing trend. The mutual promotion effect of the two systems between regions shows the characteristics of "the east is better than the west" and "the coast is better than the inland". The coupling and coordination degree of different provinces and city types is unbalanced. The above phenomenon reflects the mutual promotion effect of China's digital economy and car-bon emissions reduction still have great potential for improvement.
- It was found that the overall difference of the coupling and coordination be-
tween China's digital economy development and carbon emissions reduction shows an expanding trend, which is reflected within and between regions. Especially among regions, the gap between developing regions and developed regions has widened significantly. And the regional difference is the main source of the overall difference of the coupling and coordination.
- On the whole, the spatial "spillover" effect of the interaction between China's
digital economy development and carbon emission reduction has been obtained. The improvement of the coupling and coordination degree of the two systems in various regions will have an indirect impact on the surrounding regions. The development of the coupling and coordination degree of the two systems has a significant spatial correlation.
Point 4:
Comments and Suggestions for Authors:The interaction between digital economy and carbon emissions is a frontier issue. This study uses econometric models to explore the coupling and co scheduling of China's digital economy system and carbon emission reduction system over the years, and obtains some valuable empirical results. My suggestions are as follows.
Response 4:
Thank you very much for your affirmation of our article. It is your patient review that has made it possible to improve the quality of our article, which has provided great help for the improvement of our research and inspiration for future paper writing. In view of the shortcomings that you proposed to correct in this article, all authors will carefully and carefully read it and try their best to correct them to improve the quality of the manuscript. Thanks again for your help and support!
Point 5:
Figure 1 should be optimized.
Response 5:
Thank you very much for your advice. Figure 1 is a presentation of the internal logic of the construction of the digital economy system in this paper, mainly referring to the definition of digital economy in Bukht and Heeks (2019) and combining the author's innovation. According to the connotation of digital economy, data is the key factor for digital economy to play its role. Therefore, data flow should be a typical phenomenon throughout the entire digital economy. In view of this, we fully agree with your suggestion. Figure 1 can be further optimized. Specifically, the author integrates the manifestation of data flow, making it run through the whole process of the three subsystems of the digital economy. On the one hand, this is conducive to reflecting the role of data elements, on the other hand, it is also conducive to showing its relationship with the digital economy from the side. In addition, in order to highlight the marginal contribution of the author in drawing Figure 1, we have supplemented the image source in the annotation section. The revised contents are as follows:
Line 235-238:
Source: Figure 1 made by the author.
Point 6:
It is suggested that the author refine the most valuable findings.
Response 6:
Thank you very much for your guidance and suggestions. We fully agree with you. Indeed, making clear the most valuable discovery of this article is an important part of attracting readers and highlighting the marginal contribution of this article. Therefore, in order to highlight the core research results of this paper, based on the research content of this paper, we mainly made the following modifications. First, we rewrote the summary. The abstract is a high summary of this study, and also a place where readers can quickly understand the research content of this article. Therefore, the revised version of the abstract includes a simple review of existing research, the purpose of this study, the use of methods, research results and research significance. At the same time, we simplify the language expression, so that the abstract can convey information more clearly. Secondly, at the end of the introduction, we added the research purpose and task of this paper. This will help to clarify the research focus of this paper, so that readers can better grasp the problems to be solved in this paper. Finally, we re present the conclusion. We highly refined the research results and re expressed them with commonly used summary words to avoid duplication with the text. At the same time, we also add the research limitations and research directions that can be expanded in the future, so that the article has more elements that a scientific paper should have. To sum up, we made these parts consistent by improving the abstract, introduction (research purpose and research task) and conclusion, so as to highlight the most valuable findings of this paper. The specific modifications are as follows:
Line 14-30: Abstract: Previously conducted studies have established that digital economy has a one-way inhibition effect on carbon emissions. Under this background, this paper aims to analyze the coordinated development effect of the interaction between digital economy and carbon emissions reduction. The entropy weight method, coupling and coordination degree model, Dagum Gini coefficient and Moran's I index have been carried out as research methods in this paper. The results showed that: (1) The coupling and coordination of China's digital economy and carbon emissions reduction shows an overall growth trend, but the coupling and coordination among regions, provinces and cities show a large imbalance. (2) In the sample period, the overall difference of the coupling and coordination between digital economy development and carbon emissions reduction shows an expanding trend, and the overall difference results are attributed to regional differences. (3)There is a significant spatial correlation in the coupling and coordination degree of digital economy development and carbon emissions reduction among cities. The paper systematically grasps the status of coupling and coordination development, the source of difference and spatial correlation between digital economy and carbon reduction in Chinese cities. A dependence relationship has been established which is digital economy development and carbon emissions reduction, and an interactive promotion pattern has been revealed between digital economy system and carbon emissions reduction system.
Line 178-199: By sorting out relevant domestic and foreign studies, this paper finds that most of the existing literatures focus on the unilateral relationship between digital economy development and carbon emissions based on the measurement of digital economy and carbon emissions. At present, there is still a gap in the discussion of the interaction effect of digital economy and carbon emissions reduction. It can be noted that the relationship between digital economy and carbon emissions reduction is a very topical issue. Therefore, the purpose of this study is to identify the interaction and coordination effect of digital economy and carbon emissions reduction. The application of digital technology can optimize the end treatment technology of enterprise carbon emissions [56,57], and promote carbon emission reduction [58,59]. The promotion of enterprise carbon emission reduction process brings about internal technology innovation and the strengthening of external digital supervision and management, which provide a good technical environment support for the development of digital economy. Therefore, it is necessary to study the coupling and coordination relationship between digital economy development and carbon emissions reduction. And to achieve this, it is necessary to solve the following tasks: (1) It is necessary to measure and comprehensively grasp the status and space-time characteristics of the coupling and coordination between China's digital economy development system and carbon emissions reduction system; (2) The source of the difference for the coupling and coordination between China's digital economy and carbon emissions reduction needs to be explored; (3) The spatial "spillover" effect of the coupling and coordination of digital economy development and carbon emissions reduction among cities is tested.
Line 835-866: This paper measures the coupling and coordination relationship between China's digital economy development and carbon emission reduction by using entropy weight method, building a coupling and coordination degree model, measuring Dagum Gini coefficient and Moran's I index, and analyzes the spatial-temporal features, spatial difference sources and spatial correlation of the coupling and coordination of digital economy and carbon emissions reduction in detail. Due to the limitation of data dis-closure, this paper focuses on the measurement of digital economy development from 2011 to 2018. In the long-term, this may produce certain restrictions on the identification of more significant features of the evaluation results and the capture of development stability. However, according to the research in this paper, China's digital economy is developing rapidly, which makes it possible to supplement the data in the future and improve the long-term dynamic research in this paper. Through systematic research, the research conclusions are as follows:
- The interactive promotion effect and interdependence between China's digital
economy development and carbon emissions reduction have gradually increased, which means that the coupling and coordinated has shown an increasing trend. The mutual promotion effect of the two systems between regions shows the characteristics of "the east is better than the west" and "the coast is better than the inland". The coupling and coordination degree of different provinces and city types is unbalanced. The above phenomenon reflects the mutual promotion effect of China's digital economy and car-bon emissions reduction still have great potential for improvement.
- It was found that the overall difference of the coupling and coordination be-
tween China's digital economy development and carbon emissions reduction shows an expanding trend, which is reflected within and between regions. Especially among regions, the gap between developing regions and developed regions has widened significantly. And the regional difference is the main source of the overall difference of the coupling and coordination.
- On the whole, the spatial "spillover" effect of the interaction between China's
digital economy development and carbon emission reduction has been obtained. The improvement of the coupling and coordination degree of the two systems in various re-gions will have an indirect impact on the surrounding regions. The development of the coupling and coordination degree of the two systems has a significant spatial correla-tion.
Point 7:
Misuse of two reference methods. It is suggested that the author choose one of the two reference methods, either name plus year or serial number.
Response 7:
Thank you very much. We checked the articles that have been published in International Journal of Environmental Research and Public Health. As you said, the citation format of references in the articles needs to be revised. According to your suggestion, we have eliminated the author information corresponding to the references in the article, and only retain the document serial number. At the same time, we carefully checked the correspondence between the serial number of the literature and the information in the references to ensure that the reader can correctly find the relevant author information in the last part. The specific modifications are as follows:
Line 46-49: The digital economy relies on technological innovation to continuously promote industrial integration [1] and economic restructuring, while the transformation of industrial and economic development mode will directly cause changes in carbon dioxide emissions [2].
Line 67-71: Relying on the new digital infrastructure, the digital economy takes knowledge and information as the key production factors, modern information network as the carrier, and the use of information technology as the key driving technology for development, which profoundly affects and fundamentally changes the economic development mode and social activity structure [3].
Line 76-77: Coordination enables the two systems to work together effectively and sustainably [4,5].
Line 82-83: This reflection of the connection between the two systems is coupling [6].
Line 93-94: There is a long way to go to promote the carbon emissions reduction process [7].
Line 100-113: It is believed that the real digital economy refers to the part of economic output completely determined by digital technology, with a business framework based on digital services and goods [8]. With the integration of communication and computing technologies in the network and the flow of technology and data, it is strengthening the transformation of e-commerce and large-scale business [9]. Computer aided data flow has enabled digitization to transform various parts of the current economy [10]. Therefore, the digital economy is also divided into four different parts: digital services and goods; Mixed digital services and goods; IT based intensive production services; And IT industry [11].It is believed that the digital economy is an economic activity based on data application and data technology innovation, with data as the core element [12]. Data technology is the basic support for the development of digital economy. It is a new general technology, mainly through the combination of digital information and the Internet [13], including hardware, software and network technology [14]. There is an opinion that digital economy is a more advanced economic form after industrial economy [15].
Line 114-124: Most of the existing literatures measure the digital economy based on the method of summarizing the digital industry output, digital product classification, and digital related comprehensive indicators. Scholar first discussed the boundary problem of digital economy, which provides a useful reference for measuring digital economy [16]. The digital economy is also divided into five parts for measurement, namely infrastructure, e-commerce, industry structure, digital labor and digital price [17]. Some organizations have proposed an accounting framework with intelligent infrastructure investment, social promotion, innovation release, growth and employment as the main body to measure the digital economy [18]. In addition, there are also scholars who calculate the digital economy by building a comprehensive indicator system [19-21].
Line 126-153: The calculation of carbon dioxide emissions is mainly based on different energy consumption. China's industrial energy carbon emissions have been calculated based on industrial energy consumption [22]. Based on the IPCC inventory method, a research uses the multiplication method of different fuel consumption and fuel carbon emission factors to calculate the regional power carbon emissions [23]. A study took a telecommunications company in Slovenia as the research object, calculated the greenhouse gas emissions within the enterprise scope 3, and took the purchase of electricity, the commuting of employees, the use of vehicles owned by the company and heating as the measured objects [24]. In addition, the research calculates the grid emission factors, deduces the emission factors of specific European countries, and calculates the carbon intensity based on the power generation of power plants [25]. The other is the exploration of carbon emissions reduction path. It is generally believed that economic development is closely related to carbon emissions [26]. The exploration of carbon emissions reduction path mainly involves the improvement of carbon emissions trading market [27,28], the analysis of the policy effect of carbon emissions reduction pilot [29,30] and the research of other paths [31]. In the study of other carbon emissions reduction paths, there is other scholar believed that carbon capture, utilization and storage technology (CCUS) is the only technological path to achieve net zero emissions [32]. And another research believed that the promotion of " Nature Based Solution" (NBS) can provide positive reference for China to solve climate and environmental problems [33]. Mean-while, there is also study carried out from the perspective of energy, believing that the development of renewable energy is of great significance for carbon emissions reduction in resource-based countries [34]. Another study assessed the existing climate policy portfolio and believed that the energy technology portfolio needs to be adjusted [35]. Some scholars discussed the environmental impact of mineral mining based on the mining industry [36,37]. The adjustment of savings also contributes to the development of green economy [38]. In addition, international climate cooperation and climate assistance can also effectively achieve carbon emissions reduction targets [39].
Line 155-171: One is the impact of digital economy development on carbon emissions. Some scholars believe that the digital economy can inhibit carbon emissions [40-42]. A study have proved the role of Taiwan's digital development in the circular economy. It is found that through digital waste management, the consumption of raw materials can be reduced by 25% by 2030, and half of greenhouse gas emissions can be avoided [43]. There is study believed that ICT can contribute to the construction of low-carbon cities [44], mainly through governance capacity of the government [45] and energy systems [46]. Study have also targeted African countries and found an inverted "U" relationship between ICT and carbon emissions in the sample countries [47]. In addition, some scholars believe that R&D investment played a positive role in regulating the relationship between digital economy development and carbon emissions reduction [48-50]. It can be seen that the development of the digital economy has a positive contribution to the promotion of China's green development process [51]. Therefore, it is necessary to study the impact of the digital economy on the climate and environment, which has reference significance for the formulation and improvement of economic and environmental policies [52]. Different from this conclusion, other scholars believe that the development of digital economy is unfavorable to carbon emission reduction [53,54].
Line 176-177: It can be considered that low carbon development can help reverse enterprises' digital transformation [55].
Line 185-187: The application of digital technology can optimize the end treatment technology of enterprise carbon emissions [56,57], and promote carbon emission reduction [58,59].
Line 224-225: For the digital economy, there is scholar believed that there are three layers from the inside to the outside [60].
Line 251-252: The number of enterprises based on Internet development can represent the application level of digital infrastructure [61].
Line 278-280: Number of Internet users. The number of Internet users can reflect the development of digital economy related businesses [62].
Line 287-288: Therefore, it can be calculated according to the digital inclusive financial index of prefecture level cities based on the compilation of indicators by [63].
Line 289-291: The calculation of carbon emissions reduction system mainly refers to the construction method of the green development system, and the sub index construction is based on the input-output model [64]. Low carbon is one of the core concepts of green development [65], so it is reasonable to use the green development system for reference in the construction of carbon emissions reduction system.
Line 301-304: For the carbon dioxide emissions accounting of prefecture level cities, this paper selects the carbon emissions of industrial consumption of natural gas, liquefied petroleum gas and industrial coal-fired electricity to sum up [66].
Line 338-339: Coordination means that two or more systems are interrelated and properly coordinated to form a mutually beneficial development situation [67,68].
Line 379: carbon emissions reduction system [69].
Line 398-401: Dagum Gini coefficient and its decomposition method is to decompose the overall Gini coefficient G into intra group (intra regional) differential contribution , inter group (inter regional) difference contribution and hypervariable density contribution difference , three parts [70].
Point 8:
some subtitles are too miscellaneous, it is recommended that the author refine the title to avoid repetition.
Response 8:
Thank you very much for your patient guidance. We have checked all the subheadings in the original text and fully agree with you. The subheadings should be concise and separate from the first level headings. Therefore, according to your suggestion, we have mainly revised the sub title in order to make it more accurately reflect the research content and avoid duplication with the first level title. The specific modifications are as follows:
Line 450: 3.1. Calculation of the Coupling and Coordination Degree of Two Systems.
Line 492: 3.2. Calculation of the Coupling and Coordination Degree of Two Systems in Eight Regions.
Line 574: 3.3. Calculation of the Coupling and Coordination Degree of Two Systems in Each Province.
Line 652-653: 3.4. Calculation of the Coupling and Coordination Degree of Two Systems in Different Types of Cities.
Point 9:
Table 7 is too redundant and unclear.
Response 9:
Thank you for your suggestion. Table 7 mainly reflects the results of regional difference sources in the coupling and coordination of digital economy and carbon emission reduction. The analysis of the sources of differences in the coupling and coordination is one of the important contents of the three parts of this paper. Therefore, in order to present the results of the sources of differences more clearly, we have readjusted the contents of Table 7 with reference to your suggestions, retaining the core contents, and deleting the rest. At the same time, we explained it to the readers through notes. If the readers are interested in asking for all the results, they can contact the author. The specific modifications are as follows:
Line 799-803:
Table 7. Analysis of national and regional differences in the coupling and coordination of digital economy development and carbon emissions reduction.
|
Year |
2011 |
2012 |
2013 |
2014 |
2015 |
2016 |
2017 |
2018 |
|
|
Gaps inside region |
Overall |
0.1985 |
0.2026 |
0.2029 |
0.2000 |
0.1974 |
0.1992 |
0.2200 |
0.2106 |
|
the northeast economic zone |
0.1277 |
0.1439 |
0.1468 |
0.1386 |
0.1542 |
0.1268 |
0.1646 |
0.1426 |
|
|
the northern economic zone |
0.1222 |
0.1491 |
0.1547 |
0.1520 |
0.1507 |
0.1482 |
0.1383 |
0.1438 |
|
|
the eastern economic zone |
0.0588 |
0.0516 |
0.0637 |
0.0693 |
0.0711 |
0.0732 |
0.0787 |
0.0706 |
|
|
the southern economic zone |
0.1418 |
0.1384 |
0.1464 |
0.1536 |
0.1388 |
0.1418 |
0.1353 |
0.1304 |
|
|
middle reaches of the Yellow River zone |
0.1013 |
0.1206 |
0.0987 |
0.1109 |
0.1433 |
0.1319 |
0.1276 |
0.1359 |
|
|
middle reaches of the Yangtze River zone |
0.1301 |
0.1279 |
0.1175 |
0.1115 |
0.0956 |
0.1056 |
0.1121 |
0.1004 |
|
|
the southwest economic zone |
0.1409 |
0.1322 |
0.1571 |
0.1448 |
0.1304 |
0.1537 |
0.1609 |
0.1603 |
|
|
the northwest economic zone |
0.1389 |
0.0970 |
0.1472 |
0.1445 |
0.1534 |
0.1602 |
0.1528 |
0.1664 |
|
|
|
|
|
|
|
|
|
|
|
|
|
Gaps between regions |
the northeast zone and the northwest zone |
0.1398 |
0.1314 |
0.1697 |
0.1456 |
0.1747 |
0.2080 |
0.1793 |
0.2069 |
|
the northern zone and the eastern zone |
0.2144 |
0.2391 |
0.1926 |
0.1949 |
0.1995 |
0.1870 |
0.1755 |
0.1738 |
|
|
the northern zone and the southern zone |
0.2029 |
0.2121 |
0.1963 |
0.2127 |
0.2025 |
0.2030 |
0.1823 |
0.1690 |
|
|
the northern zone and the northwest zone |
0.2002 |
0.1690 |
0.1697 |
0.2301 |
0.2499 |
0.2862 |
0.3480 |
0.3577 |
|
|
the eastern zone and Yangtze River zone |
0.3163 |
0.1625 |
0.2990 |
0.2772 |
0.2659 |
0.2606 |
0.2583 |
0.2550 |
|
|
the southern zone and Yangtze River zone |
0.2858 |
0.2779 |
0.2737 |
0.2734 |
0.2542 |
0.2622 |
0.2471 |
0.2265 |
|
|
Yellow River zone and the northwest zone |
0.1252 |
0.1161 |
0.1293 |
0.1357 |
0.1610 |
0.1862 |
0.2024 |
0.2207 |
|
|
Yangtze River zone and the northwest zone |
0.1444 |
0.1204 |
0.1566 |
0.1540 |
0.1775 |
0.2090 |
0.2611 |
0.2725 |
|
|
the southwest zone and the northwest zone |
0.1807 |
0.1598 |
0.2031 |
0.2047 |
0.2280 |
0.2470 |
0.2942 |
0.2973 |
|
|
Contribution rate |
|
|
|
|
|
|
|
|
|
|
Within the region |
7.9200 |
8.0105 |
8.2580 |
8.3493 |
8.3617 |
8.4053 |
7.8096 |
7.9497 |
|
|
Between regions |
74.3517 |
73.3009 |
71.7543 |
71.9802 |
71.7502 |
69.8273 |
75.9021 |
74.2676 |
|
|
Hypervariable density |
17.7283 |
18.6886 |
19.9877 |
19.6705 |
19.8881 |
21.7674 |
16.2883 |
17.7827 |
|
Table 7 calculated by the authors. Table 7 only presents the core content. If readers need information about the gaps between other regions, please ask the author for it.
Point 10:
There are too many self-references.
Response 10:
Thank you very much for your advice. Since the research direction of the author is carbon emission reduction, and the previous research of the author is closely related to the theme of this paper, we have cited some articles previously published by some authors as part of the literature support. We fully agree with you and need to reduce our own citations. Therefore, according to your suggestions, we have deleted other self- citations and retained only one article related to the author. Thank you again for your suggestions.
Thank you very much for your patient guidance. We carefully reviewed this article, and learned the very targeted suggestions you gave together for several rounds, aiming to improve the quality of this paper so that it can meet the international journal publishing requirements. All the authors would like to express their sincere thanks to you for your patient review and detailed theoretical guidance for the improvement of this article. It is your careful guidance and help that we have the opportunity to improve this paper, and your suggestions will help us avoid premature ideas in the future paper writing. All the suggestions you put forward have given us great inspiration for this article and the writing of future papers, and let us understand the characteristics of a scientific and interesting high-quality international article. Such valuable guidance opportunities are very difficult to meet in our daily writing, and all authors cherish them very much. Finally, all the authors sincerely thank you for your valuable time to help and support us, so that we can make continuous progress. Thank you very much!
Thanks very much to the reviewers and editors for your valuable comments and suggestions, so that the quality of the manuscript can be improved. If there is also some improper expression in the revised manuscript, please let us know and we will further improve it.

Reviewer 4 Report
Dear authors,
Please find my observations below:
The manuscript needs professional English proofreading in order to improve message delivery.
Abstract: it is a bit too long. Try to be more synthetic and express your ideas in more concise phrases.
“Data” is plural, so you have to state “Data show that…”.
Format references according to journal requirements.
Pay attention when using words between quotation marks. Sometimes you put a blank space after the first quotation mark (see page 2).
Page 3: pay attention when referring to studies in the literature. “Pei et al.(2018) believes..”should be “Pei at al. (2018) believe that..”, since there is more than one author in this study. Apply this in the entire paper.
What is the source of Figure 1? Please indicate the source under this figure.
Page 7: when presenting variables, you should use the verb in singular. Currently you use a mix of both singular and plural. (“CO2 is the total carbon emissions….electricCO2 are the carbon…”).
In the last section, please add study limitations, which currently are missing.
The reference list should be expanded with the following titles:
· Batrancea, L., Rathnaswamy, M.M., Batrancea, I., Nichita, A., Rus, M.-I., Tulai, H., Fatacean, G., Masca, E.S., & Morar, I.D. (2020). Adjusted net savings of CEE and Baltic nations in the context of sustainable economic growth: A panel data analysis. Journal of Risk and Financial Management, 13(10), 234.
· Batrancea, I., Rathnaswamy, M.K., Batrancea, L., Nichita, A., Gaban, L., Fatacean, G., Tulai, H., Bircea, I., & Rus, M.-I. (2020). A panel data analysis on sustainable economic growth in India, Brazil, and Romania. Journal of Risk and Financial Management, 13(8), 170.
Author Response
Analysis on the Coupling Effect and Space-Time Difference between China's Digital Economy Development and Carbon Emissions Reduction
Response to Reviewer 4 Comments
We are grateful for the reviewer’s thoughtful comments and suggestions. We have thoroughly revised our manuscript according to the comments. Our responses (in blue) and the reviewer's original comments (in black) are given below:
(The specific modification content can also be seen in the word version uploaded by the author)
Point 1:
English language and style:Moderate English changes required.
Response 1:
Thank you very much for your advice. All authors fully agree with you. We carefully checked the article and found syntax errors in the model design part, so we corrected them. In order to ensure the professionalism of the full text language, we submitted the article to the language experts for modification to avoid language errors in this article. The specific modifications made by the authors are as follows:
Line 306-309: In Formula (1), is the total carbon emissions of prefecture level cities, is the carbon emissions of natural gas consumed by industry, is the carbon emissions of liquefied petroleum gas consumed by industry and is the carbon emissions of electricity consumed by industry.
Point 2:
Does the introduction provide sufficient background and include all relevant references?——Can be improved.
Response 2:
Thank you very much for your comments. According to your comments, we checked the introduction and referred to the articles published in International Journal of Environmental Research and Public Health. Indeed, as you said, it is necessary to modify the introduction. We mainly made the following modifications:
First, we revised the research background. The policy background with low relevance to the theme of this paper is deleted, and the background content of digital economy and carbon emissions is re-presented, making the research background more consistent with the theme of this paper.
Second, in the introduction, we re-elaborated the connotation of coordinated development of digital economy and carbon emission reduction, so that readers can better understand the content of the coupling and coordination degree of the two systems. At the same time, it is also helpful for readers to understand the content of this study.
Third, we have made a lot of supplements to the literature in the field of digital economy and carbon emissions, presenting as much international research as possible, especially research from different countries. On the one hand, it reflects the comprehensiveness of the literature review of this paper, on the other hand, it also makes this paper more in line with the requirements of international journals. In addition, while supplementing references, we also added relevant documents to the arguments in the introduction to better support the establishment of the arguments, making the views in the article more persuasive.
Finally, in the introduction part, we clarified the purpose and task of this study, and corresponded it with the research contents and conclusions of this paper. In the introduction, the purpose and task of the study are presented, which will help readers to better understand the research content of the article and improve the logic of this study.
The specific modifications are as follows:
Line 42-54: Digital economy is an economic system with technology as its core driving force, and digitalization of economic activities is a typical feature of digital economic system. the new development mode led by technological innovation driven digital economy development has laid the foundation for the coordinated promotion of digital economy and green development. The digital economy relies on technological innovation to continuously promote industrial integration [1] and economic restructuring, while the transformation of industrial and economic development mode will directly cause changes in carbon dioxide emissions [2]. At the same time, under background of " emission peak" and "carbon neutrality", the digital regulation on which the carbon emissions reduction process is based will also have a certain impact on the development of the digital economy. Therefore, it is urgent to systematically answer the interactive development relationship between digital economy and carbon emissions reduction, so that the development of digital economy can support the advancement of carbon emissions reduction.
Line 71-83: The analysis of the interaction between digital economy and carbon emissions reduction is the analysis of the coordinated development between digital economy and carbon emissions reduction. Coordinated development includes two explanations. On the one hand, from the perspective of coordination, it emphasizes the positive interaction between digital economy development and carbon emission reduction and the formation of a virtuous circle situation that promotes each other. Coordination enables the two systems to work together effectively and sustainably [4,5]. On the other hand, from the perspective of development, it emphasizes the promotion of digital economy and carbon emissions reduction. The coordinated development between digital economy and carbon emissions reduction requires that digital economy elements can help promote the process of carbon emissions reduction through technology, which in turn can continue to provide technical and environmental support for digital development. This reflection of the connection between the two systems is coupling [6].
Line 93-97: There is a long way to go to promote the carbon emissions reduction process [7]. Therefore, promoting the research on the status and trend of coordinated development of digital economy and carbon emissions reduction is of practical significance for im-proving the layout of digital economy elements and promoting green development through digitalization.
Line 100-108: It is believed that the real digital economy refers to the part of economic output completely determined by digital technology, with a business framework based on digital services and goods [8].. With the integration of communication and computing technologies in the network and the flow of technology and data, it is strengthening the transformation of e-commerce and large-scale business [9]. Computer aided data flow has enabled digitization to transform various parts of the current economy [10]. Therefore, the digital economy is also divided into four different parts: digital services and goods; Mixed digital services and goods; IT based intensive production services; And IT industry [11].
Line 114-124: Most of the existing literatures measure the digital economy based on the method of summarizing the digital industry output, digital product classification, and digital related comprehensive indicators. Scholar first discussed the boundary problem of digital economy, which provides a useful reference for measuring digital economy [16]. The digital economy is also divided into five parts for measurement, namely infrastructure, e-commerce, industry structure, digital labor and digital price [17]. Some organizations have proposed an accounting framework with intelligent infrastructure investment, social promotion, innovation release, growth and employment as the main body to measure the digital economy [18]. In addition, there are also scholars who calculate the digital economy by building a comprehensive indicator system [19-21].
Line 131-138: A study took a telecommunications company in Slovenia as the research object, calculated the greenhouse gas emissions within the enterprise scope 3, and took the purchase of electricity, the commuting of employees, the use of vehicles owned by the company and heating as the measured objects [24]. In addition, the research calculates the grid emission factors, deduces the emission factors of specific European countries, and calculates the carbon intensity based on the power generation of power plants [25]. The other is the exploration of carbon emissions reduction path. It is generally believed that economic development is closely related to carbon emissions [26].
Line 140: The exploration of carbon emissions reduction path mainly involves the
improvement of carbon emissions trading market [27,28], the analysis of the policy effect of carbon emissions reduction pilot [29,30] and the research of other paths [31].
Line 145-152: Meanwhile, there is also study carried out from the perspective of energy, believing that the development of renewable energy is of great significance for carbon emissions reduction in resource-based countries [34]. Another study assessed the existing climate policy portfolio and believed that the energy technology portfolio needs to be adjusted [35]. Some scholars discussed the environmental impact of mineral mining based on the mining industry [36,37]. The adjustment of savings also contributes to the development of green economy [38].
Line 155-166: One is the impact of digital economy development on carbon emissions. Some scholars believe that the digital economy can inhibit carbon emissions [40-42]. A study have proved the role of Taiwan's digital development in the circular economy. It is found that through digital waste management, the consumption of raw materials can be reduced by 25% by 2030, and half of greenhouse gas emissions can be avoided [43]. There is study believed that ICT can contribute to the construction of low-carbon cities [44], mainly through governance capacity of the government [45] and energy systems [46]. Study have also targeted African countries and found an inverted "U" relationship between ICT and carbon emissions in the sample countries [47]. In addition, some scholars believe that R&D investment played a positive role in regulating the relationship between digital economy development and carbon emissions reduction [48-50].
Line 170-171: Different from this conclusion, other scholars believe that the development of digital economy is unfavorable to carbon emission reduction [53,54].
Line 178-185: By sorting out relevant domestic and foreign studies, this paper finds that most of the existing literatures focus on the unilateral relationship between digital economy development and carbon emissions based on the measurement of digital economy and carbon emissions. At present, there is still a gap in the discussion of the interaction effect of digital economy and carbon emissions reduction. It can be noted that the relationship between digital economy and carbon emissions reduction is a very topical issue. Therefore, the purpose of this study is to identify the interaction and coordination effect of digital economy and carbon emissions reduction.
Line 190-199: Therefore, it is necessary to study the coupling and coordination relationship between digital economy development and carbon emissions reduction. And to achieve this, it is necessary to solve the following tasks: (1) It is necessary to measure and comprehensively grasp the status and space-time characteristics of the coupling and coordination between China's digital economy development system and carbon emissions reduction system; (2) The source of the difference for the coupling and coordination between China's digital economy and carbon emissions reduction needs to be explored; (3) The spatial "spillover" effect of the coupling and coordination of digital economy development and carbon emissions reduction among cities is tested.
Point 3:
Are all the cited references relevant to the research?——Can be improved.
Response 3:
Thank you very much for your patient guidance. We checked the citation of references and found that the citation of some references was indeed unnecessary. Therefore, we fully agree with your point of view, and need to modify the references. We mainly made the following three changes: First, we added references to the important arguments in the introduction to support them, so as to improve the scientific expression of the arguments. In the process of adding, we mainly choose to add relevant documents in the field of digital economy and environment represented by carbon emission reduction to ensure that the references are highly consistent with the theme of this paper. Second, we have added 31 relevant articles, especially international articles. After re searching, we found that there have been a lot of research on digital economy and carbon emission reduction in the world. Therefore, we added more literature in this field to sort out the existing research under this theme as completely as possible. Finally, we deleted marginal literature, mainly focusing on the positive impact of digital economy on carbon emission reduction. The specific modifications are as follows:
Line 42-54: Digital economy is an economic system with technology as its core driving force, and digitalization of economic activities is a typical feature of digital economic system. the new development mode led by technological innovation driven digital economy development has laid the foundation for the coordinated promotion of digital economy and green development. The digital economy relies on technological innovation to continuously promote industrial integration [1] and economic restructuring, while the transformation of industrial and economic development mode will directly cause changes in carbon dioxide emissions [2]. At the same time, under background of " emission peak" and "carbon neutrality", the digital regulation on which the carbon emissions reduction process is based will also have a certain impact on the development of the digital economy. Therefore, it is urgent to systematically answer the interactive development relationship between digital economy and carbon emissions reduction, so that the development of digital economy can support the advancement of carbon emissions reduction.
Line 71-83: The analysis of the interaction between digital economy and carbon emissions reduction is the analysis of the coordinated development between digital economy and carbon emissions reduction. Coordinated development includes two explanations. On the one hand, from the perspective of coordination, it emphasizes the positive interaction between digital economy development and carbon emission reduction and the formation of a virtuous circle situation that promotes each other. Coordination enables the two systems to work together effectively and sustainably [4,5]. On the other hand, from the perspective of development, it emphasizes the promotion of digital economy and carbon emissions reduction. The coordinated development between digital economy and carbon emissions reduction requires that digital economy elements can help promote the process of carbon emissions reduction through technology, which in turn can continue to provide technical and environmental support for digital development. This reflection of the connection between the two systems is coupling [6].
Line 93-97: There is a long way to go to promote the carbon emissions reduction process [7]. Therefore, promoting the research on the status and trend of coordinated development of digital economy and carbon emissions reduction is of practical significance for improving the layout of digital economy elements and promoting green development through digitalization.
Line 100-108: It is believed that the real digital economy refers to the part of economic output completely determined by digital technology, with a business framework based on digital services and goods [8]. With the integration of communication and computing technologies in the network and the flow of technology and data, it is strengthening the transformation of e-commerce and large-scale business [9]. Computer aided data flow has enabled digitization to transform various parts of the current economy [10]. Therefore, the digital economy is also divided into four different parts: digital services and goods; Mixed digital services and goods; IT based intensive production services; And IT industry [11].
Line 114-124: Most of the existing literatures measure the digital economy based on the method of summarizing the digital industry output, digital product classification, and digital related comprehensive indicators. Scholar first discussed the boundary problem of digital economy, which provides a useful reference for measuring digital economy [16]. The digital economy is also divided into five parts for measurement, namely infrastructure, e-commerce, industry structure, digital labor and digital price [17]. Some organizations have proposed an accounting framework with intelligent infrastructure investment, social promotion, innovation release, growth and employment as the main body to measure the digital economy [18]. In addition, there are also scholars who calculate the digital economy by building a comprehensive indicator system [19-21].
Line 131-141: A study took a telecommunications company in Slovenia as the research object, calculated the greenhouse gas emissions within the enterprise scope 3, and took the purchase of electricity, the commuting of employees, the use of vehicles owned by the company and heating as the measured objects [24]. In addition, the research calculates the grid emission factors, deduces the emission factors of specific European countries, and calculates the carbon intensity based on the power generation of power plants [25]. The other is the exploration of carbon emissions reduction path. It is generally believed that economic development is closely related to carbon emissions [26]. The exploration of carbon emissions reduction path mainly involves the improvement of carbon emissions trading market [27,28], the analysis of the policy effect of carbon emissions reduction pilot [29,30].
Line 145-152: Meanwhile, there is also study carried out from the perspective of energy, believing that the development of renewable energy is of great significance for carbon emissions reduction in resource-based countries [34]. Another study assessed the existing climate policy portfolio and believed that the energy technology portfolio needs to be adjusted [35]. Some scholars discussed the environmental impact of mineral mining based on the mining industry [36,37]. The adjustment of savings also contributes to the development of green economy [38].
Line 155-166: One is the impact of digital economy development on carbon emissions. Some scholars believe that the digital economy can inhibit carbon emissions [40-42]. A study have proved the role of Taiwan's digital development in the circular economy. It is found that through digital waste management, the consumption of raw materials can be reduced by 25% by 2030, and half of greenhouse gas emissions can be avoided [43]. There is study believed that ICT can contribute to the construction of low-carbon cities [44], mainly through governance capacity of the government [45] and energy systems [46]. Study have also targeted African countries and found an inverted "U" relationship between ICT and carbon emissions in the sample countries [47]. In addition, some scholars believe that R&D investment played a positive role in regulating the relationship between digital economy development and carbon emissions reduction [48-50].
Line 170-171: Different from this conclusion, other scholars believe that the development of digital economy is unfavorable to carbon emission reduction [53,54].
Line 178-185: By sorting out relevant domestic and foreign studies, this paper finds that most of the existing literatures focus on the unilateral relationship between digital economy development and carbon emissions based on the measurement of digital economy and carbon emissions. At present, there is still a gap in the discussion of the interaction effect of digital economy and carbon emissions reduction. It can be noted that the relationship between digital economy and carbon emissions reduction is a very topical issue. Therefore, the purpose of this study is to identify the interaction and coordination effect of digital economy and carbon emissions reduction.
Line 190-199: Therefore, it is necessary to study the coupling and coordination relationship between digital economy development and carbon emissions reduction. And to achieve this, it is necessary to solve the following tasks: (1) It is necessary to measure and comprehensively grasp the status and space-time characteristics of the coupling and coordination between China's digital economy development system and carbon emissions reduction system; (2) The source of the difference for the coupling and coordination between China's digital economy and carbon emissions reduction needs to be explored; (3) The spatial "spillover" effect of the coupling and coordination of digital economy development and carbon emissions reduction among cities is tested.
Line 450: 3.1. Calculation of the Coupling and Coordination Degree of Two Systems.
Line 492: 3.2. Calculation of the Coupling and Coordination Degree of Two Systems in Eight Regions.
Line 574: 3.3. Calculation of the Coupling and Coordination Degree of Two Systems in Each Province.
Line 652: 3.4. Calculation of the Coupling and Coordination Degree of Two Systems in Different Types of Cities.
Added References:
- Zuo, P.; Jiang, Q.; Chen, J. Internet development, urbanization and the upgrading of China's industrial structure. Quantitative Economics and Technical Economics. 2020, 7: 71-91.
- Hosseinzadeh, B.H.; Nabavi, P. A.; Khanali, M.; Ghahderijani, M.; Chau, K. Application of data envelopment analysis ap-proach for optimization of energy use and reduction of greenhouse gas emission in peanut production of Iran. Journal of Cleaner Production. 2018,172:1327-1335.
- Kunz, N.C.; Moran, C.J.; Kastelle, T. Conceptualising “coupling” for sustainability implementation in the industrial sector: a review of the field and projection of future research opportunities. Journal of Cleaner Production. 2013, 53:69-80.
- Carl, F. Resilience: the emergence of a perspective for social-ecological systems analyses. Global Environmental Change. 2006, 16 (3):253-267.
- Solymar, L., Webb, D., Grunnet, J, A. The Physics and Applications of Photorefractive Materials. 1996, 11,Clarendon Press.
- Yan, B.; Erda, W.; Dominik, M.; Martin, L. How population migration affects carbon emissions in China: Factual and coun-terfactual scenario analysis. Technological Forecasting and Social Change. 2022,184: 122023
- Luyanda, D.W. Concepts of digital economy and industry 4.0 in intelligent and information systems. International Journal of Intelligent Networks. 2021,2:122-129.
- Mahmod, S. A. 5G wireless technologies- future generation communication technologies. International Journal of Computing and Digital Systemss. 2017, 6 (3):139-147.
- Chouhan, N.; Rathore, D. Role of digitalization after demonetization in economy. International Journal of Computer Sciences and Engineering. 2018, 6(9): 88-90.
- Szeto, K. Keeping score, digitally. Music Reference Services Quarterly. 2018, 21 (2):98-100.
- Margheriol, D. H.; Cook, S.; Montes, S. The emerging digital economy. US Department of Commerce. 1998.
- Kahin, B.; Brynjolfsson, E. Understanding the digital economy. The MIT Press. 2000.
- OECD. Measuring the digital economy: an new perspective. Pairs: OECD Publishing. 2014.
- Shahzad, M.; Wang, J.; Dong, K.; Zhao, J. The impact of digital economy on energy transition across the globe: the mediating role of government governance. Renewable and Sustainable Energy Reviews. 2022, 166 (3) :112620.
- Gregor, R.; Saša, T. Carbon footprint calculation in telecommunications companies – The importance and relevance of scope 3 greenhouse gases emissions. Renewable and Sustainable Energy Reviews. 2018, 98: 361-375.
- Jan, F.U.; Anke, W.; Leonhard, G.; Mirko, S. Open-data based carbon emission intensity signals for electricity generation in European countries – top down vs. bottom up approach. Cleaner Energy Systems. 2022, 3: 100018
- Batrancea, I.; Rathnaswamy, M.K.; Batrancea, L.; Nichita, A.; Gaban, L.; Fatacean, G.; Tulai, H.; Bircea, I.; Rus, M. I. A panel data analysis on sustainable economic growth in India, Brazil, and Romania. Journal of Risk and Financial Management. 2020, 13(8): 170.
- Huo, W.D.; Qi, J.; Yang, T.; Liu, J.L.; Liu, M.M.; Zhou, Z.Q. Effects of China's pilot low-carbon city policy on carbon emission reduction: A quasi-natural experiment based on satellite data. Technological Forecasting and Social Change. 2022,175: 121422.
- Liu, X.; Li, Y.C.; Chen, X.H.; Liu, J. Evaluation of low carbon city pilot policy effect on carbon abatement in China: An em-pirical evidence based on time-varying DID model. Cities. 2022, 123: 103582.
- Tcvetkov, P. Engagement of resource-based economies in the fight against rising carbon emissions. Energy Reports. 2022, 8: 874-883.
- Tcvetkov, P. Climate policy imbalance in the energy sector: Time to focus on the value of CO2 utilization. Energies. 2021, 14(2): 411.
- Khayrutdinov, M.M.; Golik, V.I.; Aleksakhin, A.V.; Trushina, E.V.; Lazareva, N.V.; Aleksakhina, Y.V. Proposal of an Algo-rithm for Choice of a Development System for Operational and Environmental Safety in Mining. Resources. 2022, 11: 88.
- Kongar, S.C.; Ubysz, A.; Faradzhov, V. Models and algorithms of choice of development technology of deposits when se-lecting the composition of the backfilling mixture. IOP Conference Series Earth and Environmental Science. 2021, 684(1): 012008.
- Batrancea, L., Rathnaswamy, M.M., Batrancea, I., Nichita, A., Rus, M.-I., Tulai, H., Fatacean, G., Masca, E.S., Morar, I.D. Adjusted net savings of CEE and Baltic nations in the context of sustainable economic growth: A panel data analysis. Journal of Risk and Financial Management. 2020, 13(10): 234.
- Tonni, A.K.; Aleksandra, M.; Marina, K.; Elena,B.; Mohd, H.D.O.; Hui, H.G. Accelerating sustainability transition in St. Petersburg (Russia) through digitalization-based circular economy in waste recycling industry: A strategy to promote carbon neutrality in era of Industry 4.0. Journal of Cleaner Production. 2022,363: 132452
- Jacob, P. Information and communication technology in shaping urban low carbon development pathways.Current Opinion in Environmental Sustainability. 2018, 30:133-137
- Muhammad, S.; Wang, J.D.; Dong, K.Y.; Zhao, J. The impact of digital economy on energy transition across the globe: The mediating role of government governance. Renewable and Sustainable Energy Reviews. 2022, 166:112620,
- Moyer, J.D.; Hughes, B.B. ICTs: Do they contribute to increased carbon emissions?. Technological Forecasting & Social Change. 2012, 79 (5) :919-931
- Asongu, S.A.; Roux, S. L.; Biekpe, N. Enhancing ICT for environmental sustainability in sub-Saharan Africa. Technological Forecasting & Social Change. 2018, 127: 209-21.
- Ilhan, O.; Sana, U. Does digital financial inclusion matter for economic growth and environmental sustainability in OBRI economies? An empirical analysis. Resources, Conservation and Recycling. 2022, 185:106489.
- Salahuddin, M.; Alam, K. Internet usage, electricity consumption and economic growth in Australia: a time series evidence. Telematics and Informatics. 2015, 32 (4): 862-878.
Point 4:
Is the research design appropriate?——Can be improved.
Response 4:
Thank you very much for your patient guidance. We checked the full text and agreed with your opinion that it is necessary to improve the research design. A good research design can make the article more logical and help readers better understand the content of the article. Therefore, according to your suggestions, we have made design modifications to the article. It mainly includes two aspects: one is to modify the sub title of the paper to make the content more consistent with the title. The other is to modify the content, mainly including introduction and conclusion. In the introduction, we have added new contents, including research background, new references to the support of the argument and the sorting of international literature, as well as the research purpose and task of this paper. We have deleted the policy introduction that is less relevant to this article. At the same time, it has supplemented the reference basis for the important argument to make the argument more convincing. In addition, we have also added 31 international documents from different countries, which expands the scope of this paper's sorting of previous studies, enriches the literature review part of this paper, and increases the interest and internationality of this paper. In addition, in order to make readers better understand the problems to be solved in this article, we added the research purpose and task of this article at the end of the introduction, making them correspond to the content and conclusion, which helps to improve the internal logic of the article. In the conclusion, we made a new summary, highly refined the research results, and avoided the repetition of its expression and the text. At the same time, we also add the research limitations and research directions that can be expanded in the future, so that the article has more elements that a scientific paper should have. To sum up, we have revised the title design and research content. The specific modifications are as follows:
Line 42-54: Digital economy is an economic system with technology as its core driving force, and digitalization of economic activities is a typical feature of digital economic system. the new development mode led by technological innovation driven digital economy development has laid the foundation for the coordinated promotion of digital economy and green development. The digital economy relies on technological innovation to continuously promote industrial integration [1] and economic restructuring, while the transformation of industrial and economic development mode will directly cause changes in carbon dioxide emissions [2]. At the same time, under background of " emission peak" and "carbon neutrality", the digital regulation on which the carbon emissions reduction process is based will also have a certain impact on the development of the digital economy. Therefore, it is urgent to systematically answer the interactive development relationship between digital economy and carbon emissions reduction, so that the development of digital economy can support the advancement of carbon emissions reduction.
Line 71-83: The analysis of the interaction between digital economy and carbon emissions reduction is the analysis of the coordinated development between digital economy and carbon emissions reduction. Coordinated development includes two explanations. On the one hand, from the perspective of coordination, it emphasizes the positive interaction between digital economy development and carbon emission reduction and the formation of a virtuous circle situation that promotes each other. Coordination enables the two systems to work together effectively and sustainably [4,5]. On the other hand, from the perspective of development, it emphasizes the promotion of digital economy and carbon emissions reduction. The coordinated development between digital economy and carbon emissions reduction requires that digital economy elements can help promote the process of carbon emissions reduction through technology, which in turn can continue to provide technical and environmental support for digital development. This reflection of the connection between the two systems is coupling [6].
Line 93-97: There is a long way to go to promote the carbon emissions reduction process [7]. Therefore, promoting the research on the status and trend of coordinated development of digital economy and carbon emissions reduction is of practical significance for improving the layout of digital economy elements and promoting green development through digitalization.
Line 100-108: It is believed that the real digital economy refers to the part of economic output completely determined by digital technology, with a business framework based on digital services and goods [8]. With the integration of communication and computing technologies in the network and the flow of technology and data, it is strengthening the transformation of e-commerce and large-scale business [9]. Computer aided data flow has enabled digitization to transform various parts of the current economy [10]. Therefore, the digital economy is also divided into four different parts: digital services and goods; Mixed digital services and goods; IT based intensive production services; And IT industry [11].
Line 114-124: Most of the existing literatures measure the digital economy based on the method of summarizing the digital industry output, digital product classification, and digital related comprehensive indicators. Scholar first discussed the boundary problem of digital economy, which provides a useful reference for measuring digital economy [16]. The digital economy is also divided into five parts for measurement, namely infrastructure, e-commerce, industry structure, digital labor and digital price [17]. Some organizations have proposed an accounting framework with intelligent infrastructure investment, social promotion, innovation release, growth and employment as the main body to measure the digital economy [18]. In addition, there are also scholars who calculate the digital economy by building a comprehensive indicator system [19-21].
Line 131-141: A study took a telecommunications company in Slovenia as the research object, calculated the greenhouse gas emissions within the enterprise scope 3, and took the purchase of electricity, the commuting of employees, the use of vehicles owned by the company and heating as the measured objects [24]. In addition, the research calculates the grid emission factors, deduces the emission factors of specific European countries, and calculates the carbon intensity based on the power generation of power plants [25]. The other is the exploration of carbon emissions reduction path. It is generally believed that economic development is closely related to carbon emissions [26]. The exploration of carbon emissions reduction path mainly involves the improvement of carbon emissions trading market [27,28], the analysis of the policy effect of carbon emissions reduction pilot [29,30].
Line 145-152: Meanwhile, there is also study carried out from the perspective of energy, believing that the development of renewable energy is of great significance for carbon emissions reduction in resource-based countries [34]. Another study assessed the existing climate policy portfolio and believed that the energy technology portfolio needs to be adjusted [35]. Some scholars discussed the environmental impact of mineral mining based on the mining industry [36,37]. The adjustment of savings also contributes to the development of green economy [38].
Line 155-166: One is the impact of digital economy development on carbon emissions. Some scholars believe that the digital economy can inhibit carbon emissions [40-42]. A study have proved the role of Taiwan's digital development in the circular economy. It is found that through digital waste management, the consumption of raw materials can be reduced by 25% by 2030, and half of greenhouse gas emissions can be avoided [43]. There is study believed that ICT can contribute to the construction of low-carbon cities [44], mainly through governance capacity of the government [45] and energy systems [46]. Study have also targeted African countries and found an inverted "U" relationship between ICT and carbon emissions in the sample countries [47]. In addition, some scholars believe that R&D investment played a positive role in regulating the relationship between digital economy development and carbon emissions reduction [48-50].
Line 170-171: Different from this conclusion, other scholars believe that the development of digital economy is unfavorable to carbon emission reduction [53,54].
Line 178-185: By sorting out relevant domestic and foreign studies, this paper finds that most of the existing literatures focus on the unilateral relationship between digital economy development and carbon emissions based on the measurement of digital economy and carbon emissions. At present, there is still a gap in the discussion of the interaction effect of digital economy and carbon emissions reduction. It can be noted that the relationship between digital economy and carbon emissions reduction is a very topical issue. Therefore, the purpose of this study is to identify the interaction and coordination effect of digital economy and carbon emissions reduction.
Line 190-199: Therefore, it is necessary to study the coupling and coordination relationship between digital economy development and carbon emissions reduction. And to achieve this, it is necessary to solve the following tasks: (1) It is necessary to measure and comprehensively grasp the status and space-time characteristics of the coupling and coordination between China's digital economy development system and carbon emissions reduction system; (2) The source of the difference for the coupling and coordination between China's digital economy and carbon emissions reduction needs to be explored; (3) The spatial "spillover" effect of the coupling and coordination of digital economy development and carbon emissions reduction among cities is tested.
Line 450: 3.1. Calculation of the Coupling and Coordination Degree of Two Systems.
Line 492: 3.2. Calculation of the Coupling and Coordination Degree of Two Systems in Eight Regions.
Line 574: 3.3. Calculation of the Coupling and Coordination Degree of Two Systems in Each Province.
Line 652: 3.4. Calculation of the Coupling and Coordination Degree of Two Systems in Different Types of Cities.
Added References:
- Zuo, P.; Jiang, Q.; Chen, J. Internet development, urbanization and the upgrading of China's industrial structure. Quantitative Economics and Technical Economics. 2020, 7: 71-91.
- Hosseinzadeh, B.H.; Nabavi, P. A.; Khanali, M.; Ghahderijani, M.; Chau, K. Application of data envelopment analysis ap-proach for optimization of energy use and reduction of greenhouse gas emission in peanut production of Iran. Journal of Cleaner Production. 2018,172:1327-1335.
- Kunz, N.C.; Moran, C.J.; Kastelle, T. Conceptualising “coupling” for sustainability implementation in the industrial sector: a review of the field and projection of future research opportunities. Journal of Cleaner Production. 2013, 53:69-80.
- Carl, F. Resilience: the emergence of a perspective for social-ecological systems analyses. Global Environmental Change. 2006, 16 (3):253-267.
- Solymar, L., Webb, D., Grunnet, J, A. The Physics and Applications of Photorefractive Materials. 1996, 11,Clarendon Press.
- Yan, B.; Erda, W.; Dominik, M.; Martin, L. How population migration affects carbon emissions in China: Factual and coun-terfactual scenario analysis. Technological Forecasting and Social Change. 2022,184: 122023
- Luyanda, D.W. Concepts of digital economy and industry 4.0 in intelligent and information systems. International Journal of Intelligent Networks. 2021,2:122-129.
- Mahmod, S. A. 5G wireless technologies- future generation communication technologies. International Journal of Computing and Digital Systemss. 2017, 6 (3):139-147.
- Chouhan, N.; Rathore, D. Role of digitalization after demonetization in economy. International Journal of Computer Sciences and Engineering. 2018, 6(9): 88-90.
- Szeto, K. Keeping score, digitally. Music Reference Services Quarterly. 2018, 21 (2):98-100.
- Margheriol, D. H.; Cook, S.; Montes, S. The emerging digital economy. US Department of Commerce. 1998.
- Kahin, B.; Brynjolfsson, E. Understanding the digital economy. The MIT Press. 2000.
- OECD. Measuring the digital economy: an new perspective. Pairs: OECD Publishing. 2014.
- Shahzad, M.; Wang, J.; Dong, K.; Zhao, J. The impact of digital economy on energy transition across the globe: the mediating role of government governance. Renewable and Sustainable Energy Reviews. 2022, 166 (3) :112620.
- Gregor, R.; Saša, T. Carbon footprint calculation in telecommunications companies – The importance and relevance of scope 3 greenhouse gases emissions. Renewable and Sustainable Energy Reviews. 2018, 98: 361-375.
- Jan, F.U.; Anke, W.; Leonhard, G.; Mirko, S. Open-data based carbon emission intensity signals for electricity generation in European countries – top down vs. bottom up approach. Cleaner Energy Systems. 2022, 3: 100018
- Batrancea, I.; Rathnaswamy, M.K.; Batrancea, L.; Nichita, A.; Gaban, L.; Fatacean, G.; Tulai, H.; Bircea, I.; Rus, M. I. A panel data analysis on sustainable economic growth in India, Brazil, and Romania. Journal of Risk and Financial Management. 2020, 13(8): 170.
- Huo, W.D.; Qi, J.; Yang, T.; Liu, J.L.; Liu, M.M.; Zhou, Z.Q. Effects of China's pilot low-carbon city policy on carbon emission reduction: A quasi-natural experiment based on satellite data. Technological Forecasting and Social Change. 2022,175: 121422.
- Liu, X.; Li, Y.C.; Chen, X.H.; Liu, J. Evaluation of low carbon city pilot policy effect on carbon abatement in China: An em-pirical evidence based on time-varying DID model. Cities. 2022, 123: 103582.
- Tcvetkov, P. Engagement of resource-based economies in the fight against rising carbon emissions. Energy Reports. 2022, 8: 874-883.
- Tcvetkov, P. Climate policy imbalance in the energy sector: Time to focus on the value of CO2 utilization. Energies. 2021, 14(2): 411.
- Khayrutdinov, M.M.; Golik, V.I.; Aleksakhin, A.V.; Trushina, E.V.; Lazareva, N.V.; Aleksakhina, Y.V. Proposal of an Algo-rithm for Choice of a Development System for Operational and Environmental Safety in Mining. Resources. 2022, 11: 88.
- Kongar, S.C.; Ubysz, A.; Faradzhov, V. Models and algorithms of choice of development technology of deposits when se-lecting the composition of the backfilling mixture. IOP Conference Series Earth and Environmental Science. 2021, 684(1): 012008.
- Batrancea, L., Rathnaswamy, M.M., Batrancea, I., Nichita, A., Rus, M.-I., Tulai, H., Fatacean, G., Masca, E.S., Morar, I.D. Adjusted net savings of CEE and Baltic nations in the context of sustainable economic growth: A panel data analysis. Journal of Risk and Financial Management. 2020, 13(10): 234.
- Tonni, A.K.; Aleksandra, M.; Marina, K.; Elena,B.; Mohd, H.D.O.; Hui, H.G. Accelerating sustainability transition in St. Petersburg (Russia) through digitalization-based circular economy in waste recycling industry: A strategy to promote carbon neutrality in era of Industry 4.0. Journal of Cleaner Production. 2022,363: 132452
- Jacob, P. Information and communication technology in shaping urban low carbon development pathways.Current Opinion in Environmental Sustainability. 2018, 30:133-137
- Muhammad, S.; Wang, J.D.; Dong, K.Y.; Zhao, J. The impact of digital economy on energy transition across the globe: The mediating role of government governance. Renewable and Sustainable Energy Reviews. 2022, 166:112620,
- Moyer, J.D.; Hughes, B.B. ICTs: Do they contribute to increased carbon emissions?. Technological Forecasting & Social Change. 2012, 79 (5) :919-931
- Asongu, S.A.; Roux, S. L.; Biekpe, N. Enhancing ICT for environmental sustainability in sub-Saharan Africa. Technological Forecasting & Social Change. 2018, 127: 209-21.
- Ilhan, O.; Sana, U. Does digital financial inclusion matter for economic growth and environmental sustainability in OBRI economies? An empirical analysis. Resources, Conservation and Recycling. 2022, 185:106489.
- Salahuddin, M.; Alam, K. Internet usage, electricity consumption and economic growth in Australia: a time series evidence. Telematics and Informatics. 2015, 32 (4): 862-878.
Point 5:
Are the methods adequately described?——Can be improved.
Response 5:
Thank you very much for your comments. We fully agree with you. Indeed, a detailed description of the method of the article can help readers better understand the operation process of the article. Therefore, based on your suggestions, we have mainly made the following modifications. First, let's improve Figure 1. Since the construction of the digital economy system in this paper is based on the internal logic of Figure 1, our improvement of Figure 1 can help readers better understand the theoretical basis of the construction of specific indicators of the digital economy system. Figure 1 is mainly based on the definition of digital economy in Bukht and Heeks (2019) and the author's innovation. According to the connotation of digital economy, data is the key factor for digital economy to play its role. Therefore, data flow should be a typical phenomenon throughout the entire digital economy. In view of this, we will integrate the manifestation of data flow and make it run through the whole process of the three subsystems of the digital economy. On the one hand, this is conducive to reflecting the role of data elements, on the other hand, it is also conducive to showing its relationship with the digital economy from the side. In addition, in order to highlight the marginal contribution of the author in drawing Figure 1, we have supplemented the image source in the annotation section.
Second, we re-described the calculation method of carbon emission reduction. Carbon emission reduction system is one of the important research objects in this paper, so it is necessary to introduce the carbon emission reduction calculation process in detail. We found that the previous description of the consumption of natural gas, oil and electricity was unclear, so we restated it here.
Finally, we supplemented the collection methods of all indicators. We explained the collection method below Table 1 by means of annotation. On the one hand, we can clarify the data search method and provide ideas for readers to find the same data. On the other hand, it can also better reflect the workload of this paper. The specific modifications are as follows:
Line 235-238:
Source: Figure 1 made by the author.
Line 307-310: In Formula (1), is the total carbon emissions of prefecture level cities, is the carbon emissions of natural gas consumed by industry, is the carbon emissions of liquefied petroleum gas consumed by industry and is the carbon emissions of electricity consumed by industry.
Line 325-327: Note: (a1) data and (a5) data in elements of digital infrastructure are obtained through python and manual collection respectively. And (b5) data in carbon emissions reduction system is obtained through manual collection.
Point 6:
Are the results clearly presented?——Can be improved.
Response 6:
Thank you for your suggestion. We checked the full text and found that the results presented in Table 7 in Part IV need to be improved. We agree that a high-quality article should clearly present the research results, and the presentation of the most valuable research results is the key to clearly present the research results. Table 7 mainly reflects the results of regional difference sources of digital economy and carbon emission reduction coupling and coordination. The analysis of difference sources of coupling and coordination is one of the three important parts of this paper. Therefore, in order to present the results of difference sources more clearly, we have readjusted the contents of Table 7, retained the core contents, and deleted the rest. At the same time, we explained it to the readers through notes. If the readers are interested in asking for all the results, they can contact the author. The specific modifications are as follows:
Line 799-803:
Table 7. Analysis of national and regional differences in the coupling and coordination of digital economy development and carbon emissions reduction.
|
Year |
2011 |
2012 |
2013 |
2014 |
2015 |
2016 |
2017 |
2018 |
|
|
Gaps inside region |
Overall |
0.1985 |
0.2026 |
0.2029 |
0.2000 |
0.1974 |
0.1992 |
0.2200 |
0.2106 |
|
the northeast economic zone |
0.1277 |
0.1439 |
0.1468 |
0.1386 |
0.1542 |
0.1268 |
0.1646 |
0.1426 |
|
|
the northern economic zone |
0.1222 |
0.1491 |
0.1547 |
0.1520 |
0.1507 |
0.1482 |
0.1383 |
0.1438 |
|
|
the eastern economic zone |
0.0588 |
0.0516 |
0.0637 |
0.0693 |
0.0711 |
0.0732 |
0.0787 |
0.0706 |
|
|
the southern economic zone |
0.1418 |
0.1384 |
0.1464 |
0.1536 |
0.1388 |
0.1418 |
0.1353 |
0.1304 |
|
|
middle reaches of the Yellow River zone |
0.1013 |
0.1206 |
0.0987 |
0.1109 |
0.1433 |
0.1319 |
0.1276 |
0.1359 |
|
|
middle reaches of the Yangtze River zone |
0.1301 |
0.1279 |
0.1175 |
0.1115 |
0.0956 |
0.1056 |
0.1121 |
0.1004 |
|
|
the southwest economic zone |
0.1409 |
0.1322 |
0.1571 |
0.1448 |
0.1304 |
0.1537 |
0.1609 |
0.1603 |
|
|
the northwest economic zone |
0.1389 |
0.0970 |
0.1472 |
0.1445 |
0.1534 |
0.1602 |
0.1528 |
0.1664 |
|
|
|
|
|
|
|
|
|
|
|
|
|
Gaps between regions |
the northeast zone and the northwest zone |
0.1398 |
0.1314 |
0.1697 |
0.1456 |
0.1747 |
0.2080 |
0.1793 |
0.2069 |
|
the northern zone and the eastern zone |
0.2144 |
0.2391 |
0.1926 |
0.1949 |
0.1995 |
0.1870 |
0.1755 |
0.1738 |
|
|
the northern zone and the southern zone |
0.2029 |
0.2121 |
0.1963 |
0.2127 |
0.2025 |
0.2030 |
0.1823 |
0.1690 |
|
|
the northern zone and the northwest zone |
0.2002 |
0.1690 |
0.1697 |
0.2301 |
0.2499 |
0.2862 |
0.3480 |
0.3577 |
|
|
the eastern zone and Yangtze River zone |
0.3163 |
0.1625 |
0.2990 |
0.2772 |
0.2659 |
0.2606 |
0.2583 |
0.2550 |
|
|
the southern zone and Yangtze River zone |
0.2858 |
0.2779 |
0.2737 |
0.2734 |
0.2542 |
0.2622 |
0.2471 |
0.2265 |
|
|
Yellow River zone and the northwest zone |
0.1252 |
0.1161 |
0.1293 |
0.1357 |
0.1610 |
0.1862 |
0.2024 |
0.2207 |
|
|
Yangtze River zone and the northwest zone |
0.1444 |
0.1204 |
0.1566 |
0.1540 |
0.1775 |
0.2090 |
0.2611 |
0.2725 |
|
|
the southwest zone and the northwest zone |
0.1807 |
0.1598 |
0.2031 |
0.2047 |
0.2280 |
0.2470 |
0.2942 |
0.2973 |
|
|
Contribution rate |
|
|
|
|
|
|
|
|
|
|
Within the region |
7.9200 |
8.0105 |
8.2580 |
8.3493 |
8.3617 |
8.4053 |
7.8096 |
7.9497 |
|
|
Between regions |
74.3517 |
73.3009 |
71.7543 |
71.9802 |
71.7502 |
69.8273 |
75.9021 |
74.2676 |
|
|
Hypervariable density |
17.7283 |
18.6886 |
19.9877 |
19.6705 |
19.8881 |
21.7674 |
16.2883 |
17.7827 |
|
Table 7 calculated by the authors. Table 7 only presents the core content. If readers need information about the gaps between other regions, please ask the author for it.
Point 7:
Are the conclusions supported by the results?——Can be improved.
Response 7:
Thank you for your patient review and suggestions. All the authors checked the correspondence between the conclusion and the research results, and found that the conclusion part can also present the research content more concisely, so as to avoid duplication of expression with the research content part and improve the reading interest of readers. In addition, in order to make the whole article more enlightening, we also add the limitations of this study and the areas that can be broken through in the future in the conclusion part. Therefore, according to your suggestions and in combination with the research content of the article, we restated the conclusion. The specific modifications are as follows:
Line 840-866: Due to the limitation of data disclosure, this paper focuses on
the measurement of digital economy development from 2011 to 2018. In the long-term, this may produce certain restrictions on the identification of more significant features of the evaluation results and the capture of development stability. However, according to the research in this paper, China's digital economy is developing rapidly, which makes it possible to supplement the data in the future and improve the long-term dynamic research in this paper. Through systematic research, the research conclusions are as follows:
- The interactive promotion effect and interdependence between China's digital
economy development and carbon emissions reduction have gradually increased, which means that the coupling and coordinated has shown an increasing trend. The mutual promotion effect of the two systems between regions shows the characteristics of "the east is better than the west" and "the coast is better than the inland". The coupling and coordination degree of different provinces and city types is unbalanced. The above phenomenon reflects the mutual promotion effect of China's digital economy and car-bon emissions reduction still have great potential for improvement.
- It was found that the overall difference of the coupling and coordination be-
tween China's digital economy development and carbon emissions reduction shows an expanding trend, which is reflected within and between regions. Especially among regions, the gap between developing regions and developed regions has widened significantly. And the regional difference is the main source of the overall difference of the coupling and coordination.
- On the whole, the spatial "spillover" effect of the interaction between China's
digital economy development and carbon emission reduction has been obtained. The improvement of the coupling and coordination degree of the two systems in various regions will have an indirect impact on the surrounding regions. The development of the coupling and coordination degree of the two systems has a significant spatial correlation.
Point 8:
Comments and Suggestions for Authors
Dear authors,
Please find my observations below:
The manuscript needs professional English proofreading in order to improve message delivery.
Response 8:
Thank you very much for your advice. All authors fully agree with you. We carefully checked the article and found syntax errors in the model design part, so we corrected them. In order to ensure the professionalism of the full text language, we submitted the article to the language experts for modification, so as to avoid language errors in this article. The specific modifications made by the authors are as follows:
Line 306-309: In Formula (1), is the total carbon emissions of prefecture level cities, is the carbon emissions of natural gas consumed by industry, is the carbon emissions of liquefied petroleum gas consumed by industry and is the carbon emissions of electricity consumed by industry.
Point 9:
Abstract: it is a bit too long. Try to be more synthetic and express your ideas in more concise phrases.
Response 9:
Thank you for your patient review and guidance. Indeed, as you said, the abstract section of this article needs to present the content concisely. Based on your suggestions, the summary of the revised version includes a brief review of existing research, the purpose of this study, the method of use, the results and the significance of the study. We have simplified the language expression, so that the abstract can more clearly convey information and meet the length requirements of International Journal of Environmental Research and Public Health publication. The number of words in the modified abstract is limited to 216 words. The specific modifications are as follows:
Line 14-30:Abstract: Previously conducted studies have established that digital economy has a one-way inhibition effect on carbon emissions. Under this background, this paper aims to analyze the coordinated development effect of the interaction between digital economy and carbon emissions reduction. The entropy weight method, coupling and coordination degree model, Dagum Gini coefficient and Moran's I index have been carried out as research methods in this paper. The results showed that: (1) The coupling and coordination of China's digital economy and carbon emissions reduction shows an overall growth trend, but the coupling and coordination among regions, provinces and cities show a large imbalance. (2) In the sample period, the overall difference of the coupling and coordination between digital economy development and carbon emissions reduction shows an expanding trend, and the overall difference results are attributed to regional differences. (3)There is a significant spatial correlation in the coupling and coordination degree of digital economy development and carbon emissions reduction among cities. The paper systematically grasps the status of coupling and coordination development, the source of difference and spatial correlation between digital economy and carbon reduction in Chinese cities. A dependence relationship has been established which is digital economy development and carbon emissions reduction, and an interactive promotion pattern has been revealed between digital economy system and carbon emissions reduction system.
Point 10:
“Data” is plural, so you have to state “Data show that…”.
Response 10:
Thank you very much for your advice. We rechecked the full text and updated the incorrect use of data syntax in the full text to improve the standardization of the article language. The specific modifications are as follows:
Line 84: Data show that the market size of China's digital economy has increased from 31.3 trillion yuan in 2018 to 40.5 trillion yuan in 2020.
Line 346: In order to ensure the consistency of data set in terms of quantity and dimension, this paper first gives priority to the standardized processing of data.
Point 11:
Format references according to journal requirements.
Response 11:
Thank you very much. We checked the articles that have been published in International Journal of Environmental Research and Public Health. Indeed, as you said, the citation format of references in the articles needs to be revised. According to the requirements of the journal, the author information corresponding to the references in the article has been removed, and only the document number has been retained. At the same time, we carefully checked the correspondence between the serial number of the literature and the information in the references to ensure that the reader can correctly find the relevant author information in the last part. The specific modifications are as follows:
Line 46-49: The digital economy relies on technological innovation to continuously promote industrial integration [1] and economic restructuring, while the transformation of industrial and economic development mode will directly cause changes in carbon dioxide emissions [2].
Line 67-71: Relying on the new digital infrastructure, the digital economy takes knowledge and information as the key production factors, modern information network as the carrier, and the use of information technology as the key driving technology for development, which profoundly affects and fundamentally changes the economic development mode and social activity structure [3].
Line 76-77: Coordination enables the two systems to work together effectively and sustainably [4,5].
Line 82-83: This reflection of the connection between the two systems is coupling [6].
Line 93-94: There is a long way to go to promote the carbon emissions reduction process [7].
Line 100-113: It is believed that the real digital economy refers to the part of economic output completely determined by digital technology, with a business framework based on digital services and goods [8]. With the integration of communication and computing technologies in the network and the flow of technology and data, it is strengthening the transformation of e-commerce and large-scale business [9]. Computer aided data flow has enabled digitization to transform various parts of the current economy [10]. Therefore, the digital economy is also divided into four different parts: digital services and goods; Mixed digital services and goods; IT based intensive production services; And IT industry [11].It is believed that the digital economy is an economic activity based on data application and data technology innovation, with data as the core element [12]. Data technology is the basic support for the development of digital economy. It is a new general technology, mainly through the combination of digital information and the Internet [13], including hardware, software and network technology [14]. There is an opinion that digital economy is a more advanced economic form after industrial economy [15].
Line 114-124: Most of the existing literatures measure the digital economy based on the method of summarizing the digital industry output, digital product classification, and digital related comprehensive indicators. Scholar first discussed the boundary problem of digital economy, which provides a useful reference for measuring digital economy [16]. The digital economy is also divided into five parts for measurement, namely infrastructure, e-commerce, industry structure, digital labor and digital price [17]. Some organizations have proposed an accounting framework with intelligent infrastructure investment, social promotion, innovation release, growth and employment as the main body to measure the digital economy [18]. In addition, there are also scholars who calculate the digital economy by building a comprehensive indicator system [19-21].
Line 126-153: The calculation of carbon dioxide emissions is mainly based on different energy consumption. China's industrial energy carbon emissions have been calculated based on industrial energy consumption [22]. Based on the IPCC inventory method, a research uses the multiplication method of different fuel consumption and fuel carbon emission factors to calculate the regional power carbon emissions [23]. A study took a telecommunications company in Slovenia as the research object, calculated the greenhouse gas emissions within the enterprise scope 3, and took the purchase of electricity, the commuting of employees, the use of vehicles owned by the company and heating as the measured objects [24]. In addition, the research calculates the grid emission factors, deduces the emission factors of specific European countries, and calculates the carbon intensity based on the power generation of power plants [25]. The other is the exploration of carbon emissions reduction path. It is generally believed that economic development is closely related to carbon emissions [26]. The exploration of carbon emissions reduction path mainly involves the improvement of carbon emissions trading market [27,28], the analysis of the policy effect of carbon emissions reduction pilot [29,30] and the research of other paths [31]. In the study of other carbon emissions reduction paths, there is other scholar believed that carbon capture, utilization and storage technology (CCUS) is the only technological path to achieve net zero emissions [32]. And another research believed that the promotion of " Nature Based Solution" (NBS) can provide positive reference for China to solve climate and environmental problems [33]. Mean-while, there is also study carried out from the perspective of energy, believing that the development of renewable energy is of great significance for carbon emissions reduction in resource-based countries [34]. Another study assessed the existing climate policy portfolio and believed that the energy technology portfolio needs to be adjusted [35]. Some scholars discussed the environmental impact of mineral mining based on the mining industry [36,37]. The adjustment of savings also contributes to the development of green economy [38]. In addition, international climate cooperation and climate assistance can also effectively achieve carbon emissions reduction targets [39].
Line 155-171: One is the impact of digital economy development on carbon emissions. Some scholars believe that the digital economy can inhibit carbon emissions [40-42]. A study have proved the role of Taiwan's digital development in the circular economy. It is found that through digital waste management, the consumption of raw materials can be reduced by 25% by 2030, and half of greenhouse gas emissions can be avoided [43]. There is study believed that ICT can contribute to the construction of low-carbon cities [44], mainly through governance capacity of the government [45] and energy systems [46]. Study have also targeted African countries and found an inverted "U" relationship between ICT and carbon emissions in the sample countries [47]. In addition, some scholars believe that R&D investment played a positive role in regulating the relationship between digital economy development and carbon emissions reduction [48-50]. It can be seen that the development of the digital economy has a positive contribution to the promotion of China's green development process [51]. Therefore, it is necessary to study the impact of the digital economy on the climate and environment, which has reference significance for the formulation and improvement of economic and environmental policies [52]. Different from this conclusion, other scholars believe that the development of digital economy is unfavorable to carbon emission reduction [53,54].
Line 176-177: It can be considered that low carbon development can help reverse enterprises' digital transformation [55].
Line 185-187: The application of digital technology can optimize the end treatment technology of enterprise carbon emissions [56,57], and promote carbon emission reduction [58,59].
Line 224-225: For the digital economy, there is scholar believed that there are three layers from the inside to the outside [60].
Line 251-252: The number of enterprises based on Internet development can represent the application level of digital infrastructure [61].
Line 278-280: Number of Internet users. The number of Internet users can reflect the development of digital economy related businesses [62].
Line 287-288: Therefore, it can be calculated according to the digital inclusive financial index of prefecture level cities based on the compilation of indicators by [63].
Line 289-291: The calculation of carbon emissions reduction system mainly refers to the construction method of the green development system, and the sub index construction is based on the input-output model [64]. Low carbon is one of the core concepts of green development [65], so it is reasonable to use the green development system for reference in the construction of carbon emissions reduction system.
Line 301-304: For the carbon dioxide emissions accounting of prefecture level cities, this paper selects the carbon emissions of industrial consumption of natural gas, liquefied petroleum gas and industrial coal-fired electricity to sum up [66].
Line 338-339: Coordination means that two or more systems are interrelated and properly coordinated to form a mutually beneficial development situation [67,68].
Line 379: carbon emissions reduction system [69].
Line 398-401: Dagum Gini coefficient and its decomposition method is to decompose the overall Gini coefficient G into intra group (intra regional) differential contribution , inter group (inter regional) difference contribution and hypervariable density contribution difference , three parts [70].
Point 12:
Pay attention when using words between quotation marks. Sometimes you put a blank space after the first quotation mark (see page 2).
Response 12:
Thank you very much for your advice. We totally agree with you, indeed. A standard format is an essential element of a high-quality article. Therefore, with your advice, we checked the quotation mark format of the full text and corrected the problems. The specific modifications are as follows:
Line 49-51: At the same time, under background of "emission peak" and "carbon neutrality", the digital regulation on which the carbon emissions reduction process is based will also have a certain impact on the development of the digital economy.
Point 13:
Page 3: pay attention when referring to studies in the literature. “Pei et al.(2018) believes..” should be “Pei at al. (2018) believe that..”, since there is more than one author in this study. Apply this in the entire paper.
Response 13:
Thank you very much for your advice. We ignored the quotation of multiple authors in the writing process. Thank you for your reminding. Since we modified the citation format and finally cited the full text literature by serial number, the author's name did not appear in the revised version, thus avoiding incorrect citation of multiple authors. Thank you for your patient guidance. The specific modifications are as follows:
Line 46-49: The digital economy relies on technological innovation to continuously promote industrial integration [1] and economic restructuring, while the transformation of industrial and economic development mode will directly cause changes in carbon dioxide emissions [2].
Line 67-71: Relying on the new digital infrastructure, the digital economy takes knowledge and information as the key production factors, modern information network as the carrier, and the use of information technology as the key driving technology for development, which profoundly affects and fundamentally changes the economic development mode and social activity structure [3].
Line 76-77: Coordination enables the two systems to work together effectively and sustainably [4,5].
Line 82-83: This reflection of the connection between the two systems is coupling [6].
Line 93-94: There is a long way to go to promote the carbon emissions reduction process [7].
Line 100-113: It is believed that the real digital economy refers to the part of economic output completely determined by digital technology, with a business framework based on digital services and goods [8]. With the integration of communication and computing technologies in the network and the flow of technology and data, it is strengthening the transformation of e-commerce and large-scale business [9]. Computer aided data flow has enabled digitization to transform various parts of the current economy [10]. Therefore, the digital economy is also divided into four different parts: digital services and goods; Mixed digital services and goods; IT based intensive production services; And IT industry [11].It is believed that the digital economy is an economic activity based on data application and data technology innovation, with data as the core element [12]. Data technology is the basic support for the development of digital economy. It is a new general technology, mainly through the combination of digital information and the Internet [13], including hardware, software and network technology [14]. There is an opinion that digital economy is a more advanced economic form after industrial economy [15].
Line 114-124: Most of the existing literatures measure the digital economy based on the method of summarizing the digital industry output, digital product classification, and digital related comprehensive indicators. Scholar first discussed the boundary problem of digital economy, which provides a useful reference for measuring digital economy [16]. The digital economy is also divided into five parts for measurement, namely infrastructure, e-commerce, industry structure, digital labor and digital price [17]. Some organizations have proposed an accounting framework with intelligent infrastructure investment, social promotion, innovation release, growth and employment as the main body to measure the digital economy [18]. In addition, there are also scholars who calculate the digital economy by building a comprehensive indicator system [19-21].
Line 126-153: The calculation of carbon dioxide emissions is mainly based on different energy consumption. China's industrial energy carbon emissions have been calculated based on industrial energy consumption [22]. Based on the IPCC inventory method, a research uses the multiplication method of different fuel consumption and fuel carbon emission factors to calculate the regional power carbon emissions [23]. A study took a telecommunications company in Slovenia as the research object, calculated the greenhouse gas emissions within the enterprise scope 3, and took the purchase of electricity, the commuting of employees, the use of vehicles owned by the company and heating as the measured objects [24]. In addition, the research calculates the grid emission factors, deduces the emission factors of specific European countries, and calculates the carbon intensity based on the power generation of power plants [25]. The other is the exploration of carbon emissions reduction path. It is generally believed that economic development is closely related to carbon emissions [26]. The exploration of carbon emissions reduction path mainly involves the improvement of carbon emissions trading market [27,28], the analysis of the policy effect of carbon emissions reduction pilot [29,30] and the research of other paths [31]. In the study of other carbon emissions reduction paths, there is other scholar believed that carbon capture, utilization and storage technology (CCUS) is the only technological path to achieve net zero emissions [32]. And another research believed that the promotion of " Nature Based Solution" (NBS) can provide positive reference for China to solve climate and environmental problems [33]. Mean-while, there is also study carried out from the perspective of energy, believing that the development of renewable energy is of great significance for carbon emissions reduction in resource-based countries [34]. Another study assessed the existing climate policy portfolio and believed that the energy technology portfolio needs to be adjusted [35]. Some scholars discussed the environmental impact of mineral mining based on the mining industry [36,37]. The adjustment of savings also contributes to the development of green economy [38]. In addition, international climate cooperation and climate assistance can also effectively achieve carbon emissions reduction targets [39].
Line 155-171: One is the impact of digital economy development on carbon emissions. Some scholars believe that the digital economy can inhibit carbon emissions [40-42]. A study have proved the role of Taiwan's digital development in the circular economy. It is found that through digital waste management, the consumption of raw materials can be reduced by 25% by 2030, and half of greenhouse gas emissions can be avoided [43]. There is study believed that ICT can contribute to the construction of low-carbon cities [44], mainly through governance capacity of the government [45] and energy systems [46]. Study have also targeted African countries and found an inverted "U" relationship between ICT and carbon emissions in the sample countries [47]. In addition, some scholars believe that R&D investment played a positive role in regulating the relationship between digital economy development and carbon emissions reduction [48-50]. It can be seen that the development of the digital economy has a positive contribution to the promotion of China's green development process [51]. Therefore, it is necessary to study the impact of the digital economy on the climate and environment, which has reference significance for the formulation and improvement of economic and environmental policies [52]. Different from this conclusion, other scholars believe that the development of digital economy is unfavorable to carbon emission reduction [53,54].
Line 176-177: It can be considered that low carbon development can help reverse enterprises' digital transformation [55].
Line 185-187: The application of digital technology can optimize the end treatment technology of enterprise carbon emissions [56,57], and promote carbon emission reduction [58,59].
Line 224-225: For the digital economy, there is scholar believed that there are three layers from the inside to the outside [60].
Line 251-252: The number of enterprises based on Internet development can represent the application level of digital infrastructure [61].
Line 278-280: Number of Internet users. The number of Internet users can reflect the development of digital economy related businesses [62].
Line 287-288: Therefore, it can be calculated according to the digital inclusive financial index of prefecture level cities based on the compilation of indicators by [63].
Line 289-291: The calculation of carbon emissions reduction system mainly refers to the construction method of the green development system, and the sub index construction is based on the input-output model [64]. Low carbon is one of the core concepts of green development [65], so it is reasonable to use the green development system for reference in the construction of carbon emissions reduction system.
Line 301-304: For the carbon dioxide emissions accounting of prefecture level cities, this paper selects the carbon emissions of industrial consumption of natural gas, liquefied petroleum gas and industrial coal-fired electricity to sum up [66].
Line 338-339: Coordination means that two or more systems are interrelated and properly coordinated to form a mutually beneficial development situation [67,68].
Line 379: carbon emissions reduction system [69].
Line 398-401: Dagum Gini coefficient and its decomposition method is to decompose the overall Gini coefficient G into intra group (intra regional) differential contribution , inter group (inter regional) difference contribution and hypervariable density contribution difference , three parts [70].
Point 14:
What is the source of Figure 1? Please indicate the source under this figure.
Response 14:
Thank you very much for your guidance. We fully agree that the image source you said needs to be clarified. In fact, Figure 1 is made by the author, mainly referring to the definition of digital economy in Bukht and Heeks (2019), and has been innovated and improved on the basis. At your prompt, we have explained the source of Figure 1. The specific modifications are as follows:
Line 237: Source: Figure 1 made by the author.
Reference:
Bukht, R.; Heeks, R. Defining, conceptualising and measuring the digital economy. Development Informatics Working Paper. 2019.
Point 15:
Page 7: when presenting variables, you should use the verb in singular. Currently you use a mix of both singular and plural. (“CO2 is the total carbon emissions….electricCO2 are the carbon…”).
Response 15:
Thank you for your suggestion. We have checked this place and fully agree with you that variables should be presented in singular form. Therefore, according to your suggestions, we have made modifications. The specific modifications are as follows:
Line 306-309: In Formula (1), is the total carbon emissions of prefecture level cities, is the carbon emissions of natural gas consumed by industry, is the carbon emissions of liquefied petroleum gas consumed by industry and is the carbon emissions of electricity consumed by industry.
Point 16:
In the last section, please add study limitations, which currently are missing.
Response 16:
Thank you very much for your advice. We fully agree with you. Indeed, a high-quality paper needs to explain the limitations of existing research and possible future breakthroughs. Therefore, according to your suggestions, we have supplemented the limitations and future development directions of this article in the conclusion section. The specific modifications are as follows:
Line 841-847: Due to the limitation of data disclosure, this paper focuses on the measurement of digital economy development from 2011 to 2018. In the long-term, this may produce certain restrictions on the identification of more significant features of the evaluation results and the capture of development stability. However, according to the research in this paper, China's digital economy is developing rapidly, which makes it possible to supplement the data in the future and improve the long-term dynamic research in this paper.
Point 17:
The reference list should be expanded with the following titles:
Batrancea, L., Rathnaswamy, M.M., Batrancea, I., Nichita, A., Rus, M.-I., Tulai, H., Fatacean, G., Masca, E.S., & Morar, I.D. (2020). Adjusted net savings of CEE and Baltic nations in the context of sustainable economic growth: A panel data analysis. Journal of Risk and Financial Management, 13(10), 234.
Batrancea, I., R athnaswamy, M.K., Batrancea, L., Nichita, A., Gaban, L., Fatacean, G., Tulai, H., Bircea, I., & Rus, M.-I. (2020). A panel data analysis on sustainable economic growth in India, Brazil, and Romania. Journal of Risk and Financial Management, 13(8), 170.
Response 17:
Thank you very much for your patient guidance and advice. Thank you for the two international articles you recommended, which will help improve the internationalization level of this article and attract international readers. We carefully read these two articles and added them to the introduction. Specifically, these two articles are respectively placed in the exploration of carbon emission reduction paths and other emission reduction paths. The specific modifications are as follows:
Line 137-138: It is generally believed that economic development is closely related to carbon emissions [26].
Line 151-152: The adjustment of savings also contributes to the development of green economy [38].
Thank you very much for your patient guidance. We carefully reviewed this article, and learned the very targeted suggestions you gave together for several rounds, aiming to improve the quality of this paper so that it can meet the international journal publishing requirements. All the authors would like to express their sincere thanks to you for your patient review and detailed theoretical guidance for the improvement of this article. It is your careful guidance and help that we have the opportunity to improve this paper, and your suggestions will help us avoid premature ideas in the future paper writing. All the suggestions you put forward have given us great inspiration for this article and the writing of future papers, and let us understand the characteristics of a scientific and interesting high-quality international article. Such valuable guidance opportunities are very difficult to meet in our daily writing, and all authors cherish them very much. Finally, all the authors sincerely thank you for your valuable time to help and support us, so that we can make continuous progress. Thank you very much!
Thanks very much to the reviewers and editors for your valuable comments and suggestions, so that the quality of the manuscript can be improved. If there is also some improper expression in the revised manuscript, please let us know and we will further improve it.

Round 2
Reviewer 1 Report
The manuscript "Analysis on the Coupling Effect and Space-Time Difference between China's Digital Economy Development and Carbon Emissions Reduction" by Nan Li, Beibei Shi, Rong Kang was submitted for second review.
As can be seen from the submitted manuscript and the explanatory note to the review, the authors did a lot of work to make changes in accordance with the comments.
The revised manuscript is a completed scientific study on a highly relevant topic: Carbon Emissions Reduction. The revised version of the manuscript, in my opinion, fully satisfies the requirements of a scientific article and can be published in the open press.